# CITRATE 1.0: Phytoplankton continuous trait-distribution model with one-dimensional physical transport applied to the Northwest Pacific

5 Bingzhang Chen, S. Lan Smith

Research Center for Global Change Research, JAMSTEC (Japan Agency for Marine-Earth Science and Technology), 3173-25 Showa-machi, Kanazawa-ku, Yokohama 236-0001, Japan

10 *Correspondence to*: Bingzhang Chen (bzchen@jamstec.go.jp)

**Abstract.** Diversity plays critical roles in ecosystem functioning, but it remains unclear how best to model phytoplankton diversity in order to better understand those roles and reproduce consistently observed patterns in the ocean. In contrast to the typical approach of resolving distinct species or functional groups, we present a ContInuous TRAiT-basEd phytoplankton model (**CITRATE**) that
focuses on macroscopic system properties such as total biomass, mean trait values, and trait variance. This phytoplankton component is embedded within a Nitrogen-Phytoplankton-Zooplankton-Detritus-Iron model that itself is coupled with a simplified one-dimensional ocean model. Size is used as the master trait for phytoplankton. CITRATE also incorporates "trait diffusion" for sustaining diversity, as well as simple representations of physiological acclimation, i.e. flexible chlorophyll-to-carbon and
nitrogen-to-carbon ratios. We implemented CITRATE at two contrasting stations in the Northwest Pacific where several years of observational data are available. The model is driven by physical forcing including vertical eddy diffusivity imported from three-dimensional general ocean circulation models (GCMs). One common set of model parameters for the two stations was optimized using the Delayed Rejection Adaptive Metropolis-Hasting Monte Carlo (DRAM) algorithm. The model faithfully
reproduced most of the observational patterns and gave robust predictions on phytoplankton mean size and size diversity. CITRATE is a prototype upon which more sophisticated continuous trait-based models and applications within GCMs can be developed.

# 1 Introduction

Both species identity (functional traits) and diversity play critical roles in ecosystem functioning (Tilman et al., 1997, 2014). Phytoplankton are a polyphyletic group of oxygenic organisms that account for nearly half of the global primary production (Fields et al., 1998) and also play indispensable roles in other biogeochemical cycles in the Earth system (Falkowski, 2012). They have astonishingly high diversity, with several thousand species already documented and many remaining to be explored (Sournia et al., 1991; Moon-van der Staay et al., 2001). Their equivalent spherical diameter (ESD) can range from less than one micron for cyanobacteria such as *Prochlorococcus* (Chisolmn et al., 1988) to more than 1 mm for some giant diatoms (Villareal, 1993). Furthermore, physiology differs substantially even within the same genera or species and the role of intraspecific variability in population dynamics and biogeochemical cycles remains to be investigated (Strzepek and Harrison, 2004; Johnson et al., 2006; Palenik et al., 2006; Kooistra et al., 2008; Biller et al., 2015).

Although various ocean models have been developed by accounting for different functional groups or categories of phytoplankton such as cyanobacteria, diatoms, diazotrophs, etc. (e.g., Le Quéré et al., 2005; Hashioka et al., 2013), the finite number of such distinct types included limits their ability to resolve the vast diversity of trait values. Some pioneering studies have considered greater numbers of species, each of which is defined by particular set of multivariate trait axes that constitute a hyper-volume niche space (Follows et al., 2007; Barton et al., 2010; Follows and Dutkiewicz, 2011; Matsuda et al., 2016). (It is worth noting that these diversity models usually focus on "functional traits" which are the key to linking phytoplankton diversity, environmental conditions, and ecosystem functioning. Important phytoplankton traits include maximal growth rate, the light absorption and nutrient uptake affinities, optimal growth temperature, and edibility (i.e., susceptibility to grazing), etc (Litchman et al., 2007; Litchman and Klausmeier, 2008; Edwards et al., 2011, 2012, 2015; Merico et al., 2009; Thomas et al., 2012; Chen, 2015)). The total species pool in these modelling studies should ideally cover the entire multi-dimensional trait space constrained by trade-offs (Smith et al. 2011), although computational limits make this impossible in practice. As a compromise, only a limited set of trait combinations is sampled from the entire trait space. Although this approach has effectively generated large-scale patterns of plankton diversity, it generally underestimates local diversity, for two reasons: 1)

lack of appropriate mechanisms for sustaining diversity (but *see* Vallina et al., (2014)), and 2) insufficient trait resolution so that fitness differences between species are too large to allow coexistence (i.e. insufficient equalizing effect; *see* Chesson, 2000). In any case, a substantial proportion of the idealized species so modelled cannot survive under realistic oceanic conditions, and therefore the

models do not capture the functions associated with many species.

Continuous trait-based models have been developed to address the above questions (Wirtz and Eckhardt, 1996; Norberg et al., 2001; Bruggeman, 2009; Merico et al., 2009, 2014; Terseleer et al., 2014; Acevedo-Trejos et al., 2016; Smith et al., 2016). Instead of modeling the dynamics of individual species, continuous trait-based models or so-called "adaptive dynamics" models focus on macroscopic

or aggregate properties of a community such as total biomass, average trait, and trait variance by assuming that phytoplankton traits follow some distribution (usually Gaussian) (Smith et al., 2011). These models do not have the problem of inadequate trait resolution, because they have infinitesimally fine trait resolution. The trait variance, treated as a tracer in the model, serves as a measure of trait diversity; although it cannot be simply equated to species richness, it can be converted to other diversity

metrics such as the continuous entropy (Quintana et al., 2008). The diversity of functional traits is arguably a better diversity index than species richness relating to ecosystem functioning (Loreau et al., 2001). Thus, the continuous trait-based model has the advantage that the factors controlling diversity can be directly quantified and better understood because the sources (e.g. speciation or immigration) and sinks (e.g. resource competition) for diversity are specified explicitly. In addition, these models are

computationally much more efficient than classic discrete species approaches. For example, assuming two independent traits for the phytoplankton community, a continuous trait-based model only requires 1 (biomass) + 2x2 (trait mean and variance) = 5 tracers for the phytoplankton community, while a discrete species-based model requires 2x10 = 20 tracers if assuming ten discrete values in each trait dimension, which still provides only coarse trait resolution. Furthermore, this difference increases linearly with trait

dimension.

Relatively few continuous trait-based models have been coupled with physics transport and calibrated against oceanic observations. Here we describe a new one-dimensional model, **CITRATE 1.0,** built upon the classic nitrogen-phytoplankton-zooplankton-detritus (NPZD) model with a

phytoplankton community represented using a continuous distribution of size, taken as a master trait (Fig. 1). In this way, not only total phytoplankton biomass, but also phytoplankton mean size and size variance are explicitly modeled. The distributions of other important functional traits are implicitly modeled via well-established scaling power laws. Although this approach might overlook some other important traits that are not related to size and thereby underestimate trait diversity to some extent, it serves as a starting point for later development of more comprehensive diversity models that can include more traits or be integrated with the discrete functional group approach. For the model to be implemented in the subarctic North Pacific, a well-known high nitrate low chlorophyll (HNLC) region, **CITRATE** also incorporates an iron limitation module. We optimized the model parameters against the extensive observational data at two contrasting stations (K2: 160 ºE, 47 ºN; S1: 145 ºE, 30ºN) in the North Pacific (Fig. 2a). The station K2 is located within the western subarctic North Pacific gyre and is characterized by low temperature, high nitrate, and high carbon export (Matsumoto et al., 2014; Wakita et al., 2016). Iron limitation on phytoplankton growth has been suggested at this station (Fujiki et al., 2014). The station S1 is located within the western subtropical North Pacific and is characterized by high sea surface temperature, low levels of nitrate and carbon export efficiency (Matsumoto et al., 2016; Sasai et al., 2016; Wakita et al., 2016). To independently validate the model, we also use the optimized model parameters from stations K2 and S1 to run the model for station ALOHA (158 ºW, 22.75 ºN) and compare the model outputs with the extensive observational data collected there.

In the following sections, we first describe the details of the model structure and the parameter optimization subroutine. Then we show the results of parameter optimization and modeled patterns of nutrients, phytoplankton biomass, mean size and size diversity. We also discuss the merits and limitations of the model and of the continuous trait-distribution approach. **CITRATE** is intended as a prototype for later incorporation into three-dimensional (3D) general ocean circulation models (GCMs) and for further development of more comprehensive trait-based models.

## 2 Model description

The overall aim of **CITRATE 1.0** is to implement the continuous trait-based approach for modeling phytoplankton size diversity under realistic environmental conditions at the two representative stations

in the North Pacific. And our goal is not only to model phytoplankton size structure, but also to faithfully simulate the seasonal and vertical dynamics of other important quantities such as nutrients, Chl $a$, and productivity for later investigations of the roles of phytoplankton diversity in biogeochemical cycles. Hence, **CITRATE 1.0** consists of the following key features:

1)  It models the mean and variance of a continuous phytoplankton size (i.e. log cell volume ($\mu$m$^3$)) distribution and incorporates "trait-diffusion" to sustain size diversity (Merico et al., 2014).

   2)  It contains an iron cycle in addition to the nitrogen cycle because in the subarctic and equatorial Pacific iron instead of nitrogen should be the main limiting nutrient for phytoplankton growth (Behrenfeld et al., 2006; Fujiki et al., 2014).

3)  The phytoplankton cells have variable chlorophyll-to-carbon ($\theta$) and nitrogen-to-carbon ($Q_N$) ratios that respond to light and nutrient conditions in a realistic fashion.

   4)  A single set of model parameters are optimized against field observational data at two time-series stations in the Northwest Pacific.

**2.1 Description of the ecosystem model**

**CITRATE 1.0** contains 9 tracers in total: dissolved inorganic nitrogen (DIN, abbreviated as $N$ in all the equations; unit: $\mu$mol N L$^{-1}$), phytoplankton biomass ($P$; $\mu$mol N L$^{-1}$), microzooplankton biomass ($MIC$; $\mu$mol N L$^{-1}$), mesozooplankton biomass ($MES$; $\mu$mol N L$^{-1}$), detritus in terms of nitrogen ($D$; $\mu$mol N L$^{-1}$) and iron ($D_{Fe}$; nmol Fe L$^{-1}$), dissolved iron ($fer$; nmol Fe L$^{-1}$), the products of $P\bar{l}$ and $P(v + \bar{l}^2)$ where $\bar{l}$ (ln $\mu$m$^3$) is the phytoplankton mean log cell volume and $v$ ((ln $\mu$m$^3$)$^2$) is the

log volume variance (Fig. 1).

     We assume that phytoplankton size is the master trait that determines all physiological functions (Litchman et al., 2007; Finkel et al., 2010; Edwards et al., 2011, 2012, 2015). We also assume that phytoplankton size follows a lognormal distribution, which is supported by some observational data (Finkel, 2007; Quintana et al., 2008). Since $\bar{l}$ and $v$ are not real standing stocks that can be directly

transported in hydrodynamic models but are emergent properties of phytoplankton size structure, we follow Bruggeman (2009) to use the raw moments of biomass probability (i.e. $P\bar{l}$ and $P(v + \bar{l}^2)$ for

mean and variance) as independent tracers involved in transport. All the assumptions made here will be discussed later in **Sect.** 4.

Below we will describe the equations for each tracer. For simplicity, phytoplankton cells are assumed not to excrete inorganic nitrogen or to have any natural mortality to be converted into detritus.

5   Phytoplankton are eaten by both micro- and mesozooplankton:

$$\frac{dP}{dt} = P\mu_{com} - e^{\frac{E_z}{k}\left(\frac{1}{T_0} - \frac{1}{T}\right)} \left[ MICg_{max,1} \frac{P_{T,1}^2}{P_{T,1}^2 + K_{P,1}^2} + MESg_{max,2} \frac{(P_{T,2} + MIC)P}{(P_{T,2} + MIC)^2 + K_{P,2}^2} \right] + \frac{d}{dz}\left(K_v \frac{dP}{dz}\right)$$

(1)

where $\mu_{com}$ is the phytoplankton specific growth rate (d$^{-1}$) of the whole community (i.e. integrated over the whole size spectra). The equation of $\mu_{com}$, along with those of $\bar{l}$ and $\mathbf{v}$, will be described later in Sect. 2.2. $E_z$ is the activation energy (in electron volts [eV], 1 eV = 96.49 kJ mol$^{-1}$) for heterotrophic processes. $g_{max,i}$ ($i$ = 1 for microzooplankton and 2 for mesozooplankton) is zooplankton maximal grazing rate (d$^{-1}$). $K_{P,i}$ is the grazing half-saturation constant of zooplankton. Here we have assumed that zooplankton grazing follows a Holling Type III functional response. $P_{T,i}$ is total palatable prey concentration for zooplankton ($\mu$mol N L$^{-1}$), the details of which will be given later in Sect. 2.3. If zooplankton grazing has no size selectivity on phytoplankton, then $P_T = P$. We assume that microzooplankton preferably feed on small phytoplankton while mesozooplankton prefer large phytoplankton (Table 1). Mesozooplankton also feed on microzooplankton. More descriptions of zooplankton size-dependent grazing will be given later. $z$ is the depth of the model grid (m). $K_v$ is the vertical eddy diffusivity (m$^2$ s$^{-1}$).

The total amount of phytoplankton ingested by zooplankton is divided among three fates: zooplankton net growth, excretion into the inorganic nitrogen pool, and defecation of unassimilated food into the detritus pool (Buitenhuis et al., 2010). Mesozooplankton mortality is set to be proportional to the squares of its biomass and is also converted into detritus pool. As such, the dynamics of micro- and mesozooplankton follow:

$$\frac{dMIC}{dt} = e^{\frac{E_z}{k}\left(\frac{1}{T_0}-\frac{1}{T}\right)} \left(MIC g_{max,1} \frac{P_{T,1}^2}{P_{T,1}^2+K_{P,1}^2} NGE_1 - MES g_{max,2} \frac{(P_{T,2}+MIC)MIC}{(P_{T,2}+MIC)^2+K_{P,2}^2}\right) + \frac{d}{dz}\left(K_v \frac{dMIC}{dz}\right) \quad (2a)$$

$$\frac{dMES}{dt} = e^{\frac{E_z}{k}\left(\frac{1}{T_0}-\frac{1}{T}\right)} \left(MES g_{max,2} \frac{(P_{T,2}+MIC)^2}{(P_{T,2}+MIC)^2+K_{P,2}^2} NGE_2 - m_z MES^2\right) + \frac{d}{dz}\left(K_v \frac{dMES}{dz}\right) \quad (2b)$$

where $NGE_i$ is the net growth efficiency of zooplankton. $m_z$ is the mesozooplankton mortality coefficient (d$^{-1}$ ($\mu$mol N L$^{-1}$)$^{-1}$).

Detritus is converted to DIN at a rate ($R_{dn}$, d$^{-1}$) that has the same temperature sensitivity with zooplankton grazing. Detritus is also assumed to have a constant sinking rate ($W_d$, d$^{-1}$).

$$\frac{dD}{dt} = e^{\frac{E_z}{k}\left(\frac{1}{T_0}-\frac{1}{T}\right)} \left(MICg_{max,1}\frac{P_{T,1}^2}{P_{T,1}^2+K_{P,1}^2}unass_1 + MESg_{max,2}\frac{(P_{T,2}+MIC)^2}{(P_{T,2}+MIC)^2+K_{P,2}^2}unass_2 + m_zMES^2 - \right.$$
$$\left. R_{dn}D\right) - W_d\frac{dD}{dz} + \frac{d}{dz}\left(K_v\frac{dD}{dz}\right) \tag{3}$$

where $unass_i$ represents the fraction of unassimilated food by zooplankton.

DIN is taken up by phytoplankton and is replenished by zooplankton excretion, detritus regeneration and diffusion from the depth:

$$\frac{dN}{dt} = -P\mu_{com} + e^{\frac{E_z}{k}\left(\frac{1}{T_0}-\frac{1}{T}\right)}\left(MICg_{max,1}\frac{P_{T,1}^2}{P_{T,1}^2+K_{P,1}^2}(1-NGE_1-unass_1)\right.$$
$$\left. +MESg_{max,2}\frac{(P_{T,2}+MIC)^2}{(P_{T,2}+MIC)^2+K_{P,2}^2}(1-NGE_2-unass_2) + DR_{dn}\right) + \frac{d}{dz}\left(K_v\frac{dN}{dz}\right) \tag{4}$$

The sources and sinks of $fer$ largely follow DIN with an additional source (atmospheric deposition; Fe$_{depo}$) and sink (scavenging; $fer_{scav}$) (Aumont et al., 2003; Buitenhuis et al., 2010; Nikelsen et al., 2015):

$$\frac{dfer}{dt} = \left[\begin{array}{l} -P\mu_{com} + e^{\frac{E_z}{k}\left(\frac{1}{T_0}-\frac{1}{T}\right)}\left(MICg_{max,1}\frac{P_{T,1}^2}{P_{T,1}^2+K_{P,1}^2}(1-NGE_1-unass_1)\right. \\[2mm] \left. +MESg_{max,2}\frac{(P_{T,2}+MIC)^2}{(P_{T,2}+MIC)^2+K_{P,2}^2}(1-NGE_2-unass_2) + DR_{dn}\right) \end{array}\right] R_{fer_N}$$
$$+Fe_{depo} - fer_{scav} + \frac{d}{dz}\left(K_v\frac{dfer}{dz}\right) \tag{5a}$$

To translate between nitrogen and iron in phytoplankton and zooplankton, a constant $fer$:N ratio ($R_{fer\_N}$) of 0.0265 is assumed. The data of monthly atmospheric deposition of total soluble iron are

extracted from the Scenario III in Luo et al. (2008). Following Nikelsen et al. (2015), iron scavenging rate ($fer_{scav}$) is composed of both linear scavenging rate ($k_{scm}$) and particle absorption rate ($k_{sc}$):

$$fer_{scav} = \left(k_{scm} + k_{sc}De^{\frac{E_z}{k}\left(\frac{1}{T_0}-\frac{1}{T}\right)}\right)Fe_{prime} \tag{5b}$$

in which $Fe_{prime}$ is the concentration of free iron:

$$Fe_{prime} = \frac{\left(-A+\sqrt{4ferk_{eq}+A^2}\right)}{2k_{eq}} \tag{5c}$$

$$A = 1 + \left(l_{fe} - fer\right)k_{eq} \tag{5d}$$

where $k_{eq}$ is the equilibrium constant between free iron and ligands and is assumed to depend only on temperature:

$$k_{eq} = 10^{\left(17.27-\frac{1565.7}{T}\right)} \tag{5e}$$

Note that $T$ is in absolute temperature (K). $l_{fe}$ is the total iron ligand concentration that is assumed constant (0.6 nM).

The equation for $D_{Fe}$ is:

$$\frac{dD_{Fe}}{dt} = e^{\frac{E_z}{k}\left(\frac{1}{T_0}-\frac{1}{T}\right)}\left(MICg_{max,1}\frac{P_{T,1}^2}{P_{T,1}^2+K_{P,1}^2}unass_1 + MESg_{max,2}\frac{(P_{T,2}+MIC)^2}{(P_{T,2}+MIC)^2+K_{P,2}^2}unass_2 +\right.$$

$$\left. m_z MES^2 - R_{dn}D\right)R_{fe_N} - W_d\frac{dD_{Fe}}{dz} + fer_{scav} + \frac{d}{dz}\left(K_v\frac{dD_{Fe}}{dz}\right) \tag{6}$$

## 2.2 Continuous trait-based phytoplankton model

Following the moment closure techniques in Merico et al. (2009) and the introduction of "trait diffusion" (Merico et al., 2014), the equations for $\mu_{com}$, $l$, and $v$ can be written as:

$$\mu_{com} \approx \left.\left(\mu(l) + \frac{v}{2}\left(\frac{d^2\mu(l)}{dl^2} + u\frac{d^4\mu(l)}{dl^4}\right) - 3u\frac{d^3\mu(l)}{dl^3}\right)\right|_{l=\bar{l}} \tag{7a}$$

$$\frac{d\bar{l}}{dt} \approx \left.\left[v\left(\frac{d\mu(l)}{dl} - \sum_{i=1}^2\frac{dg_i(l)}{dl} + u\frac{d^3\mu(l)}{dl^3}\right) - 3u\frac{d\mu(l)}{dl}\right]\right|_{l=\bar{l}} \tag{7b}$$

$$\frac{dv}{dt} \approx \left.\left\{v\left[v\left(\frac{d^2\mu(l)}{dl^2} - \sum_{i=1}^2\frac{d^2g_i(l)}{dl^2} + u\frac{d^4\mu(l)}{dl^4}\right) - 5u\frac{d^2\mu(l)}{dl^2}\right] + 2u\mu(l)\right\}\right|_{l=\bar{l}} \tag{7c}$$

where $\mu(\boldsymbol{l})$ is the phytoplankton growth rate (d$^{-1}$) at mean size $\boldsymbol{l}$. $u$ is the trait diffusion parameter, which describes the probability of the parental size $l(i)$ changing to adjacent size values $l(i-1)$ or $l(i+1)$ in offspring cells (Merico et al., 2014). Eqs. (7a-c) are approximations because the higher-order moments such as the skewness and kurtosis have been ignored and a Gaussian distribution needs to be assumed for $\boldsymbol{l}$. $\frac{dg_i(l)}{dl}$ and $\frac{d^2 g_i(l)}{dl^2}$ are the first and second derivatives of zooplankton clearance rate (d$^{-1}$) against phytoplankton size and will be described in detail in Sect. 2.3.

The equations of $P\bar{l}$ and $P(v + \bar{l}^2)$ follow:

$$\frac{d(P\bar{l})}{dt} = P\frac{d\bar{l}}{dt} + \bar{l}\frac{dP}{dt} + \frac{d}{dz}\left(K_v \frac{d(P\bar{l})}{dz}\right) \tag{7d}$$

$$\frac{d(P(v+\bar{l}^2))}{dt} = P\left(\frac{dv}{dt} + 2\bar{l}\frac{d\bar{l}}{dt}\right) + (v + \bar{l}^2)\frac{dP}{dt} + \frac{d}{dz}\left(K_v \frac{d(P(v+\bar{l}^2))}{dz}\right) \tag{7e}$$

Following previous studies (Flynn, 2003; Geider et al., 1997; Follows et al., 2007; Chen and Laws, 2017), phytoplankton growth rate ($\mu$) depends on temperature ($T$, K), light ($I$, W m$^{-2}$), DIN and $\boldsymbol{fer}$:

$$\mu = \mu_m \min\left(\frac{N}{N+K_N}, \frac{fer}{fer+K_{fer}}\right)\left(1 - e^{-\frac{\alpha_c I}{\mu'_{0,m}e^{\frac{E_p}{k}\left(\frac{1}{T_0}-\frac{1}{T}\right)}}}\right) \tag{8}$$

in which $\mu_m$ is a function of $T$:

$$\mu_m = \mu'_m e^{\frac{E_p}{k}\left(\frac{1}{T_0}-\frac{1}{T}\right)} \tag{9}$$

The trait parameters $\mu'_m$, $K_N$, $K_{fer}$, and $\alpha_c$ are all dependent on cell size $\boldsymbol{l}$:

$$\mu'_m = \mu'_{0,m}e^{\alpha_\mu l + \beta_\mu l^2} \tag{10a}$$

$$K_N = K_{0,N}e^{\alpha_K l} \tag{10b}$$

$$K_{fer} = K_{0,fer}e^{\alpha_{fer}l} \tag{10c}$$

$$\alpha_c = \alpha_{0,c}e^{\alpha_l l} \tag{10d}$$

Eq. (10a) follows that maximal phytoplankton growth rate is a unimodal function of phytoplankton size (Chen et al., 2010, 2011; Marañón et al., 2013). It is worth noting that the light term of phytoplankton growth (the right side of Eq. 8) is usually modelled as $\boldsymbol{1 - e^{-\frac{\alpha_c I}{\mu_m}}}$ (Flynn, 2003), in which both $\alpha_c$ and $\mu_m$

are dependent on size. We use $\alpha_I$ to represent the net effect of size on light-dependent growth for mathematical convenience, which leads to Eq. 8.

Following Flynn (2003), we have derived equations to directly calculate phytoplankton chlorophyll-to-carbon ($\theta$, g Chl (mol C)$^{-1}$) and nitrogen-to-carbon ($Q_N$, mol N (mol C)$^{-1}$) ratios from ambient light and nutrient levels:

$$\theta = \theta_{min} + \frac{\mu}{I\alpha_C}(\theta_{max} - \theta_{min}) \tag{11a}$$

$$Q_N = \frac{Q_{min}}{1-\left(1-\frac{Q_{min}}{Q_{max}}\right)\frac{N}{N+K_N}} \tag{11b}$$

where $\theta_{min}$ and $\theta_{max}$ are minimal and maximal Chl:C ratios, respectively. $Q_{min}$ and $Q_{max}$ are minimal and maximal N:C ratios, respectively. The total Chl $a$ concentrations ($Chl$, $\mu$g L$^{-1}$) and net primary production ($NPP$, $\mu$gC L$^{-1}$ d$^{-1}$) integrated over the whole size spectra can be calculated as:

$$Chl = P\left(\frac{\theta}{Q_N} + \frac{v}{2}\frac{d^2\left(\frac{\theta}{Q_N}\right)}{dl^2}\right)\Bigg|_{l=\bar{l}} \tag{11c}$$

$$NPP = P\left(\frac{\mu}{Q_N} + \frac{v}{2}\frac{d^2\left(\frac{\mu}{Q_N}\right)}{dl^2}\right)\Bigg|_{l=\bar{l}} \tag{11d}$$

To calculate the fractions of Chl within a size range (i.e. <1 $\mu$m, 1–3 $\mu$m, 3–10 $\mu$m and > 10 $\mu$m), we had to discretize the size spectra into 60 even size classes between $\bar{l} - 6\sqrt{v}$ and $\bar{l} + 6\sqrt{v}$ and calculated the $\mu$, $\alpha_c$, $K_N$, $Q_N$, $\theta$, and eventually Chl of each size class following Eq. (11a-c). This is because the distributions of Chl do not follow the lognormal distribution of cell volume and an analytic solution is not yet available for calculating only a fraction of Chl. Fortunately, this approach only adds a minor computational cost because we only need to calculate the size-fractionated Chl once per day when saving model outputs.

## 2.3 Zooplankton size-dependent grazing

Following Smith et al. (2016), the ingestion rate of zooplankton on size class *l* can be formulated as:

$$G(l) = g_{max}Z\frac{\rho(l)P(l)}{P_T+\varepsilon}\frac{(P_T+\varepsilon)^2}{(P_T+\varepsilon)^2+K_P^2} = g_{max}Z\rho(l)P(l)\frac{P_T+\varepsilon}{(P_T+\varepsilon)^2+K_P^2} \tag{12a}$$

where $G(l)$ is the zooplankton ingestion rate ($\mu$mol N L$^{-1}$ d$^{-1}$) on the size class *l*. $\rho(l)$ is the relative
grazing preference on size class *l*. $Z$ is the biomass of either micro- or mesozooplankton. $\varepsilon$ is the food
other than phytoplankton ($\varepsilon = 0$ for microzooplankton and *MIC* for mesozooplankton). $P_T$ (total
palatable phytoplankton food) is formulated as:

$$P_T = \int_{-\infty}^{\infty}\rho(l)P(l)dl \tag{12b}$$

with $P(l)$ is the phytoplankton concentration at size *l*:

$$P(l) = \frac{P}{\sqrt{2\pi v}}e^{-\frac{1}{2}\frac{(l-\bar{l})^2}{v}} \tag{12c}$$

Zooplankton clearance rate ($g$, d$^{-1}$) on size class $l$ can be formulated as:

$$g(l) = g_{max}Z\frac{P_T+\varepsilon}{(P_T+\varepsilon)^2+K_P^2}\rho(l) \tag{12d}$$

For mathematic convenience, we parameterize $\rho(l) = e^{bl+c}$, where $b$ and $c$ are constants. $P_T$ can be
approximated as:

$$P_T \approx P\rho(l)\left(1+\frac{v}{2}b^2\right)$$

And:

$$\frac{dP_T}{dl} = bP_T$$

$$\frac{d^2P_T}{dl^2} = b^2P_T$$

Thus, the first derivative of $g(l)$ can then be derived as:

$$\frac{dg(l)}{dl} = bg(l)\left\{\frac{[K_P^2-(P_T+\varepsilon)^2]P_T}{[(P_T+\varepsilon)^2+K_P^2](P_T+\varepsilon)}+1\right\} \tag{12e}$$

And:

$$\frac{d^2g(l)}{dl^2} = bg(l)\left\{\left\{\frac{[K_P^2-(P_T+\varepsilon)^2]P_T}{[(P_T+\varepsilon)^2+K_P^2](P_T+\varepsilon)}+1\right\}^2 + bP_T\left\{-\frac{4K_P^2P_T}{[(P_T+\varepsilon)^2+K_P^2]^2}+\frac{[K_P^2-(P_T+\varepsilon)^2]}{[(P_T+\varepsilon)^2+K_P^2]}\frac{\varepsilon}{(P_T+\varepsilon)^2}\right\}\right\} \tag{12f}$$

Note that we do not optimize the parameters of $b$ and $c$ because the zooplankton data are insufficient to constrain the parameters (Table 1, 2).

**2.4 One-dimensional (1D) model**

The 1D model focuses on the upper 150 meters of the ocean. The vertical grid, a total of 30 layers,
follows a stretched vertical coordinate with increasing resolution towards the sea surface (surface stretching parameter = 2.0), similar to that used in the Regional Ocean Modelling System (ROMS) (Shchepetkin and McWilliams, 2005). For computational efficiency, instead of explicitly solving the complete moment, temperature, and salinity equations, we imported the physics variables that are directly relevant to the ecological processes from external data products.

10       Four types of external physics forcing data were imported into the 1D model: vertical eddy diffusivity ($K_v$), surface photosynthetic available radiation ($PAR_0$), atmospheric dust deposition, and vertical temperature profiles. Vertical advection of water was neglected, which had been shown relatively unimportant (Fernández-Castro et al., 2016). The most important physics forcing data, $K_v$, determined the upward nutrient flux to the upper euphotic zone and were imported from the output of a
three dimensional (3D) eddy-permitting model targeted for North Pacific (Hashioka et al., 2009). This 3D model was able to faithfully simulate the Kuroshio Current and the spatial distributions of the Chl *a* fields. The extracted vertical profiles of $K_v$ were also consistent with the *in situ* estimated mixed layer depths (MLD) at the three stations (Fig. 2). $PAR_0$ were imported from SeaWIFS satellite monthly climatology products. Seasonal temperature vertical profiles were imported from WOA2013 monthly
climatology.

Light levels ($I_z$) at depth $z$ were calculated based on $PAR_0$ and Chl *a* concentrations following the Beer-Lambert law:

$$I_z = PAR_0 e^{-z\left(K_w + K_{chl}\int_z^0 Chl(x)dx\right)} \tag{13}$$

in which $K_w$ and $K_{chl}$ are the attenuation coefficients for seawater and Chl *a*, respectively. To
realistically estimate the average light field that a phytoplankton cell should experience in a mixing

water column (Franks, 2015), the ambient light level for phytoplankton within the surface mixed layer is calculated as the average light throughout the surface mixed layer, which is defined as the deepest depth with $K_v > 10^{-3}\,m^2\,s^{-1}$. This calculation is based on eq. (1) in Franks (2015), which gives that the average time for a phytoplankton cell to move 100 m (an approximate estimate of MLD) at the local

diffusivity of $10^{-3}\,m^2\,s^{-1}$ is roughly half a day. However, to compare with *in situ NPP* estimates that were calculated from incubation bottles without continuous mixing, phytoplankton $\mu$, $\theta$, and $Q_N$ are recalculated from $I_z$ based on the Beer-Lambert law (Eq. 13).

The initial condition of inorganic nitrogen was set to the vertical profile of nitrate in January of the World Ocean Atlas (WOA) 2013 monthly climatology. Initial phytoplankton, zooplankton, and detritus

biomass were all set to 0.1 $\mu$mol $L^{-1}$ in each grid. Initial phytoplankton mean log size ($\bar{l}$) and log size variance ($v$) were set to be 1. Initial dissolved iron concentration was set to the vertical profile of iron in January from a 3D global biogeochemical model output (Aumont et al., 2003). The time step of the model was 30 minutes. All the fixed model parameters are shown in Table 1 and the model parameters that are optimized to match observational data are shown in Table 2.

We employed a Dirichlet boundary condition at the bottom for DIN and *fer* with the values predefined by the WOA2013 climatology and the model output from Aumont et al. (2003), respectively. For other tracers, we assumed no diffusive flux at the bottom. Detritus was allowed to sink out of the system with the loss of nitrogen and iron replenished by diffusion.

**2.5 Delayed Rejection Adaptive Metropolis-Hasting Monte Carlo (DRAM) algorithm**

The Metropolis-Hasting Monte Carlo (MHMC) algorithm aims to find the posterior distribution (including mean and covariance matrix) of the parameter vectors, given the data provided. The key here is to develop an appropriate proposal covariance matrix ($P_{cvm}$), which determines the magnitude and direction of the proposed perturbations to the parameter values, as the algorithm explores the parameter space. At each iteration of the algorithm the newly proposed parameter set is either accepted or rejected

based on the model-data mismatch, as explained below. In the classical random walk MHMC algorithm, the $P_{cvm}$ must be specified by the user to achieve sufficient acceptance rates for the proposed parameters, and this typically requires a great deal of effort and many trials.

The adaptive MHMC (Haario et al. 2001), uses the already accepted parameters to approximate $P_{cvm}$, which it periodically updates as more simulations are conducted. Specifically, the $P_{cvm}$ is tuned based on the covariance matrix ($C_{vm}$) of the already accepted parameter sets after a fixed number of iterations following Gelman et al. (2014) (i.e. $P_{cvm} = C_{vm} \cdot 2.4^2/d$, where $d$ is the length of the target

parameter vector). Thus, the algorithm alters the magnitude and direction of proposed 'jumps' in order to efficiently explore the parameter space.

With the delayed rejection MCMC (Mira, 2001), when a newly proposed set of parameters is rejected, $P_{cvm}$ is temporarily downscaled (to 1% of the original $P_{cvm}$ in our case) and a second set of parameters is proposed based on the rejected parameters and the downscaled $P_{cvm}$. This approach is

particularly efficient because low acceptance rates typically result when the $P_{cvm}$ is too large (i.e., the parameter jumps are too wide) to find the target distribution of the parameters. Temporarily reducing $P_{cvm}$, can substantially increase the acceptance rate. By using multiple stages of $P_{cvm}$, the algorithm can also effectively deal with the problem of non-Gaussian posteriors, which can reduce the efficiency of the adaptive MHMC (Haario et al., 2006).

The DRAM algorithm, built upon the classic Metropolis-Hasting Monte Carlo (MHMC) algorithm, incorporates the merits of both adaptive and delayed-rejection MHMC algorithm to increase the acceptance rate and thus more efficiently find the target distribution of parameter values (Haario et al., 2006; Laine, 2008). It has been shown to better explore the parameter space compared to other algorithms such as the families of Simulated Annealing, possibly because of its two-stage proposal

covariance matrices (Villagran et al., 2008). Compared with the widely used ensemble Kalman filter, DRAM is more suitable for the nonlinear response typically of ecosystems (Annan and Hargreaves, 2007).

Here we briefly outline the DRAM algorithm. For further details and proofs see Haario et al. (2006) and Laine (2008).

1)   Initialize the parameter values and $P_{cvm}$, assuming no correlation among parameters, and a standard deviation equal to one sixth the difference between the maximal and minimal value for each parameter, respectively (Table 1).

2) Run the model with the current parameter values ($\theta_{curr}$) and calculate the likelihood ($L$). Note that all the parameter values must be within the boundaries shown in Table 2.

3) Propose a new set of parameters ($\theta_{pro}$) based on $\theta_{curr}$ and $P_{cvm}$, rerun the model, and obtain a new likelihood ($L_1$).

4) If the ratio of $L_1/L$ is larger than a random number between 0 and 1, then accept $\theta_{pro}$ ($\theta_{curr} = \theta_{pro}$) and return to step 2).

5) Otherwise, propose a second set of parameters ($\theta_{pro2}$) based on $\theta_{pro}$ and $P_{cvm2}$ ($= 0.01 P_{cvm}$), rerun the model, and obtain the second likelihood ($L_2$).

6) If the ratio of $\dfrac{L_2}{L} \dfrac{q_1(\theta_{pro2},\theta_{pro1})q_2(\theta_{pro2},\theta_{pro1},\theta_{curr})}{q_1(\theta_{curr},\theta_{pro1})q_2(\theta_{curr},\theta_{pro1},\theta_{pro2})} \dfrac{1-min\left(1,\frac{L_1}{L_2}\right)}{1-min\left(1,\frac{L_1}{L}\right)}$ is larger than a random number between

0 and 1, then accept $\theta_{pro2}$ ($\theta_{curr} = \theta_{pro2}$) and return to step 2). Otherwise retain the current position, $\theta_{curr}$. Here $q_1(y,x)$ is the probability of proposing $y$ given $x$ and $q_2(z,y,x)$ is the probability of proposing $z$ given $x$ and $y$.

7) After a certain interval, update $P_{cvm}$ based on $C_{vm}$ calculated from the accepted $\theta$.

To increase the computational efficiency and avoid being trapped in local minima because of
insufficient chain length, we modified the DRAM algorithm for parallel computing (Calderhead, 2014). That is, we initialize $\theta$ and $P_{cvm}$ simultaneously for $n$ processes. Each process runs the above procedure from 1) to 7) except that at 7) all accepted $\theta$ are consolidated to update the global estimate of $P_{cvm}$, which is then distributed to all sub-processes to propose new $\theta$.

Preliminary model runs suggested that from the third year, the model reached a quasi-steady state,
exhibiting regular seasonal cycles under the climatological forcing (Fig. 3). As such, we ran the model for four years and the output of the final year was used for validation against observational data. The model outputs were linearly interpolated to the observational depths and time. To allow fair comparisons among different data types and downplay the effects of extreme values, both the model outputs and observational data were transformed to their 1/4 power and normalized between 0 and 1 to
achieve a quasi-normal distribution before calculating sum of squared errors ($SSqE$):

$$SSqE_{k,i} = \sum_{j=1}^{n_{k,i}} \left( \frac{m_{k,i,j}^{0.25}-o_{k,i,min}^{0.25}}{o_{k,i,max}^{0.25}-o_{k,i,min}^{0.25}} - \frac{o_{k,i,j}^{0.25}-o_{k,i,min}^{0.25}}{o_{k,i,max}^{0.25}-o_{k,i,min}^{0.25}} \right)^2 \qquad (14a)$$

where $SSqE_{k,i}$ is the sum of squared errors of data type $i$ at station $k$. $n_{k,i}$ is the number of observations for data type $i$ at station $k$. $o_{k,i,j}$ is the observed $j^{th}$ value for data type $i$ at station $k$. $o_{k,i,min}$ and $o_{k,i,max}$ are minimal and maximal observed values for data type $i$ at station $k$, respectively (Note that for all size-fractions of Chl $a$, we intentionally set $o_{k,i,min} = 0$ and $o_{k,i,max} = 1$ to minimize the effects of the large measurement variability). $m_{k,i,j}$ is the value linearly interpolated from model outputs to the same depth and date of $o_{k,i,j}$.

Following Laine (2008), the likelihood function was calculated as the product of the exponential of the sum of squared errors, scaled by a measure of the model-data error for each data type, respectively:

$$L = \prod_{k=1}^{2} \left[ \prod_{i=1}^{9} (2\pi)^{-\frac{n_{k,i}}{2}} \sigma_{k,i}^{-n_{k,i}} e^{-\frac{SSqE_{k,i}}{2\sigma_{k,i}^2}} \right] \qquad (14b)$$

in which $\sigma_{k,i}$ is the standard deviation of the Gaussian errors of data type $i$ at station $k$.

Following Laine (2008), we applied Gibbs sampling, which estimates the distribution of each $\sigma_k$, so as to match the ensemble distribution of model output to that of the data. This entails assuming that the prior of $1/\sigma_{k,i}$ follows a gamma distribution, with the prior mean as $S_0^2$ and prior accuracy as $n_0$. At each step the value of $1/\sigma_{k,i}$ was sampled from a conditional gamma distribution $\Gamma\left(\frac{n_0+n_{k,i}}{2}, \frac{n_0 S_0^2 + SSqE_{k,i}}{2}\right)$. The model parameters were assumed to follow multivariate normal distributions. The likelihood function contributed by the priors of the parameters was:

$$L_{pri} = (2\pi)^{-\frac{n_p}{2}} \left(\prod_{i=1}^{n_p} \eta_i^{-1}\right) e^{-\sum_{i=1}^{n_p}\left(\frac{\theta_i - \gamma_i}{\eta_i}\right)^2} \qquad (14c)$$

in which $n_p$ is the number of parameters to be estimated, $\gamma_i$ and $\eta_i$ are the prior estimates of the $i^{th}$ parameter and its standard deviation, respectively (Table 2). Values of $\eta_i$ were calculated as one-sixth of the difference between the preset maximal and minimal parameter boundaries. $\theta_i$ is the current parameter value. The MCMC chain was run for an ensemble of 10000 simulations with five processes running in parallel (i.e. a total of 50000 parameter sets were obtained). Although the model contains more than 20 parameters, we only selected 9 parameters for optimization, to minimize the possibility of parameter unidentifiability and avoid optimising highly correlated parameters (e.g. $g_{max}$ and $K_p$) simultaneously (Table 2). The data used are described below.

2.6 **Observational data**

For stations K2 and S1, the observations including MLD and nine types of data (DIN, CHL, NPP, PON, Fer, and four size fractions of CHL) were obtained from the K2S1 project (https://ebcrpa.jamstec.go.jp/k2s1/en/index.html; Honda, 2016; Table 3). The observations spanned from 2010 to 2013 at seasonal sampling frequencies. Part of the data have been published in Wakita et al. (2016), Fujiki et al. (2016), Matsumoto et al. (2016), and Sasai et al. (2016). DIN was calculated as the sum of nitrate, nitrite, and ammonia, which were measured with a continuous flow analyzer (QuAAtro 2-HR system, BL-Tech). CHL was measured using the nonacidification method following Welschmeyer (1994). NPP was measured with the technique of $NaH^{13}CO_3$ uptake (Matsumoto et al., 2016). PON was measured by an elemental analyser (Wakita et al., 2016). Size fractions of CHL were measured by filtering seawater sequentially through 10 μm, 3 μm, 1 μm polycarbonate membrane filters and finally a GF/F glass-fibre filter. The filters were soaked in N,N-dimethylformamide (DMF) and chlorophyll concentrations retained on the filters were measured with the same protocol as total CHL (Fujiki et al., 2016).

For station ALOHA, the data were downloaded from http://hahana.soest.hawaii.edu/hot/. All the data were pooled together to generate a quasi-climatological seasonal pattern and inter-annual variations were treated as random noise. To improve data coverage, we also included the nitrate data of World Ocean Atlas (WOA) 2013 for observed DIN. Due to the lack of *in situ* observational data, the data of *fer* were obtained from a global biogeochemistry model (Aumont et al., 2003). To calculate MLD from depth profiles of temperature and salinity, MLD was defined as the first depth that the seawater density exceeds surface density by 0.125 kg m$^{-3}$ (Shigemitsu et al., 2012).

## 3 Results

### 3.1 External physics forcing

The validity of external physics forcing data, particularly vertical mixing that determines upward nutrient diffusive supply to the surface mixed layer, is essential for correct results and parameter optimization with the ecosystem model. Here we show in Fig. 2 a representative year of seasonal variations of $K_v$, temperature, surface PAR, and atmospheric iron deposition. Vigorous winter mixing precedes summer water column stratification at K2 and S1, while the seasonal variations of mixing are

less pronounced at ALOHA. At all three stations, the model estimates of mixed layer depths are consistent with those measured from *in situ* temperature and salinity profiles (Fig. 2b,f,j). Water temperatures and surface PAR values at the subarctic station K2 are significantly lower than at the subtropical stations S1 and ALOHA. The station K2 is also characterized by a pronounced spring peak

of atmospheric dust deposition.

## 3.2 Parameter optimization and sensitivity analysis

For all the five parallel sub-processes, the log-likelihood continued to increase with the number of model runs and reached a plateau after 1000 iterations (Fig. 4). For most (but not all) types of data, model–data mismatches (*SSqE*) consistently decreased. Comparing the two stations, the model fits to

the Chl, and NPP were better at station K2 than S1. The model fits to the size fractions of $1 \sim 3$ $\mu$m were better at S1 than K2.

Most values of the optimized parameters fell into reasonable ranges (Table 2; Fig. 5). For example, the estimated $K_{0,N}$ is close to the value (0.2 $\mu$M) given in Ward et al. (2012). For some of the parameters such as $W_d$ and $u$, the final optimized value differed substantially from initial estimates, an expected

outcome of the algorithm striving to match with the nine different types of observations at both stations with contrasting environments. Below we show some preliminary results of sensitivity analysis particularly on those differing with *a prior* estimates (Table 4).

The mean $\mu'_{0,m}$ estimated from laboratory phytoplankton data is around 0.4 d$^{-1}$, half of the optimized value (Chen and Laws, 2017). Reducing $\mu'_{0,m}$ to 0.4 d$^{-1}$ mainly generated worse fits to the

size fractions of <1 $\mu$m fractions of CHL at both stations. This is because the lower phytoplankton growth led to higher nutrient concentrations and lower estimates of <1 $\mu$m fractions.

The estimate of $W_d$ of 20 m d$^{-1}$ is a relatively high sinking speed. Reducing $W_d$ to 10 m d$^{-1}$ only led to slightly worse fits to DIN data at station S1 (but better fits to DIN at K2) and overall did not deteriorate the results substantially.

The estimate of $m_z$ (0.2 ($\mu$M N)$^{-1}$ d$^{-1}$) is also at the high end of those used in the literature. We found that the model results were quite sensitive to the value of the closure term $m_z$. Reducing $m_z$ to 0.1 ($\mu$M N)$^{-1}$ d$^{-1}$ led to higher mesoplankton biomass and generated much worse fits particularly for DIN at K2.

We also tested whether we could assume that the light component of phytoplankton growth is size independent (i.e. $\alpha_I = 0$). The results suggested that with $\alpha_I = 0$, the model predicted much worse fits to the data. An optimized value of $-0.26$ for $\alpha_I$ is also consistent with the size scaling relationship of light dependent growth in Finkel (2001) and Edwards et al. (2015), suggesting that light limitation could drive phytoplankton being small.

The optimized $u$ value was much higher than in Acevedo-Trejos et al. (2016). Reducing $u$ to 0.05 led to worse fits to the size-fractionated chlorophyll since lower size variance failed to capture the observed size scatter. It also relates to the limitation of the model that has to assume a lognormal distribution of size (*see* Sect. 4.2.1). However, an abnormally high $u$ could drive the model to unstable conditions in which the size variance kept increasing.

### 3.3 Comparison between best model outputs and observation

The best model outputs in terms of the highest likelihood could capture most of the observational patterns quantitatively (Figs. 6–9). At both stations, the model could reproduce the vertical increasing trend of DIN with depth and the higher surface concentrations of DIN during winter than summer and autumn. It is noteworthy that the model could also successfully reproduce the relatively abundant summer DIN concentrations at surface at station K2 due to the incorporation of iron and light limitation. The vertical and seasonal patterns of Chl *a* and NPP could also be well reproduced at station K2. The only problem is that, at station S1, the high NPP at surface could not be well reproduced (Fig. 7).

Validation against observed phytoplankton size data is critical for testing **CITRATE** 1.0 in which phytoplankton size structure is the core component. The model could reproduce most patterns of the proportions of size-fractionated Chl at both stations (Figs. 8, 9). For example, the model correctly reproduced the relative dominance of picophytoplankton (<3 $\mu$m) at both stations, although nitrate concentration was high at station K2. The seasonal and vertical fractions of 3–10 $\mu$m were generally well simulated at both stations, except for an artificial surface peak at K2. The model could also simulate the relative larger sizes at K2 than at S1.

We also note some deficiencies of the model. At both stations, the fractions of > 10 $\mu$m Chl were close to zero at both stations in the model, in contrast to the substantial fractions of > 10 $\mu$m during summer at K2 and in the winter at S1. The model also tended to overestimate the 1–3 $\mu$m fractions at

both stations and underestimate the <1 $\mu$m fractions occasionally. All these problems relate to the assumption of a fixed trait distribution as discussed later.

**3.4 Modelled seasonal patterns of nutrients, phytoplankton biomass, mean size, and size diversity**

At both stations, DIN concentrations were higher during winter in the surface mixed layer due to more vigorous mixing (Figs. 10, 11). Significant drawdown of DIN occurred in surface water following water column stratification which occurs earlier in S1 than K2. At station K2, after an increase during June and July due to the peak of atmospheric deposition, dissolved iron concentration also decreased in the fall due to phytoplankton uptake. By contrast, surface iron concentrations accumulated from late summer to fall due to nitrogen limitation at station S1.

In accordance with the DIN patterns, higher concentrations of Chl $a$ were found during winter at station S1, which results from both increased phytoplankton biomass and chlorophyll-to-carbon ratios (Fig. 11). Starting from spring to fall, subsurface maximal layers of Chl $a$ formed and progressively deepened with time. By contrast, at station K2, Chl $a$ concentrations peaked in summer and subsurface chlorophyll maximum layers were not evident (Fig. 10), suggesting light limitation played a stronger role in limiting phytoplankton growth at K2 than S1.

At both stations, in spite of the nutrient increases in winter, phytoplankton mean size peaked in spring or summer. This is mostly attributed by the light limitation on large cells, which can be reflected by the negative value of $\alpha_I$ (Table 2). At both stations, the main periods of size increases were in spring when light level increased and there were still nutrients left from winter mixing. The increases in light were contributed by both increases in surface PAR and shallower mixing. Nutrient (dissolved iron in the case of K2) depletion together with light decreases led to negative values of $\frac{d\mu(l)}{dl}$ since late spring or summer at both stations, resulting in subsequent decreases in mean size. In general, the modelled mean sizes were significantly larger at station K2 than S1, mainly due to less severe nutrient limitation.

The modelled patterns of size variances (i.e. size diversity) are the focus of **CITRATE**. Within the surface mixed layer, modelled phytoplankton size diversity showed an opposite pattern with mean size, with the peaks in fall at S1 and in winter at K2 (Fig. 10,11). At first glance, we also seemed to find a negative correlation between the growth rate $\mu_{com}$ and size diversity at both stations (Fig. 12a). When growth rates were high, size variances were low, and vice versa. The paired scatterplots between $\mu_{com}$

and size variances in surface waters suggested that these two quantities were not linearly correlated, particularly at S1. Instead, their relationships depended on the timing of the season. At station S1, during the transition from the end of winter to early spring, phytoplankton cells experience a rapid increase in growth rate without much change in size diversity. During the rest of spring, phytoplankton growth rate decreased from the maximum to nearly the minimum; while size diversity first underwent a phase of moderate decrease and then recovered. From the beginning of summer to mid-fall, there were no big changes in growth rate, but size diversity increased dramatically. From mid-fall to the beginning of the winter, phytoplankton growth rate increased, but size diversity decreased to winter values. At station K2, the variability of size diversity was smaller, with high growth rates and low size diversity in summer and the opposite patterns in winter.

We decomposed the different factors in affecting the dynamics of size diversity in surface waters at both stations (Eq. 7c,e; Fig. 12b,c). Three points need to be mentioned. First, the calculated net combined effects, including the second derivatives of growth and grazing ($\frac{d^2\mu(l)}{dl^2}$ and $\frac{d^2g(l)}{dl^2}$), trait diffusion ($\frac{d^4\mu(l)}{dl^4}$ and $\mu(l)$), and vertical mixing (i.e. diffusion), were consistent with the net changes of size variances (some minor differences were because we saved the above quantities at daily interval which could not account for the changes within one day), validating our computation. Second, the contributions from the second derivatives of growth and trait diffusion (dominated by $2u\mu(l)$ with the contributions from $\frac{d^4\mu(l)}{dl^4}$ being minor; Eq. 7c) were the two largest terms, which usually offset against each other. The values of $\frac{d^2\mu(l)}{dl^2}$ were always negative in all times at both stations, suggesting that without "trait diffusion", size variance would decrease toward zero (Eq. 7c). This highlights the importance of trait diffusion (which can be interpreted as genetic mutation or transgenerational phenotypic plasticity) to sustain diversity. The values of $\frac{d^2\mu(l)}{dl^2}$ were more negative when growth rates were higher and it is the margin of these two terms that (partially) drove the changes of size variance. For example, in early April of S1, the decrease of size variance was induced by a more negative $\frac{d^2\mu(l)}{dl^2}$ (*see* also Fig. 11h). Similar situations also occurred at the end of December. Third, water column mixing played a significant role in affecting size diversity, which was the main factor leading to the

peak of size diversity in fall in surface waters at S1. The effect of mixing became important because at this time, a subsurface maximum of phytoplankton biomass still existed below the surface mixed layer. With the deepening of surface mixed layer, substantial biomass of phytoplankton was entrained into surface waters and these phytoplankton communities had different trait properties with surface ones, thereby enhancing diversity (*see* Sect. 4.1.1 for discussion).

The model also generated reasonable patterns of Chl:C and N:C ratios, which were largely determined by light and nutrient concentrations (Fig. 10i,j; Fig. 11i,j). Both Chl:C and N:C ratios were high in winter when nutrient concentrations were high and light levels were low due to strong mixing. And both ratios were low in surface stratified waters where nutrient supply from below became diminished due to strong stratification and also light levels became strong due to both increased surface PAR and shallow mixing layers.

## 3.5 Validations of the model at station ALOHA

We used the optimal parameter sets obtained at stations S1 and K2 to run the model at station ALOHA. As there were no data of size-fractionated Chl at ALOHA, we only compared the model outputs of DIN, CHL, NPP, and PON with the observational data. While the modelled profiles of DIN matched well with the observed data, the model underestimated CHL, NPP, and PON, although the qualitative patterns could be reproduced (Fig. 13).

## 4 Discussion

### 4.1 Model merits

#### 4.1.1 Facilitating understanding on ecological mechanisms

Besides the improved computational efficiency (Acevedo-Trejos et al., 2016), the most important advantage of the continuous trait-based 'adaptive dynamics' approach is expressed well in the following quote from Bak (1996): "If, following traditional scientific methods, we concentrate on an accurate description of the details, we lose perspective." (p. 10) and, "It is a futile endeavour to try to explain most natural phenomena in detail by starting from particle physics and following the trajectories of all particles." (p. 5). This modelling approach has the potential to make it much easier to understand the mechanisms regulating phytoplankton diversity, because the functional trait diversity itself (quantified

by the trait variance) is a tracer in the model, and the sources and sinks of diversity are given explicitly (Eq. 7). In particular, the second derivative of the growth rate, $\frac{d^2\mu(l)}{dl^2}$, evaluated at the mean size, is a proxy for the intensity of resource competition. The more concave is the curve of $\mu(l)$, the more intense is the competition, i.e., the fitness of suboptimal species decreases more steeply with distance from the

optimal size. In typical NPZD-type models, phytoplankton species compete for the same nutrients, but this competition is not quantified explicitly by any equation or parameter (as it is in the idealized Lotka–Volterra equations) . This makes it more difficult to quantify the dynamics of competition using typical approaches that model the trajectories of many species. Using 'adaptive dynamics', it is easier to quantify competition intensity (and other ecological quantities), which makes it easier to test ecological

theories such as Huston's "general hypothesis of species diversity" (Huston, 1979). For example, the absolute magnitude of $\frac{d^2\mu(l)}{dl^2}$ correlates positively with $\mu$ (Fig. 13), indicating that higher growth rates induced greater resource competition. This agrees well with the "dynamic equilibrium theory". Huston (1979) emphasized that in natural environments where equilibrium is rarely achieved, growth rates play a greater role in determining diversity than do steady state competitive abilities as typically quantified

by R* values (Tilman, 1982; Litchman et al., 2007). This is because when environmental conditions favour fast growth, it takes less time for the dominant species to predominate, and diversity decreases. The positive correlation between the absolute value of $\frac{d^2\mu(l)}{dl^2}$ and $\mu$ is a mathematical manifestation of the verbal argument in Huston (1979).

Similarly, Eq. 7b specifies the sources and sinks of mean size, making it easy to understanding the

factors affecting phytoplankton size. In fact, Eqs. 7a-c can be understood as derived from a Taylor expansion representing an infinite number of discrete trait classes. Hence, even if a discrete version of a diversity model is used, it may be also helpful to calculate the terms in Eqs. 7a-c in order to understand the factors affecting species diversity, biomass, and productivity.

The set of Eqs. 7 also provides an excellent platform to investigate the underlying mechanisms for

the relationship between biodiversity and ecosystem functioning (productivity, in this case), which have been extensively studied (Loreau et al., 2001; Tilman et al., 2014). While the negative correlation between productivity ($\mu_{com}$) and diversity suggests that enhanced productivity can induce greater

competition and reduce diversity (Huston, 1979), diversity can certainly be affected by other factors besides competition.

The incorporation of trait diffusion originally developed for continuous trait-based models (Merico et al., 2014) provides a means of representing mutation and other processes that sustain diversity, thus linking ecological and evolutionary processes (Rosenzweig, 1995) and allows control of the level of diversity in simulation experiments such as those conducted herein to investigate diversity-productivity relationships. The increasing effect of trait diffusion with growth rate is consistent with the Metabolic Theory of Ecology, in that metabolic rates, closely coupled with growth rates and generation time, are expected to correlate with mutation rates and affect speciation (Allen et al., 2006; Dowle et al., 2013). Our results have shown that it can be the largest term to balance competitive exclusion (Fig. 13). Without considering this mechanism, diversity could be underestimated in productive waters due to strong competition. This could also contribute to the latitudinal diversity gradient because in warm, tropical regions (ectothermic) organisms tend to growth fast (i.e. short generation time) and therefore have high mutation and speciation rates (Rohde, 1992; Allen et al., 2006; Dowle et al., 2013).

The approach of transporting trait moments across spatial grids, originally developed by Bruggeman (2009), also allows water mixing to affect diversity patterns. Although this approach is not perfect (*see* Sect. 4.2.2 and Fig. 14), it does allow that the mixing of two communities with different mean traits can generate a population with trait variance greater than the weighed mean variance of the two original populations. The larger difference of the mean traits, the greater the increase in trait variance upon mixing. Consider the case of mixing two communities with biomass $P_1$ and $P_2$, mean size $l_1$ and $l_2$, size variance $v_1$ and $v_2$. The biomass and mean size of mixed community are $P_1 + P_2$ and $\frac{P_1 l_1 + P_2 l_2}{P_1 + P_2}$, respectively. After some algebraic manipulation, we can derive the size variance ($v'$) after mixing:

$$v' = \frac{P_1(l_1^2 + v_1) + P_2(l_2^2 + v_2)}{P_1 + P_2} - \left(\frac{P_1 l_1 + P_2 l_2}{P_1 + P_2}\right)^2 = \frac{P_1 P_2 (l_1 - l_2)^2}{(P_1 + P_2)^2} + \frac{P_1 v_1 + P_2 v_2}{P_1 + P_2} \tag{15}$$

Thus, it is clear from Eq. (15) that the difference between $v'$ and biomass weighed mean variance $\frac{P_1 v_1 + P_2 v_2}{P_1 + P_2}$ depends on the difference of mean traits. Hence, mixing can enhance diversity to the extent that the traits of the original communities differ. Barton et al. (2010) have shown that the "hotspots" of

high phytoplankton diversity were usually located along areas where mixing was strong enough to allow coexistence of multiple populations with different traits. Our simulations are consistent with that view and show that vertical mixing can significantly enhance diversity, specifically during fall in the surface waters at S1.

### 4.1.2 Flexible stoichiometry

We also consider realistic phytoplankton physiology and optimized model parameters guided by real data. For example, our model has incorporated some features of phytoplankton plasticity (acclimation) such as variable Chl:C ratio and N:C ratios. Although, for the sake of simplicity, these variable ratios do not directly influence phytoplankton specific growth rate as in Geider et al. (1997), they are able to reproduce the high Chl:C ratios in the DCM layer, thus providing a more realistic mechanism for the formation of the DCM layer than with models that assume fixed ratios (Fennel and Boss, 2003). Similarly, the variable N:C ratio also allows phytoplankton cells to achieve higher carbon-based NPP in surface waters compared to models with fixed N:C ratios (Christian, 2005). Although cellular chlorophyll and nitrogen quota are not calculated as independent tracers, model comparisons suggest that more complex models do not yield better fits to the data (Flynn, 2003).

### 4.1.3 Realistic mechanisms for controlling phytoplankton size structure

We have provided both bottom-up and top-down mechanisms to affect the size structure of phytoplankton in **CITRATE 1.0**. First, we employ an observation-based unimodal relationship between maximal growth rate and size to give the nanophytoplankton the advantage under nutrient-replete conditions (Chen and Liu, 2010, 2011; Marañón et al., 2013), thus allowing a trade-off between nutrient affinity and maximal growth rate within the pico- and nano-size range. Thus, bottom-up factors alone are sufficient to reproduce the observed decrease in the fraction of small phytoplankton with nutrient enrichment (Marañón et al., 2012). We also impose a size-dependent feeding preference of zooplankton based on the general understanding that smaller microzooplankton tend to prefer smaller phytoplankton, whereas larger mesozooplankton tend to prefer larger phytoplankton (Frost, 1972; Hansen and Hansen, 1994; Liu et al., 2005; Ward et al., 2012). These top-down factors have additional effects on phytoplankton size structure. Our assumption of the preference of microzooplankton on small phytoplankton is similar to Terseleer et al., (2014) and Acevedo-Trejos et al., (2015), who assumed a

combination of decreasing maximal phytoplankton growth rate with increasing size and the grazing preference on small phytoplankton in order to offset the growth advantage of small phytoplankton in eutrophic waters. In our case, small phytoplankton lose the advantage in eutrophic waters, where larger phytoplankton grow faster because of the imposed unimodal relationship between maximal growth rate and size. Meanwhile, in eutrophic waters, mesozooplankton dominate and preferentially feed on larger phytoplankton to balance the growth advantages of larger cells.

Interestingly, counter to our intuition, field incubation experiments have often found that microzooplankton feed on diatoms faster than on picophytoplankton, and that diatoms grow faster than picophytoplankton even in oligotrophic waters (Latasa et al., 1997; Zhou et al., 2015). These results raise a paradoxical question: "How can diatoms grow so fast with negligible nutrients in oligotrophic waters, but without accumulating high biomass?". Whether this is because of experimental bias is an open question. Although the feeding preference of mesozooplankton on large prey seems less disputable (Frost, 1972; Liu et al., 2005) (but *see* Terseleer et al., (2014) for an assumption of decreasing feeding preference of copepods on large diatoms), it implies strong top-down control of large phytoplankton in eutrophic waters where mesozooplankton dominate, limiting the biomass of large phytoplankton. This is at odds with the common observation that large phytoplankton dominate total biomass in eutrophic waters (Marañón et al., 2012). Future refinements might include a unimodal feeding preference, similar to the grazing kernel proposed earlier (Hansen and Hansen, 1994; Poulin and Franks, 2010). In any case, more and better zooplankton data are much needed for model calibration and validation.

## 4.2 Model limitations

### 4.2.1 Assumption of trait distribution

To facilitate calculation of trait moments, a certain distribution has to be assumed for the trait (Merico et al., 2009; 2014). In the literature, phytoplankton abundance (and also biomass) is usually modelled as a power-law function of cell size (Gin et al., 1999; Cavender-Bares et al., 2001; Cermeño et al., 2006). The slope of log abundance versus log size (e.g. cell volume) tends to be between −0.7 and −1 (Cermeño et al., 2006), suggesting that the slope of log biomass versus log size should be between 0 and 0.3. However, aside from fact that the power-law distribution is unrealistic in predicting phytoplankton

biomass at the size limits (there must be upper and lower limits of size at which phytoplankton biomass becomes zero, which the power-law cannot reproduce), the power-law distribution is much more inconvenient for mathematical manipulations (e.g. calculating mean and variance) compared to the normal distribution. The lognormal distribution is a much better alternative for phytoplankton size in terms of mathematic properties (e.g., zero probability of negative size) and can be fit well to empirical data (Quintana et al., 2008). Therefore, it is not surprising that the lognormal distribution has been widely used in continuous size models (Terseleer et al., 2014; Acevedo-Trejos et al., 2015, 2016; Smith et al., 2016).

However, this does not guarantee that a fixed probability distribution can hold for all situations (Coutinho et al., 2016). In oligotrophic waters where picophytoplankton, particularly the unicellular cyanobacteria *Prochlorococcus* and *Synechococcus*, dominate (Campbell et al., 1994; Liu et al., 1997), the distribution of phytoplankton log size is more likely right skewed. In other words, abundances of large species are higher than expected from a pure lognormal distribution, which is consistent with the observation that some large diatoms, with significant contributions to new production, can be found in the oligotrophic gyres (Villareal et al., 1999). This is probably one major reason that our model tends to underestimate the fraction of $> 10~\mu$m size. This is an inevitable consequence of aggregating the description of the entire community into only the three descriptors (i.e. total biomass, mean and variance), which reduces the degrees of freedom, thus sacrificing detailed accuracy for generality and perspective.

One remedy for this problem might be to assign more functional groups in phytoplankton and assume a lognormal distribution for each group, respectively (Terseleer et al., 2014). Having a number of functional groups also circumvents the problem of size-independent functional differences among phytoplankton, such as the different maximal growth rates of diatoms and dinoflagellates despite their similar sizes (Chen and Laws, 2017). We expect that in the near future such a combination of continuous trait distributions and functional groups will likely provide more realistic representations of marine phytoplankton diversity.

**4.2.2 Transport of moments**

Another potential problem is the transport of trait moments in ocean circulation models. Unlike nutrients or plankton biomass, trait moments are not real "concentrations" that can be directly involved in advection and diffusion. The immediate summation of two Gaussian curves with different areas (representing the total biomass), mean, and variance is usually not another perfect Gaussian curve (Fig. 14a). Bruggeman (2009) has derived that, if following the assumption of normal distribution of traits, the raw moments of the biomass distributional can behave as normal tracers in GCMs. We have shown a few examples of mixing of communities of different biomass, mean size, and size variance in Fig. 14. These examples demonstrate that when the mean sizes and size variances differ greatly and biomasses are similar, the mixed community may deviate from the assumed normal distribution, making this a poor approximation. For now, we assume that across adjacent grids, phytoplankton communities should be similar enough for this approximation to work reasonably well.

### 4.2.3 Lack of multiple traits

As a first step, we incorporated only size as the master trait that affects all physiological functions of phytoplankton. In reality, many phytoplankton functional traits, such as optimal temperature, $N_2$ fixation, and mixotrophy, are independent of size. For example, the optimal growth temperature of phytoplankton is closely related to environmental temperature, but only weakly relates to size (Thomas et al., 2012; Chen, 2015). The optimal growth temperature and irradiances are certainly function traits that deserve to be incorporated into trait-based models (Follows et al., 2007; Norberg, 2004; Edwards et al., 2015) and are expected to strongly affect phytoplankton functional identity and diversity at large scales.

### 4.2.4 Difficulty in modelling surface peaks of NPP at oligotrophic stations

The near-surface peak of NPP at the oligotrophic stations S1 and ALOHA during summer is not expected if we assume that the source of nutrients comes from below the euphotic zone. Even if variable N:C ratios are used in the model to allow more carbon to be fixed given the same amount of nitrogen near surface waters, surface NPP is still likely to be underestimated even with the presence of $N_2$ fixation because of phosphorus limitation (Christian, 2005). It is possible that other mechanisms such as vertical migration of phytoplankton need to be taken into account (Villareal et al., 1999; Chavez et al., 2012). Therefore, this problem is not only restricted to **CITRATE** 1.0.

#### 4.2.5 Optimized parameters for 3D GCM

One purpose of finding a common parameter set optimized to two stations with contrasting environmental conditions is to use this parameter set for 3D GCMs with the expectation that, since this parameter set can work for the two stations, it should work for other stations as well. However, our validation exercise at station ALOHA reveals that the parameter set optimized for stations K2 and S1 only succeeds in matching the DIN data well, but underestimates CHL, NPP and PON at station ALOHA. This suggests that we might be overlooking some unique but important processes at ALOHA. Alternatively, it is also possible that the uneven sampling at K2 and S1 might bias the parameter optimisation to some extent. Similar difficulties in parameter optimisation have been shown previously (Ward et al., 2010). For optimising parameters for 3D GCMs, a better approach might be to use the "Transport Matrix" technique that has been successfully implemented for some biogeochemistry models (Khatiwala, 2007; Kriest et al., 2017). Nonetheless, our optimized parameters can provide a useful initial estimate for modelling other stations and for use in 3D GCMs.

#### 4.3 Future directions

Considering the above limitations, one future direction is to increase the number of traits in the model to generate more realistic phytoplankton diversity patterns, which requires both an "envelope" function relating the maximal growth rate with the optimal trait value and a relationship between growth rate and trait value for each species (Norberg, 2004). Another refinement as noted above is to model a continuous trait distribution for each functional group, respectively, thus combining the continuous trait-distribution and functional group approaches to better capture deviations of overall trait distributions from normality.

It is relatively easy to couple the one-dimensional **CITRATE** model with 3D global or regional ocean models, thus providing a means to model the large-scale patterns of phytoplankton size and size diversity. Furthermore, it should be possible in the near future to optimize parameters for such a 3D model using the "transport matrix" technique. In particular, by including both trait diffusion and competitive exclusion it may be possible to begin to untangle the relative roles of ecological versus evolutionary processes in shaping global phytoplankton diversity patterns.

# 5. Conclusions

➢ We present a 1D model with continuous size distribution for phytoplankton (**CITRATE**). The dynamics of phytoplankton mean size and size variance are directly linked to environmental factors and moments of the size distribution (Eq. 7), facilitating understanding of the underlying mechanisms controlling phytoplankton size and diversity. **CITRATE** 1.0 also incorporates "trait diffusion" as an eco-evolutionary process to sustain phytoplankton diversity.

➢ We optimized the parameters of **CITRATE** using the DRAM algorithm, which revealed that the model can faithfully reproduce observed seasonal patterns of inorganic nitrogen, Chl *a*, and phytoplankton size structure at two contrasting time-series stations The model structure and associated parameters obtained herein can be useful for 3D regional and global ocean modelling.

➢ The limitations of **CITRATE** include its assumption of a lognormal distribution for phytoplankton size as the sole master trait, which to some extent limits the precision with which it can reproduce large size classes of phytoplankton. These limitations and others may be overcome in future studies by building on **CITRATE** 1.0 to construct more elaborate continuous trait-distribution models capable of reproducing more realistic patterns of phytoplankton diversity.

# 6. Code and data availability

The code and data of **CITRATE** 1.0 are freely available at: https://github.com/BingzhangChen/citrate under the MIT license.

## 6.1 General instruction

Tutorial: The code for CITRATE 1.0 (DOI: 10.5281/zenodo.1034805) is written in Fortran90 with the Intel Fortran compiler used. We have tested the codes on macOS Sierra 10.12.5 (i386 processor) and also a GNU/Linux cluster with x86-64 architechture. The user is supposed to be familiar with the Fortran language and has some basic knowledge of BASH. Some post-processing scripts are also written in the free software R (version 3.3.2). Before compiling the codes and running the model, the user needs to install the mpi (e.g. openmpi) library for parallel computation. Below we give some instructions and explanations of the codes and how to run the model.

1) Go to the directory you want to run the model (we assume that the root directory is under home directory: ~/).

2) To download the codes, type: "git clone https://github.com/BingzhangChen/citrate.git".

3) Type: "cd DRAM/NPZDcont/BOTH_TD" to go to the working directory.

4) Type: "vi run" to change the setting for model run:

Test = 0 means a fast run, usually for a formal model run for a large number of iterations. Test = 1 means running a model for debugging, which is much slower than the fast run. The user can also modify the compiler flags depending on the purpose in the script. The user needs to confirm the directory where the library of mpifort exists.

5) Type "./run", the model will compile and an executable (CITRATE) will be generated.

6) Type "vi Model.nml", which contains two namelists. The namelist &Model contains the options for station names, the type of ecological model, the type of nutrient uptake function (1 only for CITRATE), and the type for grazing function (four different grazing functions including the three Holling type functions and the Ivlev function). The station name determines the right physics files to be

read and the filenames for model output. For now we only allow three possible stations: S1, K2, and HOT. Other station names will generate an error. If the user wants to add more station names, the subroutine Setup_OBSdata within MOD_1D.f90 is the place to be modified. A number of ecological models besides **CITRATE** have been developed. It is beyond the scope of the present study to describe all of them in detail. Just note that the model lists are in the fortran file bio_MOD.f90 and some other

details are in choose_model.f90 and MOD_1D.f90.

The namelist &MCMCrun contains the options for defining the total length of the MCMC chain which is at least 2, the number of the ensemble runs, the number of days for each model run, whether the model should start from previous runs (Readfile = 1) or start a new run (Readfile = 0), and the number of runs in the historical files (enssig and enspar).

7) After defining all the model settings, type "mpirun –np 5 citrate" and then the model will run with 5 parallel processes and some outputs will be shown on the screen. Type "mpirun –np 5 citrate > out" to make the model outputs stored in the "out" file. For each model run, the model saves the current parameters into the "enspar" file and the current values of $\sigma$ and $SSqE$ into the 'enssig' file. In this way,

even if the model crashes, the user can pick up the current parameter position and updated parameter covariance matrix. The model also generates the files of best parameters, best $\sigma$ and *SSqE* files, best model output files that correspond to observational data, and model output files at daily resolution at each grid after an ensemble run.

For each station, four different physics forcing data including vertical profiles of eddy diffusive coefficients and temperatures, surface PAR and atmospheric dust deposition. We already provided the relevant data for stations S1 and K2. The temporal resolution is one day for the vertical eddy diffusivity and one month for three other types of data.

**6.2 Code structure**

All the source files including the makefile are stored in the *src* folder. Here we briefly describe the functions of the most important source files:

➢    Main.f90: The main program for DRAM that calls each subroutine in serial.

➢    MOD_1D.f90: The major module that sets up and runs the 1D model. The module also generates
model output that matches with the observational data.

➢    Interface_MOD.f90: the module that initializes the absolute and normalized parameter vectors, the covariance matrix of the parameters, the prior parameter values, and the upper and lower parameter boundaries.

➢    SUB_MOD.f90: the module that calculates sum of squared errors (SSqE) between model outputs and observational data. This module also contains the I/O subroutines that save the parameters, $\sigma$, and SSqE for each iteration. It also contains the major subroutine MCMC_adapt that determines whether to accept new parameters, updates covariance matrix, proposes new parameter vectors and
calls the subroutine that runs the 1D model with the newly proposed parameters.

➢    choose_model.f90: the subroutine that defines the number and indices of tracers and the model outputs that need to be written into the output file.

➢ NPZD_cont.f90: the major biological subroutine for the CITRATE model.

➢ bio_MOD.f90: the module that declares most of the model names, indices for model input and output variables and parameters.

Table 1. Fixed parameters of the **CITRATE** 1.0 model.

| Symbol | Description | Value | Unit |
|---|---|---|---|
| $K_w$ | Light attenuation coefficient of seawater | 0.04[a] | $m^{-1}$ |
| $K_{chl}$ | Light attenuation coefficient of chlorophyll | 0.025[a] | $(mg\ Chl\ a\ m^{-2})^{-1}$ |
| $E_p$ | Activation energy of phytoplankton rates | 0.41[b] | eV |
| $E_z$ | Activation energy of heterotrophic rates | 0.65[b] | eV |
| $\theta_{min}$ | Minimal chlorophyll-to-carbon ratio | 0.02[c] | $gChl\ molC^{-1}$ |
| $\theta_{max}$ | Maximal chlorophyll-to-carbon ratio | 0.47 | $gChl\ molC^{-1}$ |
| $unass_1$ | Fraction of unassimilated food by microzooplankton | 0.24[d] | dimensionless |
| $unass_2$ | Fraction of unassimilated food by mesozooplankton | 0.31[e] | dimensionless |
| $NGE$ | Net growth efficiency of zooplankton | 0.3[d] | dimensionless |
| $R_{dn}$ | Conversion rate of detritus to inorganic nitrogen | 0.1 | $d^{-1}$ |
| $l_{fe}$ | Total iron ligand concentration | 0.6[f] | nM |
| $K_{scm}$ | Minimal iron scavenging rate | $5 \times 10^{-3}$ [f] | $d^{-1}$ |
| $K_{sc}$ | Particle dependent scavenging rate | 0.03[f] | $(\mu M\ N)^{-1}\ d^{-1}$ |
| $R_{Fe\_N}$ | Plankton iron-to-nitrogen ratio | 0.0265 | $nM{:}\mu M$ |
| $\alpha_\mu$ | First-order size scaling component for $\mu_m$ | 0.2[b] | $(\ln\ \mu m^3)^{-1}$ |
| $\beta_\mu$ | Second-order size scaling component for $\mu_m$ | -0.01[b] | $(\ln\ \mu m^3)^{-2}$ |
| $\alpha_K$ | Size scaling exponent for $K_N$ | 0.27[g] | $(\ln\ \mu m^3)^{-1}$ |
| $\alpha_{fer}$ | Size scaling exponent for $K_{fer}$ | 0.27[g] | $(\ln\ \mu m^3)^{-1}$ |
| $Q_{0N}$ | Phytoplankton minimal N:C ratio | 0.06 | mol N: mol C |
| $g_{max,1}$ | Maximal microzooplankton specific ingestion rate for phytoplankton of 1.24 $\mu$m (1 $\mu m^3$) at 15 ºC | 1.35[h] | $d^{-1}$ |
| $g_{max,2}$ | Maximal mesozooplankton specific ingestion rate for phytoplankton of 10 $\mu$m at 15 ºC | 0.53[h] | $d^{-1}$ |
| $K_{p,2}$ | Grazing half-saturation constant of mesozooplankton | 0.5[h] | $\mu M\ N$ |
| $b_1$ | Size-dependent feeding selectivity of microzooplankton | -0.05 | $(\ln\ \mu m^3)^{-1}$ |
| $b_2$ | Size-dependent feeding selectivity of mesozooplankton | 0.02 | $(\ln\ \mu m^3)^{-1}$ |

[a]Fennel et al., (2006); [b]Chen and Laws, (2017); [c]Flynn, (2003); [d]Buitenhuis et al. (2010); [e]Buitenhuis et al. (2006); [f]Nickelsen et al., (2015); [g]Ward et al., (2012); [h]Chai et al., (2002).

Table 2. Parameters optimized by the DRAM algorithm. The values inside the parentheses of the initial values indicate the "hard" boundaries for the parameters. The numbers inside the parentheses of the optimized values indicate the standard deviation after the first 10000 iterations have been removed.

| Symbol | Description | Initial | Optimized | Unit |
|---|---|---|---|---|
| $W_d$ | Sinking rate of detritus | 4 (1, 20) | 19.6 (1.0) | m d$^{-1}$ |
| $K_p$ | Grazing half-saturation constant of microzooplankton | 0.5 (0.05, 2) | 0.28 (0.01) | $\mu$M N |
| $m_z$ | Coefficient of mortality rate of mesozooplankton | 0.1 (0.05, 0.2) | 0.20 (0.002) | $(\mu$M N$)^{-1}$ d$^{-1}$ |
| $\alpha_{0,c}$ | Initial slope of photosynthesis versus light at 1 $\mu$m$^3$ | 0.055[a] (0.01, 0.1) | 0.05 (0.004) | (W m$^{-2}$)$^{-1}$ d$^{-1}$ |
| $\alpha_I$ | Size scaling exponent for $\alpha_c$ | -0.1 (−0.3, 0.1) | -0.26 (0.01) | (ln $\mu$m$^3$)$^{-1}$ |
| $K_{0,N}$ | Growth half-saturation constant for nitrogen for a phytoplankton cell of 1 $\mu$m$^3$ | 0.2[b] (0.05, 0.5) | 0.29 (0.03) | $\mu$M N |
| $\mu'_{0,m}$ | Phytoplankton maximal growth rate at 1 $\mu$m$^3$ at 15 ºC | 1.2[c] (0.3, 2.7) | 0.85 (0.05) | d$^{-1}$ |
| $K_{0,fer}$ | Growth half-saturation constant for iron of phytoplankton with 1 $\mu$m$^3$ | 0.08[d] (0.02, 0.2) | 0.17 (0.02) | nM Fe |
| $u$ | Trait diffusion parameter | 0.08[e] (0, 0.1) | 0.1 (0.0008) | d$^{-1}$ (ln $\mu$m$^3$)$^{-2}$ |

[a]Fennel et al., (2006); [b]Ward et al., (2012); [c]Flynn et al., (2016); [d]Gregg et al., (2003); [e]Merico et al., (2014).

Table 3. Observational data at stations S1 and K2. N: number of observations. Min and Max are minimal and maximal values used in data normalization (*see* Sect. 2.4 for details). DIN: dissolved inorganic nitrogen ($\mu$mol L$^{-1}$). Chl *a*: total chlorophyll a concentration ($\mu$g L$^{-1}$). NPP: net primary production measured by $^{13}$C uptake ($\mu$gC L$^{-1}$ d$^{-1}$). PON: particulate organic nitrogen ($\mu$mol L$^{-1}$). Fer: dissolved iron concentration (nmol L$^{-1}$). SF Chl: percentages of four size fractionated Chl *a*. Note that the data of Fer were from model outputs of Aumont et al., (2003) instead of real observations.

| Type | K2 | | | S1 | | |
|---|---|---|---|---|---|---|
| | N | Min | Max | N | Min | Max |
| DIN | 974 | 4.1 | 45.7 | 902 | 0 | 11.2 |
| Chl *a* | 470 | 0 | 3.4 | 426 | 0 | 1.0 |
| NPP | 112 | 0.1 | 37.1 | 128 | 0.1 | 34.9 |
| PON | 29 | 0.1 | 2.2 | 32 | 0.1 | 1.0 |
| Fer | 168 | 0.02 | 1.12 | 168 | 0.02 | 0.95 |
| SF Chl | 143 x 4 | 0 | 1.0 | 166 x 4 | 0 | 1.0 |

Table 4. Sum of squared errors between model outputs and observational data for sensitive analysis. The standard run used the optimized parameter values in Table 2. In other runs, only the value of the parameter shown was changed while others were kept constant.

| | Stn | DIN | CHL | NPP | PON | %3–10 μm | %1–3 μm | %<1 μm |
|---|---|---|---|---|---|---|---|---|
| Standard | K2 | 21.5 | 8.1 | 3.0 | 0.92 | 3.6 | 7.3 | 13.3 |
| | S1 | 11.2 | 16.5 | 6.7 | 1.5 | 4.9 | 4.0 | 4.1 |
| $\mu'_{0,m}=0.4$ | K2 | 19.7 | 11.6 | 5.4 | 0.86 | 6.7 | 7.1 | 23.4 |
| | S1 | 13.2 | 12.8 | 6.6 | 1.3 | 4.4 | 4.3 | 16.3 |
| $W_d = 10$ | K2 | 15.3 | 8.1 | 3.0 | 1.1 | 3.6 | 7.3 | 13.4 |
| | S1 | 17.8 | 12.7 | 4.6 | 1.0 | 2.9 | 4.6 | 6.6 |
| $\alpha_I = 0$ | K2 | 167.3 | 35.8 | 11.8 | 2.0 | 10.4 | 8.6 | 36.4 |
| | S1 | 13.0 | 16.1 | 6.1 | 1.8 | 22.0 | 5.4 | 56.4 |
| $m_z = 0.1$ | K2 | 523.8 | 17.6 | 8.0 | 1.9 | 12.9 | 6.5 | 11.3 |
| | S1 | 11.5 | 17.7 | 7.4 | 1.6 | 9.1 | 2.9 | 3.0 |
| $u = 0.05$ | K2 | 22.2 | 8.3 | 3.2 | 0.94 | 4.7 | 8.5 | 15.4 |
| | S1 | 11.1 | 16.9 | 6.8 | 1.6 | 11.0 | 4.2 | 4.0 |

**Author contribution**

B. Chen and S. L. Smith conceived and designed the study. S. L. Smith wrote the initial MCMC code. B. Chen acquired and organized the observational data, did subsequent coding, and wrote the first draft of the manuscript. Both authors contributed to later revision of the manuscript.

**Competing interests**

The authors declare that they have no conflict of interest.

**Acknowledgments**

We sincerely thank S. Vallina and C. Carcia-Comas for useful discussions. The comments from the editor and three anonymous reviewers have substantially improved an earlier draft of the manuscript. This study would not be possible without the cruise data shared by the K2S1 project personnel, particularly M. C. Honda, T. Fujiki, and K. Matsumoto. We also sincerely thank T. Hashioka for sharing the data of vertical eddy diffusivity. This study is supported by CREST (Grant Number JPMJCR12A3; P.I. SLS) funded by the Japan Science and Technology (JST) Agency and a Grants-in-Aid for Scientific Research (KAKENHI) (Grant Number JP16K21701; P.I. BC) funded by Japan Society for the Promotion of Science (JSPS).

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

Figure captions

Fig. 1. Schematic description of the CITRATE model. Thick arrows indicate nitrogen flows and dashed lines indicate the simplified iron cycle. The inset denotes an example of phytoplankton community with a lognormal distribution for cell volume.

Fig. 2. (a) Locations of the three stations, K2, S1 and ALOHA, overlaid on annual Chl $a$ climatology of the North Pacific. (b-e) Seasonal forcing of vertical eddy diffusivity ($K_v$), temperature, surface PAR, and atmospheric dust deposition, respectively, at station S1. The white squares are measured mixed layer depths from $in situ$ temperature and salinity profiles. The thick tan line represents mixed layer depths calculated from a threshold of $10^{-4}\,\mathrm{m^2\,s^{-1}}$. (f-i) The same as (b-e), but for station K2. (j-m) The same as (b-e), but for station ALOHA.

Fig. 3. An example of modelled patterns of total inorganic nitrogen (DIN), Chl $a$ (Chl), mean size, and ln size variance for four years at stations K2 (a-d) and S1 (e-h).

Fig. 4. (a) Time evolution of log-likelihood of the MHMC chain. (b-i) Time evolution of sum of squared errors ($SSqE$) for DIN, Chl, net primary production (NPP), particulate organic nitrogen (PON), and fractions of size-fractionated Chl $a$ concentrations of > 10 $\mu$m (P10), 3–10 $\mu$m (P03), 1–3 $\mu$m (P01), and <1 $\mu$m (P_1). (j-q) The same as (b-j), but for station S1.

Fig. 5. Time evolution of fitted model parameters.

Fig. 6. Model fittings to vertical profiles of (a-d) DIN, (e-h) CHL, (i-l) NPP, and (m-p) PON at four seasons at station K2. Black dots represent observational data and red thick solid lines represent the averaged seasonal values predicted by the model. Thin dashed lines represent 95% percentiles of the seasonal data.

Fig. 7. The same as Fig. 6, but for station S1.

Fig. 8. Model fittings for the percentages of the four size fractions of Chl $a$ at station K2. (a-d) Percentages of > 10 µm fraction. (e-h) 3-10 µm. (i-l) 1-3 µm. (m-p) <1 µm.

Fig. 9. The same as Fig. 8, but for station S1.

Fig. 10. Modelled seasonal patterns at station K2: (a) DIN, (b) dissolved iron, (c) Chl $a$, (d) phytoplankton mean size, (e) size variance, (f) community-based specific growth rate ($\mu_{com}$), (g)

first derivative of phytoplankton growth rate against ln volume evaluated at mean size, (h) second derivative of phytoplankton growth rate evaluated at mean size, (i) Chlorophyll-to-carbon ratios, and (j) Nitrogen-to-carbon ratios.

Fig. 11. The same as Fig. 10, but for station S1.

Fig. 12. (a) Scatterplots of size variance versus phytoplankton community growth rate ($\mu_{com}$). (b) Contributions of various factors to the dynamics of size variance in surface waters at S1. The term "Competition" equates to $v^2 \frac{d^2\mu}{dl^2}$. *MIC* and *MES* grazing equates to $-v^2 \frac{d^2 g_i}{dl^2}$. "d4$\mu$/dL4" equates to $v^2 u \frac{d^4\mu}{dl^4}$. "Trait diffusion" equates to $2u\mu$. All the derivatives are evaluated at the mean size. "Diffusion" means the contribution to the changes of size variance induced by diffusion with the underlying grid. "Net effect" means the sum of the above terms. "Net changes" mean the difference of size variance between adjacent days. (c) The same as (b), but at station K2.

Fig. 13. The same as Fig. 6, but for station ALOHA.

Fig. 14. Schematic diagrams for mixing of two phytoplankton communities with different biomass, mean size, and size variance, each following a lognormal size distribution.

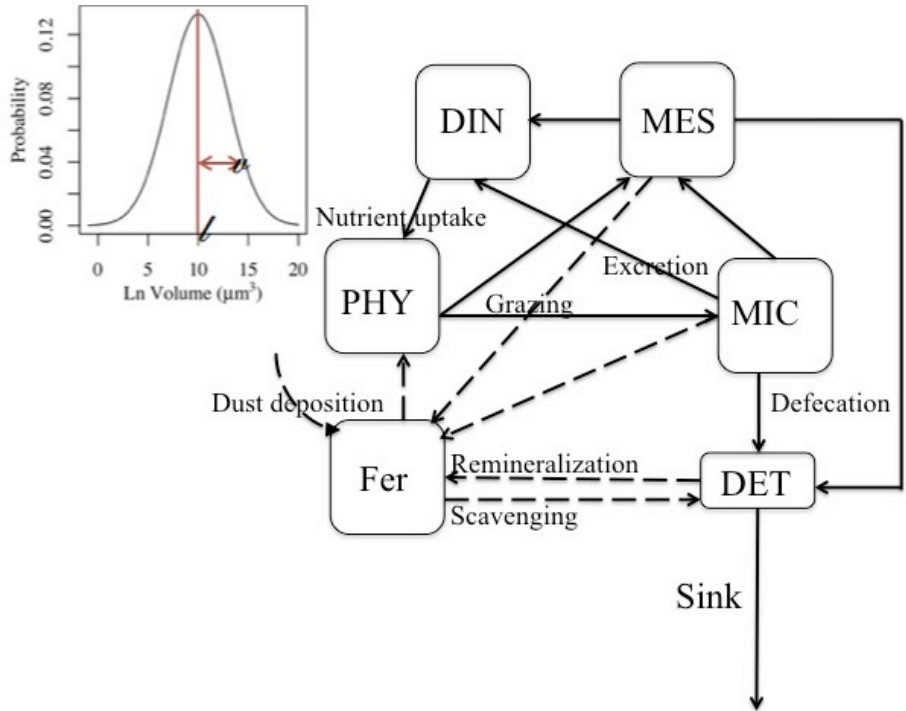

Fig. 1

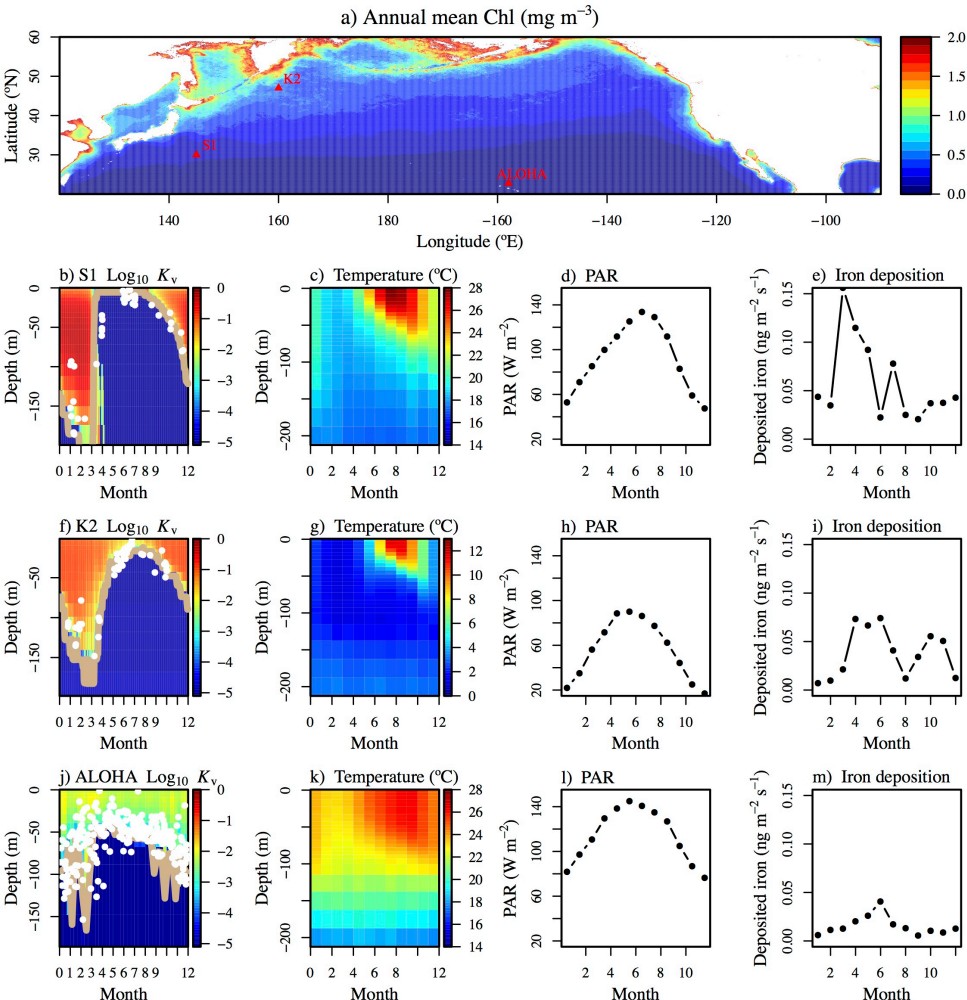

Fig. 2

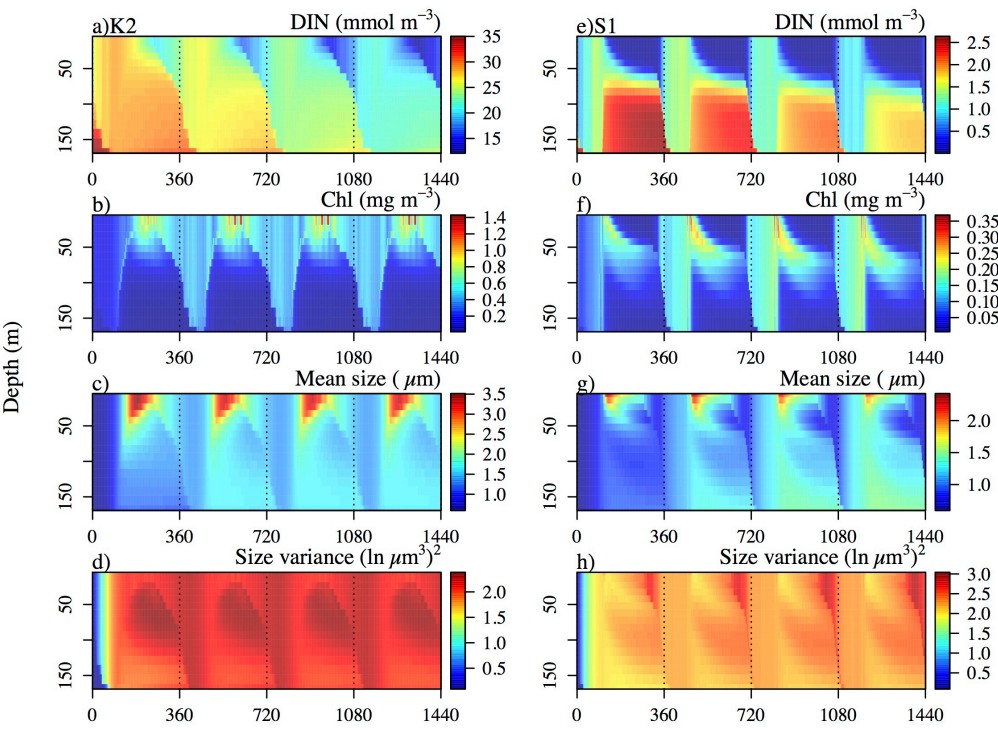

Fig. 3 An example of modelled 4 year patterns at K2 and S1

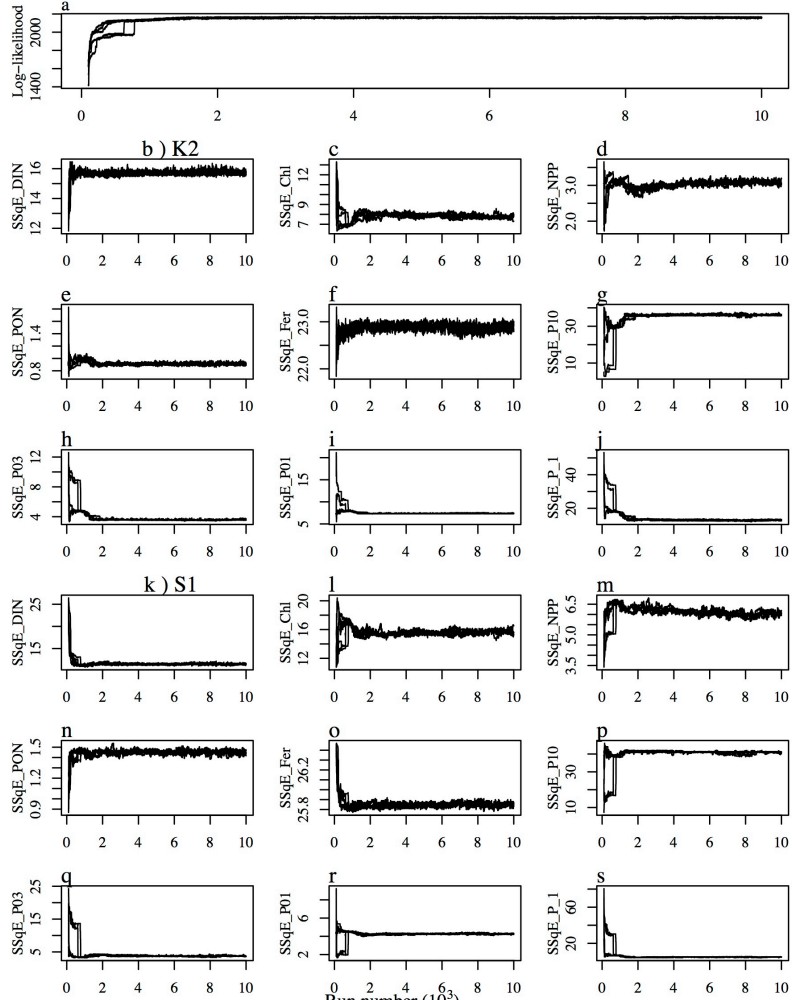

Fig. 4

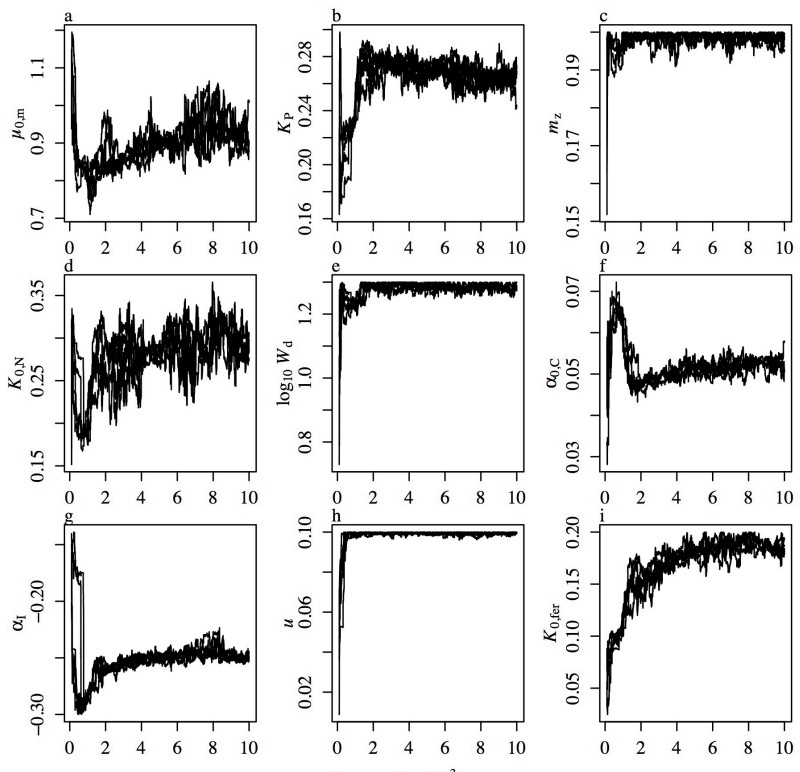

Fig. 5 Time evolution of fitted model parameters.

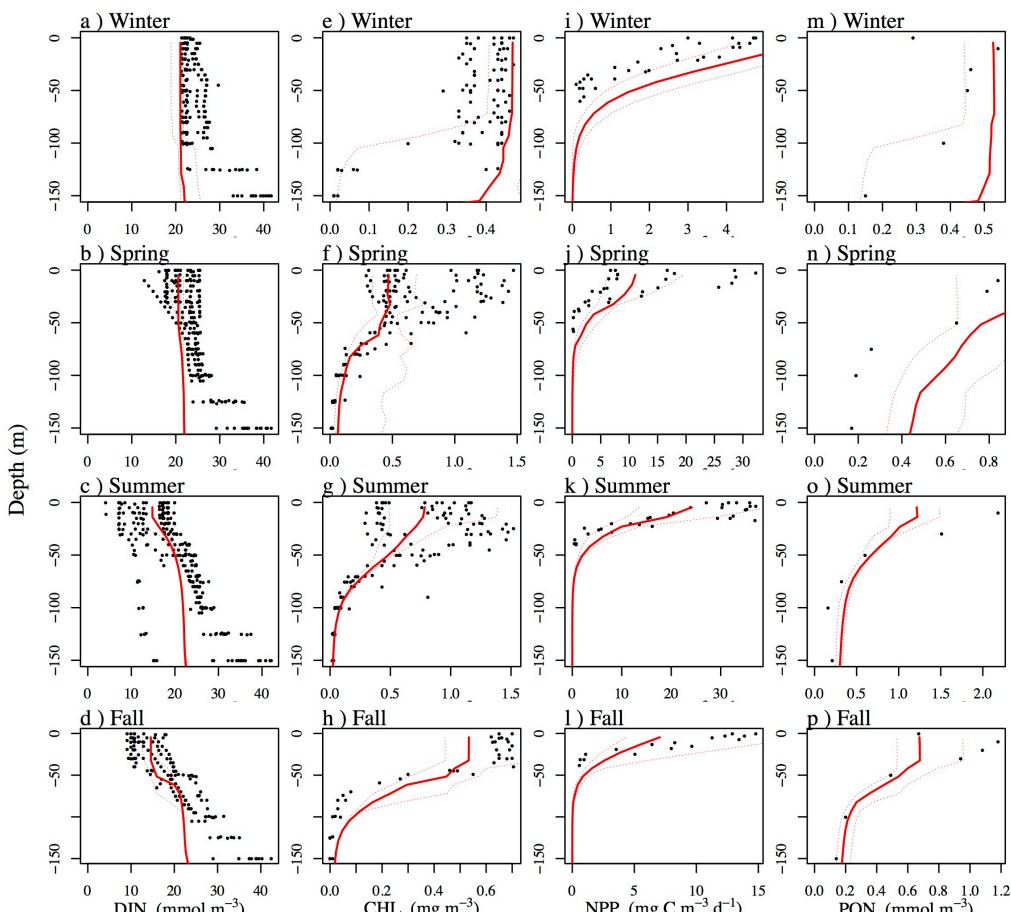

Fig. 6 . Model fittings to vertical profiles of DIN, CHL, NPP, and PON at K2

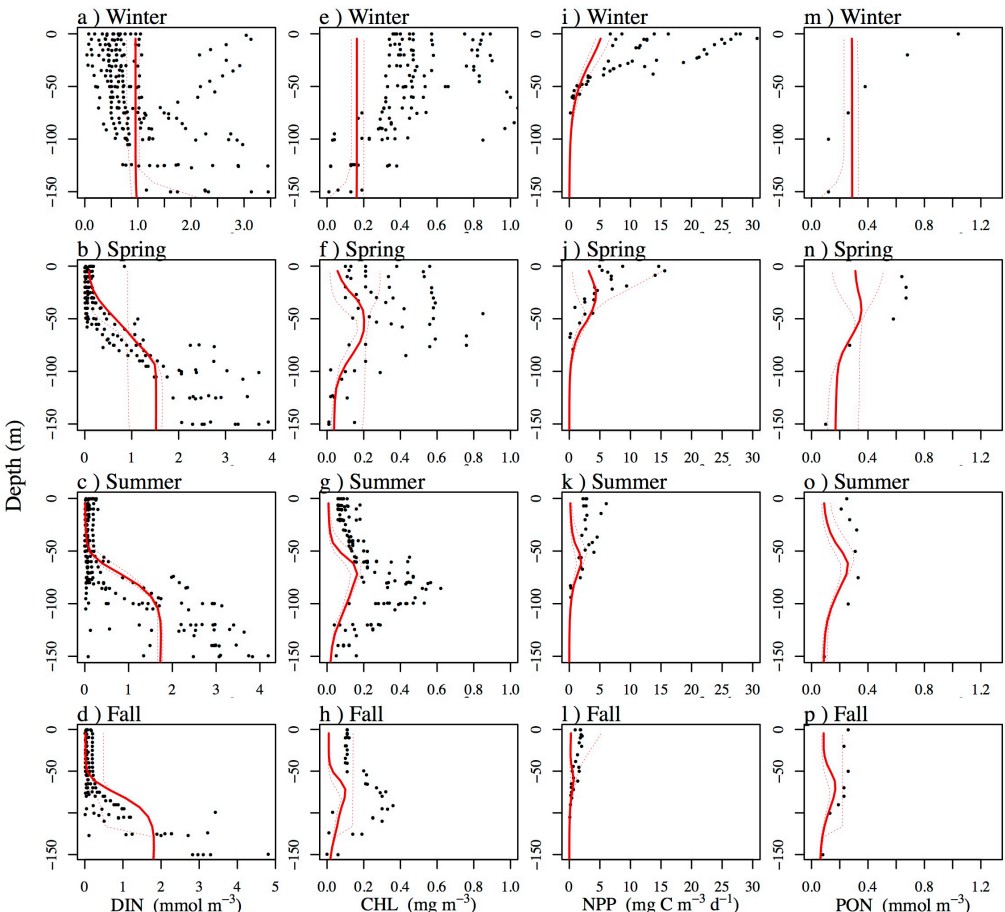

Fig. 7 . Model fittings to vertical profiles of DIN, CHL, NPP, and PON at S1

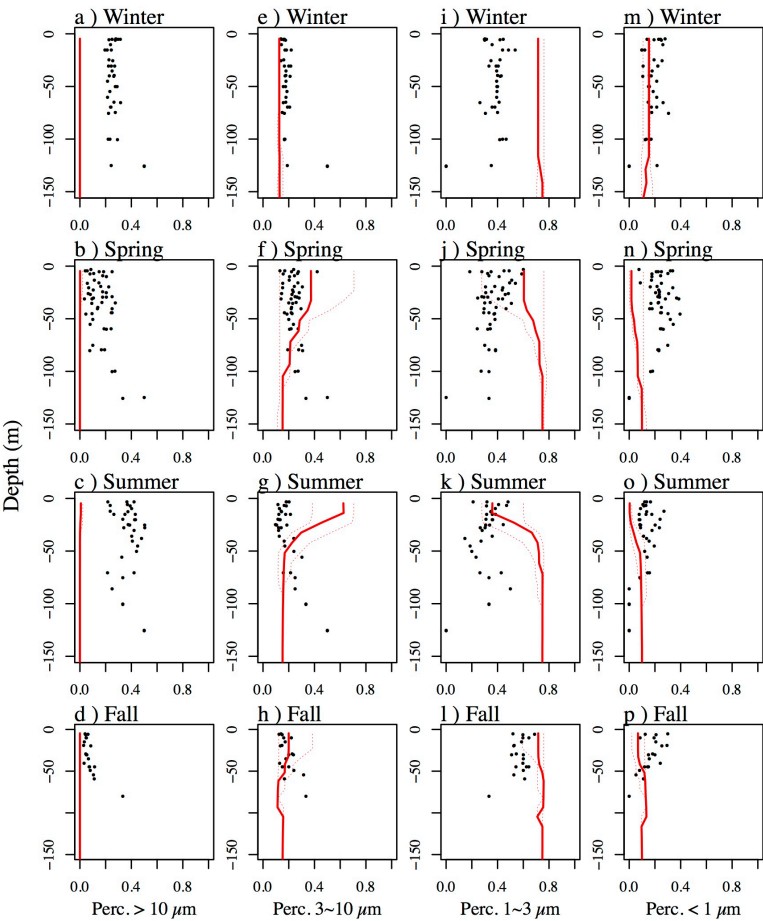

Fig. 8 . Model fittings to vertical profiles of four size fractions at K2

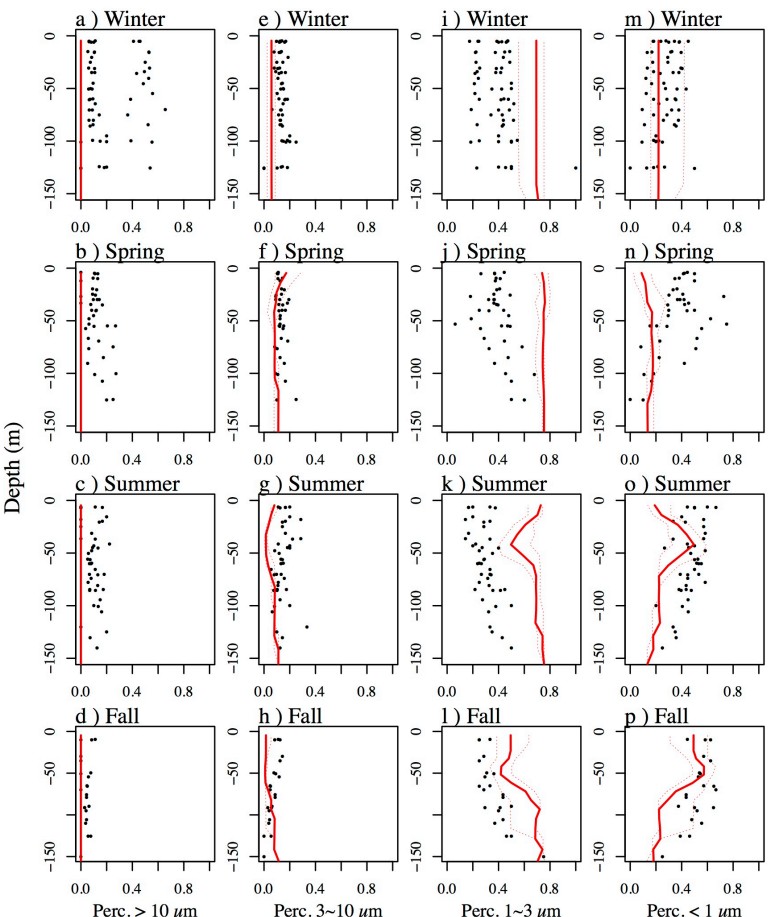

Fig. 9 . Model fittings to vertical profiles of four size fractions at S1

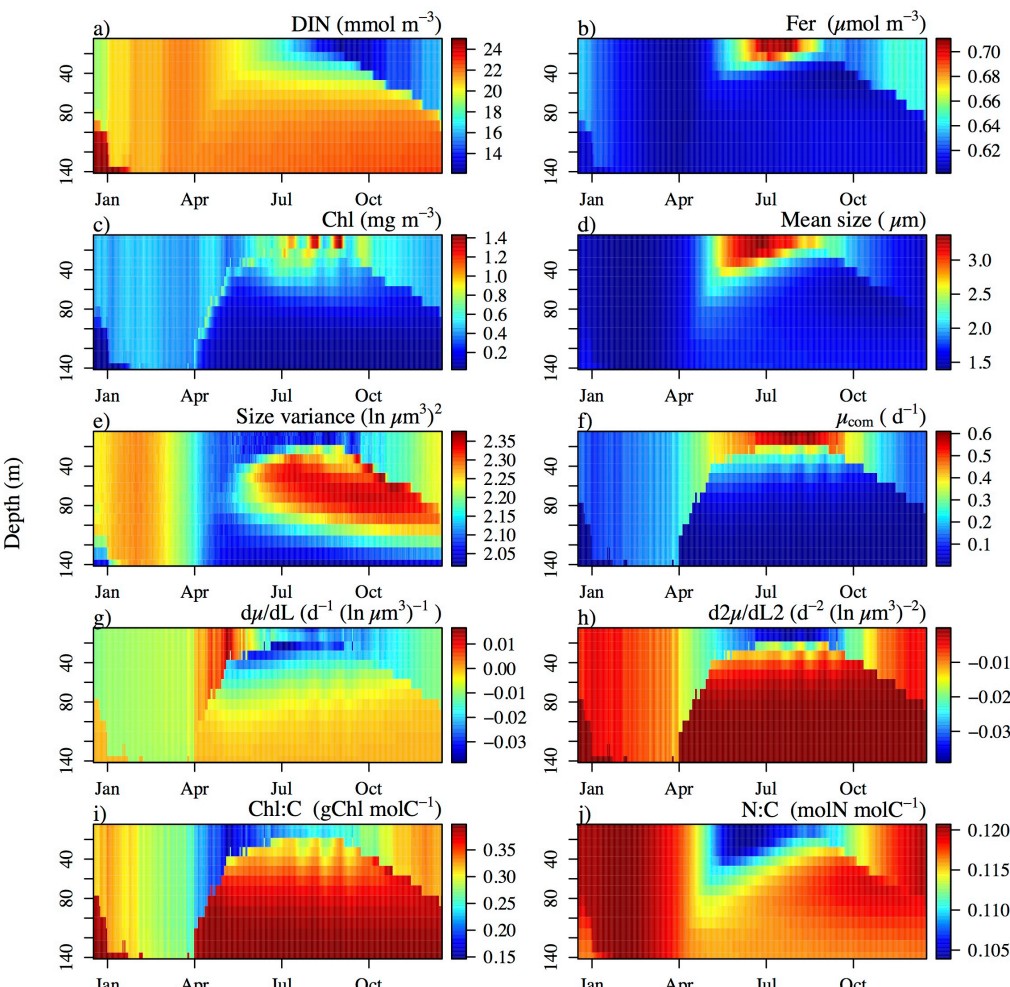

Fig. 10 . Modelled seasonal patterns at K2

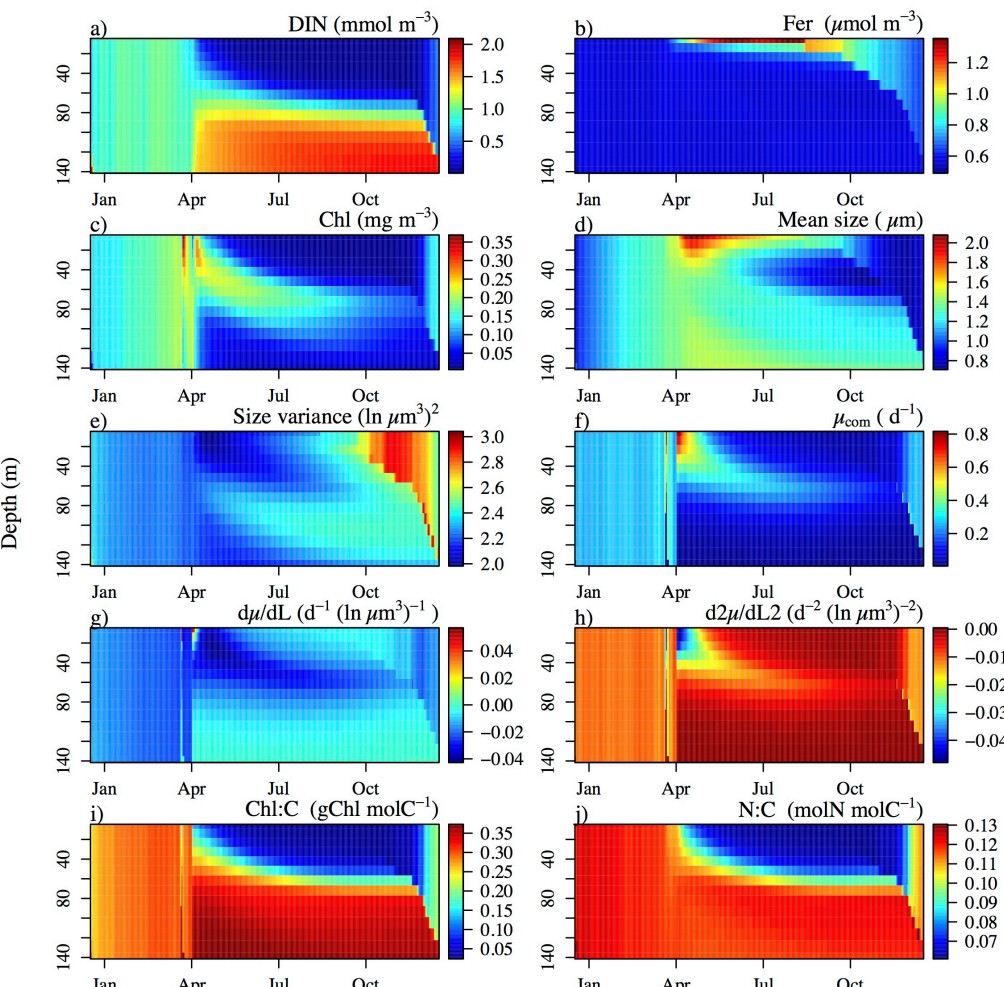

Fig. 11 . Modelled seasonal patterns at S1

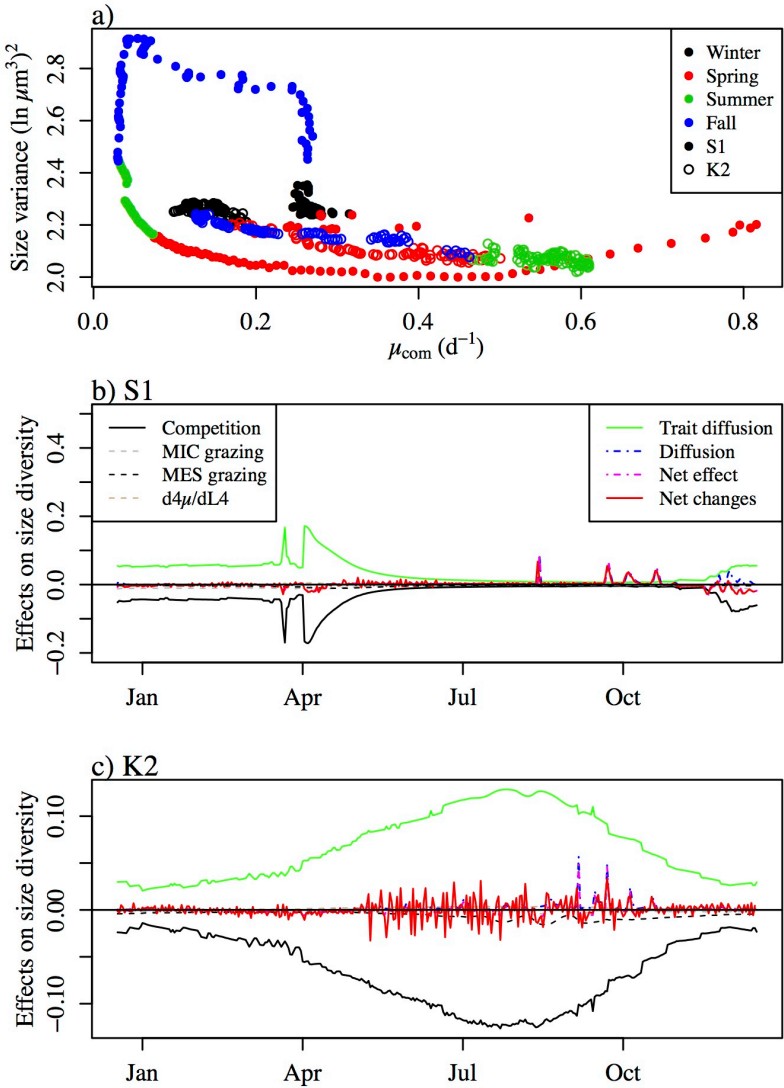

Fig. 12. (a) Scatterplots of size variance versus phytoplankton community growth rate ($\mu_{com}$). (b) Contributions of various factors to the dynamics of size variance in surface waters at S1. The term "Competition" equates to $v^2 \frac{d^2\mu}{dl^2}$. *MIC* and *MES* grazing equates to $-v^2 \frac{d^2 g_i}{dl^2}$. "d4$\mu$/dL4" equates to $v^2 u \frac{d^4\mu}{dl^4}$. "Trait diffusion" equates to $2u\mu$. All the derivatives are evaluated at the mean size. "Diffusion" means the contribution to the changes of size variance induced by diffusion with the underlying grid. "Net effect" means the sum of the above terms. "Net changes" mean the difference of size variance between adjacent days. (c) The same as (b), but at station K2.

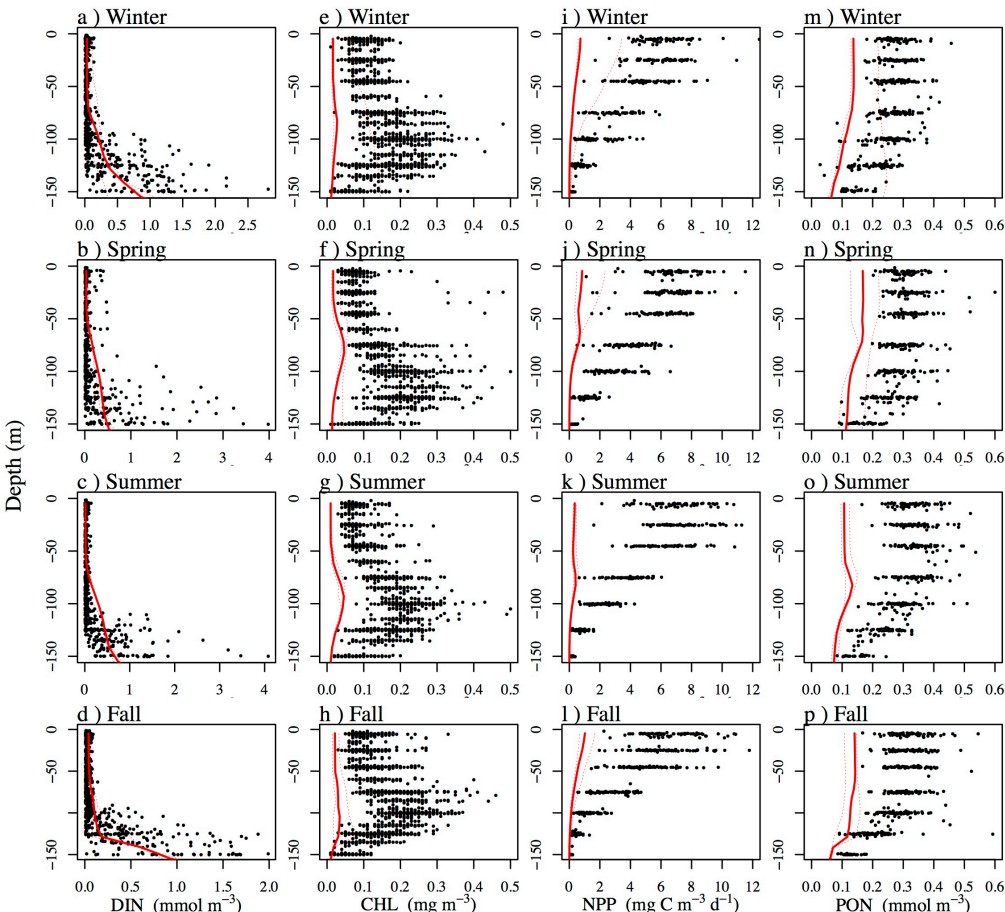

Fig. 13 . Model fittings to vertical profiles of DIN, CHL, NPP, and PON at ALOHA

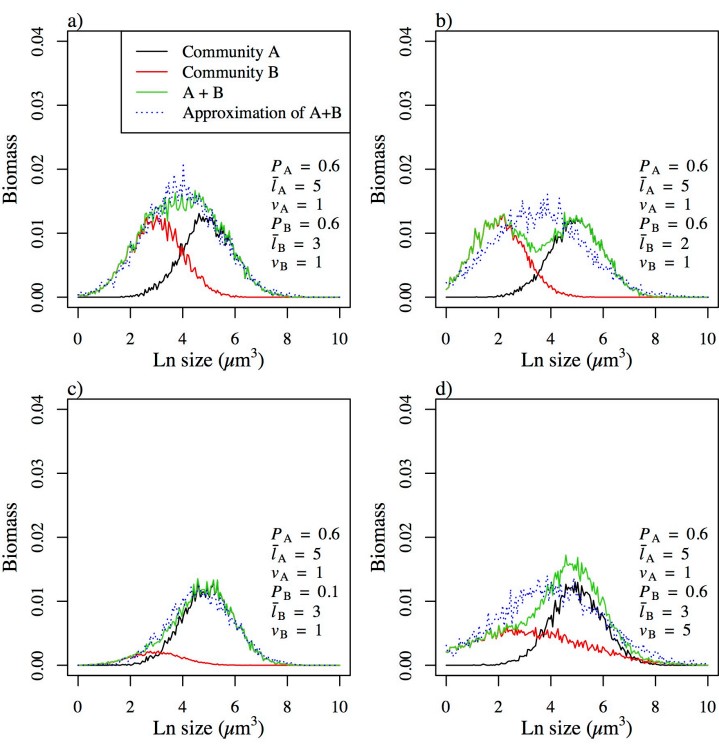

Fig. 14. Schematic diagrams for mixing of two phytoplankton communities with different biomass, mean size, and size variance, each following a lognormal size distribution.