# Peer review of "CITRATE 1.0: Phytoplankton continuous trait-distribution model with one-dimensional physical transport applied to the Northwest Pacific"

_Geoscientific Model Development, 2017_

## Editor Comment (EC1) · A. Yool (Editor) · 9 Jun 2017

Dear authors,

A colleague has reminded me that while the manuscript's code availability section includes a reference to a github project, the precise revision number used in the work within the paper is not specifically mentioned. I mention this since your code appears to have been slightly revised since submission. As such, it would be helpful to reviewers to comment on this or report a revision (or commit) number here.

More generally, when submitting a revised version of your manuscript, please ensure that this information is included in your manuscript's code availability section.

My apologies for failing to pick up on this detail when originally reviewing your submission.

Best regards,

Andrew Yool.

---

## Referee Comment (RC1) · Anonymous Referee #1 · 10 Jul 2017

This paper describes a trait-based continuum plankton model, with size as the principle trait, and successfully uses a parameter optimisation routine to extract the trait characteristics (mean size and variance) at two contrasting study sites.

Major comments. ===========

1. The lack of a size-dependent feeding preference is the single biggest limit to this model. Ideally any plankton web will have a size-range (or a size-trait continuum) for both phytoplankton and zooplankton, as size is such a structuring component of

the plankton across a broad range of sizes and trophic level. Positively, by excluding size-ranges in grazing the model presented does allow a simpler exploration of characteristics of phytoplankton size structuring.

2. Eq. 7b is missing the detrital remineralisation term, possibly where the two minus signs are.

3. P15, last paragraph. The use of trait derivatives sounds important, but it was introduced too quickly for me. Could you give a little bit further explanation?

4. The first sentence is a turn-off.

5. Section 4.2.2. Transport of moments (instead of species) is the biggest issue of this type of trait-based approach. Or is it? This section quantifies through one example for a Gaussian distribution the size of the error. But the particular example chosen seems destined to show a small error, as one community is much smaller than the other. It would be better to show the example with the greatest possible error. Would that be a Gaussian with equal biomass but very different mean size?

Minor comments. ===========

P3, L2. Distinguishing between identity and diversity in the first sentence is confusing.

P3, L22. I think you mean in practice impossible, rather than almost impossible.

P4, L15 If trait number = N, trait resolution = D, then difference = N(D-2)-1. The derivative of the difference with trait resolution is N (independent of D). So it is not exponential, it is linear, with a slope N.

P6. L16 Do you mean Eq. (4a)?; bimodal?

P8. Z = depth. [water depth sounds like the bottom depth?]

P11, L8 "and both model" – sentence has gone astray?

P14, L10 "Large phytoplankton are susceptible to light limitation" I thought it was nutrients?

Table 1. Unit of Kchl should be m-2 as written inside the -1 bracket.

[Figure]

---

## Referee Comment (RC2) · Anonymous Referee #2 · 12 Jul 2017

This is a very nice, well-written paper that describes a continuous trait model and its application to data from two stations. Continuous trait models have some nice advantages over the traditional discrete methods that are typically used, as the authors describe nicely, and this model may provide an interesting and insightful counterpoint to standard methods when applied to 3D models, as the authors state they plan to in the future. I recommend acceptance after minor revisions.

I have one major criticism of the paper, which could be resolved easily enough in the Discussion section. The criticism is that the model may be sensitive to the lognormal

distribution assumed by the authors. The model relies on describing the distribution of traits by the log-mean and the log-variance of volume. Making size the master trait is standard, and discussed by the authors, but they do not even mention the word 'log-normal' until page 18. This first of all leads the reader to assume that the authors are using a Gaussian distribution for volume, which would be disastrous, unless one reads carefully that they are considering the log-volume, which cannot be expected of every reader. Readers are likely to get confused without clarifying this earlier) The authors also cite a few papers (the oldest of which is from 2007; what did people think the size distribution of plankton was before 2007?), along with the standard use of Gaussian distributions in continuous-trait models, to justify their use of a lognormal. Arguably, a power-law distribution (sometimes referred to as a Junge distribution in this context) is much more commonly assumed in these situations, yet power laws are not discussed by the authors anywhere in the paper. Additionally, power-law distributions are often not well-described by their lower-order moments, which may in fact be divergent, and are instead better described by their lower cutoff and exponent. The authors do state that the use of a lognormal is a key assumption in their model, but they should also state that they are using a lognormal from the beginning, not just at the end, and they should also state that other size distributions are often used to model phytoplankton, and discuss how e.g. the model might look different if one were using a power-law distribution, and what this means for the authors' results. Don't get me wrong; if I had written this paper, I certainly would have started with a lognormal, because this is an easy, defensible place to start. It just merits further discussion that this is perhaps not the size distribution most people think of phytoplankton following, and that this may have (possibly large) implications for the results and how the model is set up.

Relatedly, equations for Gaussian moments cited from other models are not necessarily directly applicable to log-moments of lognormals, because a $1/x$ can pop out from taking the derivative of a logarithm (e.g. see the equation/definition of the lognormal vs. the Gaussian). I did not check every equation in the paper for this - the authors seem to have been very careful so I do not expect this is a problem - but it is worth

mentioning.

Besides this criticism, the paper scores well for significance, quality, reproducibility, and presentation. I have some specific comments below:

page 3, line 7: What is meant by 'size' here? Presumably diameter, but it should be stated, especially given the importance of 'size' to the rest of the paper, where size refers to volume.

page 4, line 27: Might be good to have a reference for the sentence that starts on line 25 and ends on line 27. . .

page 5, line 10: The authors sort of 'jump right in' to the terms and equations of the model here; an overview paragraph might be useful. What is the model trying to do? How is this accomplished? This is somewhat described in the abstract and in the Introduction, but a more detailed description of the entire modeling procedure could help some readers follow the paper much better.

page 5, line 12: What are the units for P (and for 'fer' below)? If they are what I think they should be, doesn't that make it obvious why $P^2 v$ is the term to be considered, rather than P v, so shouldn't the sentence after next be removed?

page 16, line 20: This is an interesting idea - can the authors flesh it out a bit more?

---

## Short Comment (SC1) · 12 Jul 2017

I would like to suggest that the authors make use of the release mechanism in github to tag the status of the version in the repository that defines CITRATE 1.0. This firstly makes it very clear for readers and potential user which version to be used to reproduce results of the manuscript. Secondly it allows to update and extend the code in the repository without compromise the results shown in the published paper. We are also encourage (but not enforce) authors to create a DOI of the release in order make the code persistently citable. For github releases this can easily be done using Zenodo,

see https://guides.github.com/activities/citable-code/.

Here are two suggestions which would help to improve the delivery of the code :

- The authors should be more precise on the license under which the code can be used. In the case of a "freely available" code the terminology of the Creative Commons license could be used, see https://en.wikipedia.org/wiki/Creative_ Commons_license . It is highly recommended to add an appropriate license file in the release or – even better – to add appropriate commends in the header section in each file.

- I have noticed that the repository contains a lot of swap, lock and backup files created by a text editor that has been used. It is recommended that the authors remove these files from the repository in order to improve readability and to reduce data volume.

Lutz Gross GMD Executive Editor

---

## Referee Comment (RC3) · Anonymous Referee #3 · 27 Jul 2017

This manuscript presents a newly developed model based on an emergent trait-based approach to simulate phytoplankton traits (size) and associated diversity according to environment factors. The authors chose to construct an adaptative dynamics models that employs moment closure to allow continuous distribution of traits and limit the number of variables. Indeed, the model simulates the characteristics of phytoplankton community in terms of total biomass and mean size while the diversity is approximated by the variance in size. Some important processes for phytoplankton growth, corresponding to physiological adaptation to light and variable C:N ratio are also incorporated. As such, the model described in this paper is of valuable contribution to the scientific understanding of the plankton diversity which has become a central issue of marine ecosystem management. The manuscript is well-written and gives a general overview on the ability of the model to simulate size-structured distribution of phytoplankton in various environmental conditions using two contrasted stations of the Northwest Pacific. However, I have some questions concerning the methodology which has been applied and whether/how this work can be generalized to other stations or a broader oceanic region (e.g. in the context of 3D modeling setup). Indeed, I am not familiar with the use of DRAM-type algorithm to adjust model's parameters value to observational data and it took some time to me to understand exactly the method that is implemented in this study. Therefore, a more detailed description of what is exactly done by the parameters optimization algorithm and how this will be used to apply the model to other regions would be very useful to help the reader to understand the concept behind this model. It would thus significantly raise the likely impact of this paper. I recommend these questions, detailed in the 'general comments' section below, to be addressed before the manuscript could be considered to be published in 'Geoscientific Model Development'. Some minor and more specific comments are also included at the end of this document.

General comments:

1- Technically, the proposed model setup uses a DRAM algorithm to adjust the targeted parameters values and minimize the differences between model outputs and observational data based on two available dataset for contrasted stations of the Northwest Pacific. In the discussion section (p. 21, l. 3-7), the authors argue that this model would be 'easy to couple with 3D global or regional ocean models' (see also page 5, lines 6-8). As far as I understand, the idea would be to use the single set of parameters which has been found in this study (i.e. the one which gives the highest likelihood with regards to observational data for both stations) to run the model in other oceanic regions (otherwise, I do not see how the method can be applied to large oceanic system

while seeking for a single set of parameters that would lead to the best fit to observational data over the considered region). This point is not specify in the manuscript. Could you please add further thoughts on that in your discussion section and describe the preconised method to apply this model to larger oceanic regions?

2- In the introduction section (p. 4, l. 16-17), you write that 'relatively few continuous trait-based models have been validated against oceanic observations'. The comparison that is done in results section 3.3 (p. 13-14) does not constitute a 'validation analysis' as you are using the same observational data to contrain the model's parameters and to 'validate' the results. Indeed, the specific aim of the method which is used in this study is to provide the best fit between model outputs and measured data. Therefore, the main outcome of this study is actually the parameters set you obtained after running an ensemble of 10 000 simulations. A validation analysis should involve totally independent data for model parameterization and for validation and could only be carried out if you have runned the model for a different region using the same set of parameters. Here again, a more explicit description of the aim of your method (i.e. seting up a set of parameters which can be subsequently used to run the model in other regions ?) would have been useful to avoid the confusion.

3- How do you convert the mean size and size variance into four size classes fraction? I guess the calculation is done by comparing the occurrence of each size classes from the size distribution (Gaussian distribution of the log biovolume) other time (e.g. seasonal average in fig. 8 and 9?) but this is unclear. Could you please specify the method that has been used in your method section?

4- In the introduction section, you say that functional groups (PFT) models, representing a defined and limited number of plankton types, generally 'underestimate local diversity'. You argue p.3, l. 24-26 that the main reason for that is their inability to resolve the trait space combined with their failure of representing the appropriate mechanism sustaining high level of diversity. Although these considerations are correct, I think they are not specific to aggregated models but can also apply to the model presented here.

Indeed, as you point out in the discussion section, you choose to consider the size as master trait but ignore some other major traits (temperature optima, mixotrophy, grazing resistance etc.) which may also vary among planktonic organisms of same size and enable coexistence by achieving similar fitness between different adaptation strategies (i.e. mechanisms for sustaining diversity). In that sense, I would say that the two techniques have a similar bias of taking into account a limited number of traits and mechanisms to explain the huge plankton diversity. Please modify the introduction to consider this point. Moreover, another difference between the two methods is that the measure of diversity that is provided by moment-closure models corresponds to a relative measure of diversity (variance in size in this case) which only allow a relative and comparative analysis of the phytoplankton diversity (in time and space) but does not provide any absolute measure of diversity (number of taxons or species) to compare with observational data.

p. 4, l. 9, you argue that 'the factors controlling diversity can be directly quantified and better understood' with the continuous trait-based models. This sentence is not unclear. Could you specify how and why are the factors (which factors?) controlling diversity better characterized using the latter method ?

Specific comments:

Please put 'et al' in italic while citing referenced publications throughout the manuscript.

Model description

P5 L12: Add the unit of P P5 L18: Figure 1: What is the inset in the box on the top left (with probability axis)? Please add a description in the figure caption. P5 L22: Please provide more explanations on the role of the trait diffusion parameter P6 L7-11: Please provide the references for the growth dependences to light, nutrient concentration and temperature. P6 L17: Eq. (5A) should be Eq. (4A) Section 2: I would suggest to separate the description of the biogeochemical equations (section 2.2) and the 1D implementation (section 2.3). Therefore the paragraph l. 9-13 on page

7 should be moved to the section 2.3 and the name of the section 2.2 should mention only 'Biogeochemical model (nutrient, zooplankton etc.)' P7 L13: The sentence 'the 1D model contains only biological tracers' is unclear. It should be replaced by 'the biological model is runned offline' or something similar P8 L5: Please replace 'water depth' by 'the depth of the water column depth'. P8 L11: Please verify the equation for detritus (- -). P9 L12-14: Do you assume that the surface mixed layer has a depth of 100 m? (the explanation for the use of the threshold of $Kv > 10^{-3}$ $m^2.s^{-1}$ is unclear). Why do you use a different parameterization for the MLD calculation for phytoplankton growth and MLD showed on fig. 2 from observational data (page 12, line 15)? P11 L8: 'and both model'? Please check the sentence. Fig. 2 caption: Add the description of the white scatter plots (MLD) in the legend. Change '. . .at station S1. (f-i) The same for station K2' Fig. 2: Check x-axis thick labels (subplots b, c, f and g). Fig. 3: Add x-axis labels.

Results

In general, there is some discussion points that are included in the results section and that should rather be discussed in section 4.

P12 L18-25: This section describes the physical forcing and does not concern a result of the simulation. I would suggest to move this part in section 2.3 (method). P12 L23: 'with the model estimates of MLD consistent with those measured from in situ temperature and salinity profiles': it is not clear what you are comparing exactly. (Please also add a reference to the figure showing that. What are white scatter plots on fig. 2 b and f?). Fig. 4 caption: Remove the 's' in 'log-likelihood'. Replace (b-j) by (b-i) P13 L6: The SSqE of the smallest size fraction (fig. 4, q) also increases with time at S1 P13 L9: The figure 5 is not commented in the text. Please add a sentence to describe the trend. P13 L 11-16: The discussion on the value of the trait diffusion parameter should appear in the discussion section. Fig. 6 and 8 captions: Complete the caption with the position of the different variables. P13 L20: 'the higher surface concentrations' P13 L22-24: Isn't it in apparent contradiction with the fact that you argued that the modeled

MLD is in agreement with observational data (page 12, line 23)? P13 L22-24: discussion P14 L5: The observational data on the size fractions are relatively noisy. Could you please provide more details on how these data were obtained (sampling methods, size measurments, sampling frequency) in section 2.5? At station K2, the size distribution in unclear in data and the model overestimate the proportion of 3-10 $\mu$m size class. At station S1, observational data show the dominance of smaller cells but do not show the vertical structure of the size distribution that is simulated by the model with smaller cells at the surface. Please add a comment on this. P14 L7-9: 'At station S1'. Do you mean station 'K2'? P14 L10: discussion P14 L19: 'following stratification which occurs earlier in S1 than K2' P14 L24: Show a figure of Chl/C ratio Fig. 10, g: High growth rate at the surface at K2 despite low TIN and low Chl a concentrations? P15 L5-6: discussion P15 L24 – P16 L4: discussion P16 L3: The 'dynamic equilibrium theory proposed by Huston is only briefly mentionned (see also page 17, l, 1-2). This hypothesis implies that, under non-equilibrium conditions, the outcomes of the competition depend on the timescale of the competitive displacement and the relative rate of change in competitive abilities of each competing species. This point should be further developed and discussed according to your results on diversity in the discussion part of the manuscript. P16 L6-9: As I said above, the role of the trait diffusion in maintaining diversity and the way it is used in the model is a bit tricky to understand. You could perhaps add a paragraph in the method section to clarify this point which is then only discussed briefly page 17, lines 10-14.

Discussion

P17 L22-27: There is no figures showing the N:C and Chl: C ratio patterns at the two stations. P18 L3: 'Instead, we employ ...': The word 'instead' seems unappropriate as you mention a very different process than in the previous sentence: the trade-offs between maximal growth rate and nutrient affinity in the phytoplankton is not related to the size-dependence of the grazing by predators.

As you mention l. 17-19, the role of grazing in shaping phytoplankton community has

been shown to be crucial. In order to take into account a various palatability of phytoplankton for zooplankton feeding according to the cell size, the model should ideally involved a larger number of predator size classes (and/or, at least, an additional mesozooplankton size class) which would lead to much more complexity in the model. In addition to the predator-prey size-ratio, the predators' feeding mode (Mariani et al., 2013) and the formulation that is used to constrain the herbivorous impact on primary producers community composition are also very important. Please add the information on what kind of predation function you are using in your model in 'Model description'. Also, this points should be mentionned in the discussion section (add other references such as Anderson et al., 2010 ; Prowe et al., 2012).

P20 L12: 'other mecanisms such as vertical migration': I am not sure that vertical migration is a very common process in small phytoplankton populations that are found at the surface of subtropical waters during the summer. What about the nutrient limitation terms? What should be the half-saturation constants for nitrogen/phosphorus uptake in the 1-3 $\mu$m size class found in observationnal data?

Please also note the supplement to this comment:
https://www.geosci-model-dev-discuss.net/gmd-2017-104/gmd-2017-104-RC3-supplement.pdf

---

## Author Comment (AC1) · 23 Oct 2017

Dear Dr. Gross,

Thanks for the excellent suggestion for releasing and tagging the DOI for the codes. We have released CITRATE on github to allow potential users to reproduce the results shown in the paper. We also tagged the DOI (10.5281/zenodo.1034805) for the release from Zenodo.

Regarding the license, we have copied the MIT license file under the main directory.

[Figure]

We also removed the unnecessary swap and backup files from the repository folder.

Bingzhang Chen

---

## Author Comment (AC2) · 23 Oct 2017

Dear Dr. Yool,

Thanks for pointing out the problem of code versions. We have released CITRATE on github to allow users to reproduce the results shown in the paper. We also tagged the DOI (10.5281/zenodo.1034805) as also suggested by L. Gross.

Best regards, Bingzhang Chen

---

## Author Response (AR1)

Dear Dr. Yool,

Thank you very much for considering our manuscript. We also sincerely thank the three reviewers for their substantial efforts and insightful comments that have helped us improve the manuscript. We have revised the manuscript and modified the model according to the comments. The major revisions we have made include:

1. We have improved the iron cycle by adding a tracer for detrital iron, updated the atmospheric iron deposition data from Luo et al. (2008), and optimized iron-related parameters to the outputs of dissolved iron from Aumont et al. (2003). Accordingly, we also modified the bottom boundary conditions for DIN and dissolved iron (the bottom boundary for dissolved iron cannot be closed since the deposition from the atmosphere continuously adds to the total inventory of iron).

2. We have modified the approach of transporting size variance in diffusion following the work of Bruggeman (2009).

3. As two reviewers suggested, we have added a mesozooplankton compartment and size-selective feeding for two zooplankton compartments.

4. We have modified the codes for parallel computing of the DRAM algorithm to increase its efficiency for optimizing parameter sets.

5. As one reviewer requested, we have expanded the introduction and explanation of the DRAM algorithm to make it easier to follow.

Below are our point-by-point responses to the reviewers' comments.

**==============================================================**
Reviewer #1

This paper describes a trait-based continuum plankton model, with size as the principle trait, and successfully uses a parameter optimisation routine to extract the trait characteristics (mean size and variance) at two contrasting study sites.

Major comments. ===========
1. The lack of a size-dependent feeding preference is the single biggest limit to this model. Ideally any plankton web will have a size-range (or a size-trait continuum) for both phytoplankton and zooplankton, as size is such a structuring component of the plankton

across a broad range of sizes and trophic level. Positively, by excluding size-ranges in grazing the model presented does allow a simpler exploration of characteristics of phytoplankton size structuring.

[Response] We agree that adding a size-dependent feeding preference might be useful for investigating the grazing effects on phytoplankton size structure. Accordingly, we have added size-dependent feeding and another zooplankton compartment into the model as also suggested by another reviewer. Please see Section **2.3 Zooplankton size-dependent grazing**.

2.   Eq. 7b is missing the detrital remineralisation term, possibly where the two minus signs are.

[Response] Yes, it was a typo in the previous version. We have corrected the equation in the current version (Eq. 3).

3.   P15, last paragraph. The use of trait derivatives sounds important, but it was introduced too quickly for me. Could you give a little bit further explanation?

[Response] Here we have modified the text to avoid too much interpretation in the Results section 3.4:

"The values of $\frac{d^2\mu(l)}{dl^2}$ were always negative in all times at both stations, suggesting that without "trait diffusion", size variance would decrease toward zero (Eq. 7c). This highlights the importance of trait diffusion (which can be interpreted as genetic mutation or transgenerational phenotypic plasticity) to sustain diversity."

We added some text in the Discussion section 4.1.1 to explain in more detail on the derivatives:

"In particular, the second derivative of the growth rate at mean size, $\frac{d^2\mu(l)}{dl^2}$, can be conveniently perceived as a proxy for the intensity of resource competition (The more

concave is the curve of $\mu(l)$, the more intense is the competition, i.e., the fitness of suboptimal species decreases more steeply with distance from the optimal size)."

4. The first sentence is a turn-off.
[Response] Sorry, we do not understand what exact sentence the reviewer meant.

5. Section 4.2.2. Transport of moments (instead of species) is the biggest issue of this type of trait-based approach. Or is it? This section quantifies through one example for a Gaussian distribution the size of the error. But the particular example chosen seems destined to show a small error, as one community is much smaller than the other. It would be better to show the example with the greatest possible error. Would that be a Gaussian with equal biomass but very different mean size?

[Response] The transport of moments is indeed an issue that bothered us. Fortunately, Jorn Brugemann in his thesis (Brugemann, 2009) has pointed out that the raw moments of the biomass distribution can behave as normal tracers in diffusion and advection. We have followed Jorn's suggestion to modify the transport of variance in our model (*see* **Sect. 2.1 General description of the 1D ecosystem model**). We have followed the reviewer's suggestion to test this approach in Fig. 14. It seems clear that when the differences between the two mean sizes or variances are large enough, the mixed community would not follow a Gaussian distribution. But when the two communities differ only moderately in mean size or variance, the approximate should be reasonable.

Minor comments. ===========
P3, L2. Distinguishing between identity and diversity in the first sentence is confusing.

[Response] The "identity" is a term that theoretical ecologists often use (e.g. Tilman et al. PNAS 1997). It means the (dominant) functional traits themselves in contrast to the number of species present. We have added an annotation beside.

P3, L22. I think you mean in practice impossible, rather than almost impossible.

[Response] Yes, we have modified the wording to "which is however impossible in practice due to computational limits".

P4, L15 If trait number = N, trait resolution = D, then difference = N(D-2)-1. The derivative of the difference with trait resolution is N (independent of D). So it is not exponential, it is linear, with a slope N.

[Response] Yes, we have replaced the word "exponential" with "linear".

P6. L16 Do you mean Eq. (4a)?; bimodal?
[Response] Yes. Sorry, it was a typo. We have corrected it to Eq. (10a). It should be unimodal because phytoplankton maximal growth rate usually peaks at nano size range.

P8. Z = depth. [water depth sounds like the bottom depth?]
[Response] Z is the depth of the model grid. We have made it clearer in the revision.

P11, L8 "and both model" – sentence has gone astray?
[Response] Sorry, it was a typo. We have deleted it.

P14, L10 "Large phytoplankton are susceptible to light limitation" I thought it was nutrients?
[Response] Here we meant that because large phytoplankton are susceptible to light limitation, the effect of light limitation offsets the effect of replete nutrients on phytoplankton size at depth, which can explain why the proportion of size-fractionated Chl does not change with depth as predicted based on nutrient effects alone. However, based on the revised modeled results, we have deleted this sentence during revision.

Table 1. Unit of Kchl should be m-2 as written inside the -1 bracket.
[Response] Sorry, it was a typo. We have corrected it.

**================================================================**

Reviewer #2

This is a very nice, well-written paper that describes a continuous trait model and its application to data from two stations. Continuous trait models have some nice advantages over the traditional discrete methods that are typically used, as the authors describe nicely, and this model may provide an interesting and insightful counterpoint to standard methods when applied to 3D models, as the authors state they plan to in the future. I recommend acceptance after minor revisions.

[Response] Thank you for your positive comments on this manuscript!

I have one major criticism of the paper, which could be resolved easily enough in the Discussion section. The criticism is that the model may be sensitive to the lognormal distribution assumed by the authors. The model relies on describing the distribution of traits by the log-mean and the log-variance of volume. Making size the master trait is standard, and discussed by the authors, but they do not even mention the word 'lognormal' until page 18. This first of all leads the reader to assume that the authors are using a Gaussian distribution for volume, which would be disastrous, unless one reads carefully that they are considering the log-volume, which cannot be expected of every reader. Readers are likely to get confused without clarifying this earlier) The authors also cite a few papers (the oldest of which is from 2007; what did people think the size distribution of plankton was before 2007?), along with the standard use of Gaussian distributions in continuous-trait models, to justify their use of a lognormal. Arguably, a power-law distribution (sometimes referred to as a Junge distribution in this context) is much more commonly assumed in these situations, yet power laws are not discussed by the authors anywhere in the paper. Additionally, power-law distributions are often not well-described by their lower-order moments, which may in fact be divergent, and are instead better described by their lower cutoff and exponent. The authors do state that the use of a lognormal is a key assumption in their model, but they should also state that they are using a lognormal from the beginning, not just at the end, and they should also state that other size distributions are often used to model phytoplankton, and discuss how e.g. the model might look different if one were using a power-law distribution, and what this means for the authors' results. Don't get me wrong; if I had written this paper, I certainly would have started with a lognormal, because this is an easy, defensible place to start. It just merits further discussion that this is perhaps not the size distribution most people think of

phytoplankton following, and that this may have (possibly large) implications for the results and how the model is set up.

[Response] The reviewer is entirely correct that phytoplankton size distribution is more commonly assumed to follow a power-law distribution (Sheldon et al. 1972; Gin et al. 1999; Cermeño et al. 2006). The slope of log abundance versus log size (e.g. biovolume) tends to be between −0.7 and −1 (Cermeño et al. 2006), suggesting that the slope of log biomass versus log size should be between 0 and 0.3. However, aside from fact that the power-law distribution is unrealistic in predicting phytoplankton biomass at the limit of either the largest or the smallest size, it is much more inconvenient in terms of mathematical manipulation (e.g. calculating mean and variance) compared to the normal distribution. We have included these points in the discussion in Section 4.2.1.

Relatedly, equations for Gaussian moments cited from other models are not necessarily directly applicable to log-moments of lognormals, because a $1/x$ can pop out from taking the derivative of a logarithm (e.g. see the equation/definition of the lognormal vs. the Gaussian). I did not check every equation in the paper for this - the authors seem to have been very careful so I do not expect this is a problem - but it is worth mentioning.

[Response] Actually, calculating the derivatives does not involve taking the derivative of a logarithm because the growth rates are directly dependent on log cell volume (see Eqn. 10), not on cell volume itself. In other words, in our model equations, we only have log volume and we do not have any calculations on cell volume directly.

Besides this criticism, the paper scores well for significance, quality, reproducibility, and presentation. I have some specific comments below:

page 3, line 7: What is meant by 'size' here? Presumably diameter, but it should be stated, especially given the importance of 'size' to the rest of the paper, where size refers to volume. [Response] Thanks for pointing it out. We have modified it to "ESD".

page 4, line 27: Might be good to have a reference for the sentence that starts on line 25 and ends on line 27: : :
[Response] We have added the references "(Matsumoto et al., 2014; Wakita et al., 2016)".

page 5, line 10: The authors sort of 'jump right in' to the terms and equations of the model here; an overview paragraph might be useful. What is the model trying to do? How is this accomplished? This is somewhat described in the abstract and in the Introduction, but a more detailed description of the entire modeling procedure could help some readers follow the paper much better.

[Response] This is really a helpful suggestion to improve the readability and overall organization of the paper. We have added an overall summary at the beginning of Sect. 2 (Model description).

page 5, line 12: What are the units for P (and for 'fer' below)? If they are what I think they should be, doesn't that make it obvious why P^2 v is the term to be considered, rather than P v, so shouldn't the sentence after next be removed?

[Response] We have added the units for P, and fer. However, we do not think that the units themselves can justify the use of P^2 v instead of Pv. We have followed the pioneering work of Bruggeman (2009) to use $P\left(v + \bar{l}^2\right)$ as the tracer for variance to be transported.

page 16, line 20: This is an interesting idea - can the authors flesh it out a bit more?

[Response] Yes, we have explained it in more detail in the Discussion section 4.1.1: "In particular, the second derivative of the growth rate at mean size, $\frac{d^2\mu(l)}{dl^2}$, can be conveniently perceived as a proxy for the intensity of resource competition (The more concave is the curve of $\mu(l)$, the more intense is the competition since the fitness of suboptimal species decreases more steeply with distance from the optimal size)."

**===============================================================**

Reviewer #3

This manuscript presents a newly developed model based on an emergent trait-based approach to simulate phytoplankton traits (size) and associated diversity according to environment factors. The authors chose to construct an adaptative dynamics models that employs moment closure to allow continuous distribution of traits and limit the number of variables. Indeed, the model simulates the characteristics of phytoplankton community in terms of total biomass and mean size while the diversity is approximated by the variance in size. Some important processes for phytoplankton growth, corresponding to physiological adaptation to light and variable C:N ratio are also incorporated. As such, the model described in this paper is of valuable contribution to the scientific understanding of the plankton diversity which has become a central issue of marine ecosystem management. The manuscript is well-written and gives a general overview on the ability of the model to simulate size-structured distribution of phytoplankton in various environmental conditions using two contrasted stations of the Northwest Pacific. However, I have some questions concerning the methodology which has been applied and whether/how this work can be generalized to other stations or a broader oceanic region (e.g. in the context of 3D modeling setup). Indeed, I am not familiar with the use of DRAM-type algorithm to adjust model's parameters value to observational data and it took some time to me to understand exactly the method that is implemented in this study. Therefore, a more detailed description of what is exactly done by the parameters optimization algorithm and how this will be used to apply the model to other regions would be very useful to help the reader to understand the concept behind this model. It would thus significantly raise the likely impact of this paper.

I recommend these questions, detailed in the 'general comments' section below, to be addressed before the manuscript could be considered to be published in 'Geoscientific Model Development'. Some minor and more specific comments are also included at the end of this document.

General comments:

1- Technically, the proposed model setup uses a DRAM algorithm to adjust the targeted parameters values and minimize the differences between model outputs and observational data based on two available dataset for contrasted stations of the Northwest Pacific. In the discussion section (p. 21, l. 3-7), the authors argue that this model would be 'easy to couple with 3D global or regional ocean models' (see also page 5, lines 6-8). As far as I understand,

the idea would be to use the single set of parameters which has been found in this study (i.e. the one which gives the highest likelihood with regards to observational data for both stations) to run the model in other oceanic regions (otherwise, I do not see how the method can be applied to large oceanic system while seeking for a single set of parameters that would lead to the best fit to observational data over the considered region). This point is not specify in the manuscript. Could you please add further thoughts on that in your discussion section and describe the preconised method to apply this model to larger oceanic regions?

[Response] This is a good point that we had not explained with sufficient clarity in the previous manuscript. Yes, our intention is to use the single set of parameter values, obtained by fitting the model to the data from the two observation stations, as an initial estimate of parameter values for 3D simulations. As a test of the feasibility of this approach we present an example in which we use the parameter values optimized from the two stations K2 and S1 to model another independent station (the well known station ALOHA) in the North Pacific (Results section 3.5). Although the results are not as satisfactory as we wished, the parameter values obtained in the present study nevertheless provide a useful initial estimate for modeling other stations and for 3D applications. We propose for later studies to combine the "transport matrix" technique with DRAM or other parameter optimization techniques to calibrate 3D models in the discussion (Sect. 4.2.5).

2- In the introduction section (p. 4, l. 16-17), you write that 'relatively few continuous trait-based models have been validated against oceanic observations'. The comparison that is done in results section 3.3 (p. 13-14) does not constitute a 'validation analysis' as you are using the same observational data to contrain the model's parameters and to 'validate' the results. Indeed, the specific aim of the method which is used in this study is to provide the best fit between model outputs and measured data. Therefore, the main outcome of this study is actually the parameters set you obtained after running an ensemble of 10 000 simulations. A validation analysis should involve totally independent data for model parameterization and for validation and could only be carried out if you have runned the model for a different region using the same set of parameters. Here again, a more explicit description of the aim of your method (i.e. seting up a set of parameters which can be subsequently used to run the model in other regions ?) would have been useful to avoid the confusion.

[Response] Yes, we totally agree. We have changed the word "validate" to "calibrate". We indeed attempted to validate the optimized model against independent datasets at station ALOHA. And we have emphasized our goal is "**CITRATE** is intended to be a starting model for later incorporation into three-dimensional (3D) general ocean circulation models (GCMs) and for further development of more comprehensive trait-based models" at the end of Introduction.

3- How do you convert the mean size and size variance into four size classes fraction? I guess the calculation is done by comparing the occurrence of each size classes from the size distribution (Gaussian distribution of the log biovolume) other time (e.g. seasonal average in fig. 8 and 9?) but this is unclear. Could you please specify the method that has been used in your method section?

[Response] Yes. We have added the method to calculate the four size classes of Chl based on phytoplankton biomass (in terms of nitrogen), mean size, and size variance (after Eq. 11d in Sect. 2.2). Actually, because we assumed lognormal distribution for phytoplankton cell volume, the Chl distribution is no longer a lognormal distribution (because the modeled chlorophyll to carbon ratio depends on cell volume). In order to obtain accurate estimates of the chl distribution, we had to discretize the phytoplankton size spectra to numerically estimate the fractions of size fractionated Chl.

4- In the introduction section, you say that functional groups (PFT) models, representing a defined and limited number of plankton types, generally 'underestimate local diversity'. You argue p.3, l. 24-26 that the main reason for that is their inability to resolve the trait space combined with their failure of representing the appropriate mechanism sustaining high level of diversity. Although these considerations are correct, I think they are not specific to aggregated models but can also apply to the model presented here. Indeed, as you point out in the discussion section, you choose to consider the size as master trait but ignore some other major traits (temperature optima, mixotrophy, grazing resistance etc.) which may also vary among planktonic organisms of same size and enable coexistence by achieving similar fitness between different adaptation strategies (i.e. mechanisms for sustaining diversity). In that sense, I would say that the two techniques have a similar bias of taking into account a limited number of traits and mechanisms to explain the huge plankton diversity. Please modify the introduction to consider this point. Moreover, another difference between the two methods is

that the measure of diversity that is provided by moment-closure models corresponds to a relative measure of diversity (variance in size in this case) which only allow a relative and comparative analysis of the phytoplankton diversity (in time and space) but does not provide any absolute measure of diversity (number of taxons or species) to compare with observational data.

[Response] As admitted in the Discussion, we agree that our approach also suffers from lacking some of the important functional traits so that it also underestimates diversity. We also note in Sect. 4.2.1 that a promising approach might be to combine the discrete functional group approach with the continuous trait approach, to include the merits of both approaches. We have added the following text in the Introduction:
"Although this approach might overlook some other important traits that are not related to size and thereby underestimate trait diversity to some extent, it serves as a starting point for later development of more comprehensive diversity models that can include more traits or be integrated with the discrete functional group approach."

Indeed, the diversity metrics of our model cannot be directly comparable to the classic definition of "Richness" as the reviewer mentioned. Maybe other metrics like the Shannon-Wiener index that consider species evenness can be better compared to the trait variance we used in our approach. These metrics can be also calculated from observational data (e.g. size-fractionated Chl.). We have added "The trait variance, treated as a tracer in the model, serves as a measure of trait diversity, although it cannot be simply equated to species richness but may be converted to other diversity metrics like the Shannon-Wiener index (Quintana et al., 2008). The diversity of functional traits is arguably a better diversity index than species richness to relate to ecosystem functioning (Loreau et al., 2001)." in page 4.

p. 4, l. 9, you argue that 'the factors controlling diversity can be directly quantified and better understood' with the continuous trait-based models. This sentence is not unclear. Could you specify how and why are the factors (which factors?) controlling diversity better characterized using the latter method ?

[Response] We meant that from the equations of moment closure, the factors controlling diversity can be directly quantified and better understood because the diversity itself is a tracer, and the sinks (e.g. the second derivative of growth rate indicating resource

competition) and sources (e.g. the trait diffusion terms indicating the effects of mutation) are given explicitly. We have revised the sentence to "Thus, the continuous trait-based model has the advantage that the factors controlling diversity can be directly quantified and better understood because the sources (e.g. speciation or immigration) and sinks (e.g. resource competition) for diversity are specified explicitly." to make it clearer.

Specific comments:

Please put 'et al' in italic while citing referenced publications throughout the manuscript.

[Response] Thanks for the comment. But it seems that the GMD format does not require to put 'et al' in italic.

Model description

P5 L12: Add the unit of P

[Response] Added.

P5 L18: Figure 1: What is the inset in the box on the top left (with probability axis)? Please add a description in the figure caption.

[Response] We have added a description in the figure caption.

P5 L22: Please provide more explanations on the role of the trait diffusion parameter

[Response] Yes, we have added a sentence "$u$ is the trait diffusion parameter, which describes the probability of the parental size $l(i)$ changing to adjacent size values $l(i-1)$ or $l(i+1)$ in offspring cells (Merico et al., 2014)." after Eq. 7c.

P6 L7-11: Please provide the references for the growth dependences to light, nutrient concentration and temperature.

[Response] We have added a sentence "Following previous studies (Flynn, 2003; Geider et al., 1997; Follows et al., 2007; Chen and Laws, 2017)," before this sentence.

P6 L17: Eq. (5A) should be Eq. (4A)

[Response] Thanks for pointing it out. We have corrected it (now Eq. (10A)).

Section 2: I would suggest to separate the description of the biogeochemical equations (section 2.2) and the 1D implementation (section 2.3). Therefore the paragraph l. 9-13 on

page 7 should be moved to the section 2.3 and the name of the section 2.2 should mention only 'Biogeochemical model (nutrient, zooplankton etc.)'

[Response] We have followed the suggestion to separate the description of the ecosystem model (now section 2.1) from that of the 1D implementation (now section 2.4).

P7 L13: The sentence 'the 1D model contains only biological tracers' is unclear. It should be replaced by 'the biological model is runned offline' or something similar

[Response] We have rewritten the sentence to "**For computational efficiency, instead of explicitly solving the complete moment, temperature, and salinity equations, we imported the physics variables that are directly relevant to the ecological processes from external data products.**".

P8 L5: Please replace 'water depth' by 'the depth of the water column depth'.
[Response] We have changed it to "$z$ **is the depth of the model grid (m)".**

P8 L11: Please verify the equation for detritus (- -).
[Response] Sorry, it was a typo in the previous version. We have corrected the detritus equation.

P9 L12-14: Do you assume that the surface mixed layer has a depth of 100 m? (the explanation for the use of the threshold of Kv > 10-3 m2.s-1 is unclear).
[Response] Yes. We just tried to use a simple calculation to demonstrate why we use the threshold of $Kv > 10^{-3} \text{ m}^2 \text{ s}^{-1}$.

Why do you use a different parameterization for the MLD calculation for phytoplankton growth and MLD showed on fig. 2 from observational data (page 12, line 15)?
[Response] This is a good question. The definition of MLD based on observed temperature and salinity profiles is because there were no observational data for vertical eddy diffusivity (Kv). In the model, since we have the Kv from 3D model outputs but do not have salinity variables, it is best to define MLD based on Kv.

P11 L8: 'and both model'? Please check the sentence.

[Response] Sorry, it was a typo (although we do not know why it appeared). We have deleted them.

Fig. 2 caption: Add the description of the white scatter plots (MLD) in the legend. Change ': : :at station S1. (f-i) The same for station K2'
[Response] Description of MLD added. The figures indices also corrected.

Fig. 2: Check x-axis thick labels (subplots b, c, f and g).
[Response] Sorry, we did not find problems with x-axis thick labels.

Fig. 3: Add x-axis labels.
[Response] Added.

Results
In general, there is some discussion points that are included in the results section and that should rather be discussed in section 4.

P12 L18-25: This section describes the physical forcing and does not concern a result of the simulation. I would suggest to move this part in section 2.3 (method).
[Response] Yes, it is true. But we think that acquiring physics forcing is also an important component of our modeling work and the physics background should be counted as results although they are not direct results from simulation. So we feel it should better to be put in the results instead of in the method.

P12 L23:
'with the model estimates of MLD consistent with those measured from in situ temperature and salinity profiles': it is not clear what you are comparing exactly. (Please also add a reference to the figure showing that. What are white scatter plots on fig. 2 b and f?).
[Response] As explained above, we were comparing the MLDs from CTD profiles of temp. and salinity with from modeled profiles of Kv. We have added a reference to the figure ("Fig. 2b,f,j") showing this and also descriptions of white scatter boxes in the figure caption.

Fig. 4 caption: Remove the 's' in 'log-likelihood'. Replace (b-j) by (b-i)
[Response] Removed.

P13 L6: The SSqE of the smallest size fraction (fig. 4, q) also increases with time at S1

[Response]  Yes. But in the new simulation results, SSqE of the smallest size fraction decreases with time at S1.

P13 L9: The figure 5 is not commented in the text. Please add a sentence to describe the trend.

[Response] Added in the second paragraph of Sect. 3.2.

P13 L 11-16: The discussion on the value of the trait diffusion parameter should appear in the discussion section.

[Response] We feel that this interpretation on the value of the trait diffusion parameter should immediately follow the results to make the logic smooth and so that readers will easily be able to understand. We have changed the sentence to "The optimized $u$ value was much higher than in Acevedo-Trejos et al. (2016). Reducing $u$ to 0.05 yielded worse fits to the size-fractionated chlorophyll since lower size variance failed to capture the observed size scatter. It also relates to the limitation of the model that has to assume a lognormal distribution of size (*see* Sect. 4.2.1). However, an abnormally high $u$ could drive the model to unstable conditions in which the size variance kept increasing.".  In the Discussion section, we did not present details concerning the parameter values.

Fig. 6 and 8 captions: Complete the caption with the position of the different variables.

[Response] Completed.

P13 L20: 'the higher surface concentrations'

[Response] Yes. Changed to "the higher surface concentrations of DIN during winter than summer".

P13 L22-24: Isn't it in apparent contradiction with the fact that you argued that the modeled MLD is in agreement with observational data (page 12, line 23)?

[Response] Yes, this disappears with the new simulation results.

P13 L22-24: discussion

[Response] This part has been removed during revision.

P14 L5: The observational data on the size fractions are relatively noisy. Could you please provide more details on how these data were obtained (sampling methods, size measurments, sampling frequency) in section 2.5? At station K2, the size distribution in unclear in data and the model overestimate the proportion of 3-10 _m size class. At station S1, observational data show the dominance of smaller cells but do not show the vertical structure of the size distribution that is simulated by the model with smaller cells at the surface. Please add a comment on this.

[Response] We have added a description of how the data of size fractions were obtained. The data did show some seasonal and vertical patterns, and we have described the patterns of the data and comparisons between data and model in Sect. 3.3.

P14 L7-9: 'At station S1'. Do you mean station 'K2'?
[Response] We meant at S1. However, we have removed this sentence because there was indeed some vertical pattern at S1 if one looks really carefully.

P14 L10: discussion
[Response]  We have moved this sentence to Sect. 3.2 because we feel that the comment of light limitation on large cell size should immediately follow the description of optimized $\alpha_I$.

P14 L19: 'following stratification which occurs earlier in S1 than K2'
[Response] Yes, we have added 'which occurs earlier in S1 than K2'.

P14 L24: Show a figure of Chl/C ratio
[Response] The figure of Chl:C ratio shown in Fig. 10 and 11.

Fig. 10, g: High growth rate at the surface at K2 despite low TIN and low Chl a concentrations?
[Response] With the new simulation results, high Chl and growth rate occur at the same timing at surface at K2.

P15 L5-6: discussion
[Response] This sentence has been removed during revision.

P15 L24 – P16 L4: discussion

[Response] We have either moved the text to discussion or deleted it during revision.

P16 L3: The 'dynamic equilibrium theory proposed by Huston is only briefly mentionned (see also page 17, l, 1-2). This hypothesis implies that, under non-equilibrium conditions, the outcomes of the competition depend on the timescale of the competitive displacement and the relative rate of change in competitive abilities of each competing species. This point should be further developed and discussed according to your results on diversity in the discussion part of the manuscript.

[Response] This is a good point. We have added discussions on the 'dynamic equilibrium theory' in Sect. 4.1.1:

"Using 'adaptive dynamics', it is easier to quantify competition intensity (and other ecological quantities), which makes it easier to test ecological theories such as Huston's "general hypothesis of species diversity" (Huston, 1979). For example, the absolute magnitude of $\frac{d^2\mu(l)}{dl^2}$ correlates positively with $\mu$ (Fig. 13), indicating that higher growth rates induced greater resource competition. This agrees well with the "dynamic equilibrium theory". Huston (1979) emphasized that in natural environments where equilibrium is rarely achieved, growth rates play a greater role in determining diversity than do steady state competitive abilities as typically quantified by R* values (Tilman, 1982; Litchman et al., 2007). This is because when environmental conditions favour fast growth, it takes less time for the dominant species to predominate, and diversity decreases. The positive correlation between the absolute value of $\frac{d^2\mu(l)}{dl^2}$ and $\mu$ is a mathematical manifestation of the verbal argument in Huston (1979). "

P16 L6-9: As I said above, the role of the trait diffusion in maintaining diversity and the way it is used in the model is a bit tricky to understand. You could perhaps add a paragraph in the method section to clarify this point which is then only discussed briefly page 17, lines 10-14.

[Response] We have expanded the explanations on trait diffusion in Sect. 2.2:

"$u$ is the trait diffusion parameter, which describes the probability of the parental size $l(i)$ changing to adjacent size values $l(i–1)$ or $l(i+1)$ in offspring cells (Merico et al., 2014)."

Also in Sect. 3.4:

"Second, the contributions from the second derivatives of growth and trait diffusion (dominated by $2u\mu(l)$ with the contributions from $\frac{d^4\mu(l)}{dl^4}$ being minor; Eq. 7c) were the two

largest terms, which usually offset against each other. The values of $\frac{d^2\mu(l)}{dl^2}$ were always negative in all times at both stations, suggesting that without "trait diffusion", size variance would decrease toward zero (Eq. 7c). This highlights the importance of trait diffusion (which can be interpreted as genetic mutation or transgenerational phenotypic plasticity) to sustain diversity. The values of $\frac{d^2\mu(l)}{dl^2}$ were more negative when growth rates were higher and it is the margin of these two terms that (partially) drove the changes of size variance. For example, in early April of S1, the decrease of size variance was induced by a more negative $\frac{d^2\mu(l)}{dl^2}$ (*see* also Fig. 11h). Similar situations also occurred at the end of December."

also in Sect. 4.1.1:

"The incorporation of trait diffusion originally developed for continuous trait-based models (Merico et al., 2014) also provides a mechanism similar to speciation (or mutation) for sustaining diversity, linking ecological and evolutionary processes (Rosenzweig, 1995). The increasing effect of trait diffusion with growth rate is consistent with the Metabolic Theory of Ecology that metabolic rates, closely coupled with growth rates and generation time, are expected to correlate with mutation rates and affect speciation (Allen et al., 2006; Dowle et al., 2013). Our results have shown that it can be the largest term to balance competitive exclusion (Fig. 13). Without considering this mechanism, diversity could be underestimated in productive waters due to strong competition. This could also contribute to the latitudinal diversity gradient since in tropical regions (ectothermic) organisms tend to growth fast (i.e. short generation time) due to high temperature and therefore have high mutation and speciation rates (Rohde, 1992; Allen et al., 2006; Dowle et al., 2013). "

Discussion

P17 L22-27: There is no figures showing the N:C and Chl: C ratio patterns at the two stations. [Response] We have added the figures showing the N:C and Chl: C ratios at the two stations in Fig. 10 and 11.

P18 L3: 'Instead, we employ : : :': The word 'instead' seems inappropriate as you mention a very different process than in the previous sentence: the trade-offs between maximal growth rate and nutrient affinity in the phytoplankton is not related to the size-dependence of the grazing by predators.

[Response] Here actually we meant different mechanisms to control phytoplankton mean size. These two mechanisms are the two most plausible mechanisms (one from bottom-up and the other from top-down) that may affect phytoplankton size structure. We have modified the text to :

"We have provided both bottom-up and top-down mechanisms to affect the size structure of phytoplankton in **CITRATE 1.0**. First, we employ an observation-based unimodal relationship between maximal growth rate and size to give the nanophytoplankton the advantage under nutrient-replete conditions (Chen and Liu, 2010, 2011; Marañón et al., 2013), thus allowing a trade-off between nutrient affinity and maximal growth rate within the pico- and nano-size range."

As you mention l. 17-19, the role of grazing in shaping phytoplankton community has been shown to be crucial. In order to take into account a various palatability of phytoplankton for zooplankton feeding according to the cell size, the model should ideally involved a larger number of predator size classes (and/or, at least, an additional mesozooplankton size class) which would lead to much more complexity in the model. In addition to the predator-prey size-ratio, the predators' feeding mode (Mariani et al., 2013) and the formulation that is used to constrain the herbivorous impact on primary producers community composition are also very important. Please add the information on what kind of predation function you are using in your model in 'Model description'. Also, this points should be mentionned in the discussion section (add other references such as Anderson et al., 2010 ; Prowe et al., 2012).

[Response] We admit that zooplankton feeding including predator-prey size ratio and feeding mode is indeed an important process shaping phytoplankton size structure. We have added a mesozooplankton compartment to allow more subtle effects on phytoplankton size structure and later development of more sophisticated models. However, our parameterizations do not allow zooplankton grazing to play a significant role in affecting phytoplankton size structure because 1) we do not have sufficient data to constrain the zooplankton parameters and 2) following the principle of Occam's Razor, if bottom-up factors alone can well simulate the patterns of phytoplankton size structure (we feel the bottom-up factors play the dominant role in oligotrophic oceans while top-down factors might be important in coastal waters). Therefore, we feel it unnecessary to add more complications at this time.

We have given the grazing function (Holling-Type III) in the section of 'Model description'. But as we have argued, it is not necessary for our model to contain too many details of zooplankton grazing such as feeding mode without sufficient data.

P20 L12: 'other mecanisms such as vertical migration': I am not sure that vertical migration is a very common process in small phytoplankton populations that are found at the surface of subtropical waters during the summer. What about the nutrient limitation terms? What should be the half-saturation constants for nitrogen/phosphorus uptake in the 1-3 _m size class found in observationnal data?

[Response] Vertical migration is certainly not a common process in small phytoplankton populations. However, some have found that very large phytoplankton that can perform vertical migration in the subtropical oceans (e.g. Villareal et al. Nature 1999; Villareal J. Phycol. 2004). It has been claimed that this vertical migration might be significant for new production and it seems a fair mechanism to provide nutrients to the surface waters.

For the nutrient terms, first we need to clarify that the half-saturation constants for GROWTH ($K_m$) used in the Monod equation should be much smaller than those for nutrient uptake ($K_s$; Laws Ann. Rev. Mar. Sci. 2013). Experimental measurements for $K_m$ are scarce, but generally have very low values, much lower than those used in the model (Laws et al. J. Phycol. 2011). However, we do not believe that using a very small $K_m$ will solve the problem because phytoplankton growth has to be limited by nutrient, and in any case the growth rate in surface waters will be lower than at depth closer to the nutricline. In any case, we do not know any measurements on half-saturation constants of 1-3 μm phytoplankton at our stations.

We hope that our above revisions and discussions could be accepted by the reviewers. Thank you very much.

Best regards,
Bingzhang Chen
S. Lan Smith

[revised manuscript text omitted]

Chen Bingzhang 2017/10/23 4:52 PM

| Page 6: [1] Formatted | Chen Bingzhang | 17/8/5 11:09 PM |

Font:Italic

| Page 6: [2] Deleted | Bingzhang Chen | 17/9/29 11:11 AM |

The constructed model structure and optimized model parameters are to be used for later application of the "adaptive dynamics" approach in three-dimensional ocean general circulation models (GCMs).

| Page 6: [2] Deleted | Bingzhang Chen | 17/9/29 11:11 AM |

| Page 6: [3] Formatted | Chen Bingzhang | 17/8/7 10:51 AM |

List Paragraph, Numbered + Level: 1 + Numbering Style: 1, 2, 3, ... + Start at: 1 + Alignment: Left + Aligned at: 0 cm + Indent at: 0.85 cm

| Page 6: [4] Formatted | Bingzhang Chen | 17/9/29 11:11 AM |

Font:Italic

| Page 6: [4] Formatted | Bingzhang Chen | 17/9/29 11:11 AM |

Font:Italic

| Page 6: [5] Formatted | Chen Bingzhang | 17/8/7 10:51 AM |

Font:Bold, English (US)

[revised manuscript text omitted]

Font:Italic

| Page 11: [32] Formatted | Chen Bingzhang | 17/8/22 11:42 PM |
|---|---|---|

Font:Italic

| Page 11: [32] Formatted | Chen Bingzhang | 17/8/22 11:42 PM |
|---|---|---|

Font:Italic

| Page 11: [33] Deleted | Chen Bingzhang | 17/8/3 5:01 PM |
|---|---|---|

, $K_v$ is the vertical eddy diffusivity ($m^2 s^{-1}$)

| Page 11: [33] Deleted | Chen Bingzhang | 17/8/3 5:01 PM |
|---|---|---|

, $K_v$ is the vertical eddy diffusivity ($m^2 s^{-1}$)

| Page 11: [34] Formatted | Chen Bingzhang | 17/8/8 2:17 PM |
|---|---|---|

Superscript

| Page 11: [35] Formatted | Chen Bingzhang | 17/8/8 2:11 PM |
|---|---|---|

Indent: First line: 2.36 ch

| Page 11: [36] Deleted | Chen Bingzhang | 17/8/30 9:45 AM |

| Page 11: [36] Deleted | Chen Bingzhang | 17/8/30 9:45 AM |

| Page 11: [37] Deleted | Chen Bingzhang | 17/8/30 9:44 AM |

$P^2$

| Page 11: [37] Deleted | Chen Bingzhang | 17/8/30 9:44 AM |

$P^2$

| Page 11: [37] Deleted | Chen Bingzhang | 17/8/30 9:44 AM |

$P^2$

| Page 11: [37] Deleted | Chen Bingzhang | 17/8/30 9:44 AM |

$P^2$

| Page 11: [37] Deleted | Chen Bingzhang | 17/8/30 9:44 AM |

$P^2$

| Page 11: [37] Deleted | Chen Bingzhang | 17/8/30 9:44 AM |

$P^2$

| Page 11: [37] Deleted | Chen Bingzhang | 17/8/30 9:44 AM |

$P^2$

| Page 11: [37] Deleted | Chen Bingzhang | 17/8/30 9:44 AM |

$P^2$

| Page 11: [37] Deleted | Chen Bingzhang | 17/8/30 9:44 AM |

$P^2$

| Page 11: [37] Deleted | Chen Bingzhang | 17/8/30 9:44 AM |

$P^2$

| Page 11: [38] Formatted | Chen Bingzhang | 17/8/3 5:02 PM |

Font:Italic

| Page 11: [39] Deleted | Chen Bingzhang | 17/8/3 5:11 PM |

dissolved inorganic nitrogen ($N$, $\mu$mol L$^{-1}$)

| Page 11: [39] Deleted | Chen Bingzhang | 17/8/3 5:11 PM |
|---|---|---|

dissolved inorganic nitrogen ($N$, $\mu$mol L$^{-1}$)

| Page 11: [40] Formatted | Chen Bingzhang | 17/8/3 5:12 PM |
|---|---|---|

Font:Bold, Italic

| Page 11: [41] Formatted | Bingzhang Chen | 17/9/28 12:15 PM |
|---|---|---|

English (US)

| Page 11: [42] Deleted | Chen Bingzhang | 17/8/30 9:57 AM |
|---|---|---|

| Page 11: [42] Deleted | Chen Bingzhang | 17/8/30 9:57 AM |
|---|---|---|

| Page 11: [43] Formatted | Bingzhang Chen | 17/9/28 12:15 PM |
|---|---|---|

English (US)

| Page 11: [44] Formatted | Chen Bingzhang | 17/8/22 11:52 PM |
|---|---|---|

Font:Bold, Italic

| Page 11: [45] Formatted | Bingzhang Chen | 17/9/29 11:30 AM |
|---|---|---|

English (US)

| Page 11: [45] Formatted | Bingzhang Chen | 17/9/29 11:30 AM |
|---|---|---|

English (US)

| Page 11: [45] Formatted | Bingzhang Chen | 17/9/29 11:30 AM |
|---|---|---|

English (US)

| Page 11: [45] Formatted | Bingzhang Chen | 17/9/29 11:30 AM |
|---|---|---|

English (US)

| Page 12: [46] Deleted | Chen Bingzhang | 17/8/22 11:53 PM |
|---|---|---|

are also able

| Page 12: [46] Deleted | Chen Bingzhang | 17/8/22 11:53 PM |
|---|---|---|

are also able

| Page 12: [46] Deleted | Chen Bingzhang | 17/8/22 11:53 PM |
|---|---|---|

are also able

| Page 12: [47] Formatted | Bingzhang Chen | 17/9/28 12:15 PM |
|---|---|---|

English (US)

| Page 12: [47] Formatted | Bingzhang Chen | 17/9/28 12:15 PM |
|---|---|---|

English (US)

| Page 12: [47] Formatted | Bingzhang Chen | 17/9/28 12:15 PM |
|---|---|---|

English (US)

| Page 12: [48] Formatted | Bingzhang Chen | 17/9/29 5:21 PM |
|---|---|---|

Font:Italic

| Page 12: [48] Formatted | Bingzhang Chen | 17/9/29 5:21 PM |
|---|---|---|

Font:Italic

| Page 12: [48] Formatted | Bingzhang Chen | 17/9/29 5:21 PM |
|---|---|---|

Font:Italic

| Page 12: [48] Formatted | Bingzhang Chen | 17/9/29 5:21 PM |
|---|---|---|

Font:Italic

| Page 12: [49] Formatted | Bingzhang Chen | 17/9/29 5:32 PM |
|---|---|---|

Font:Italic

| Page 12: [49] Formatted | Bingzhang Chen | 17/9/29 5:32 PM |
|---|---|---|

Font:Italic

| Page 12: [49] Formatted | Bingzhang Chen | 17/9/29 5:32 PM |
|---|---|---|

Font:Italic

| Page 12: [49] Formatted | Bingzhang Chen | 17/9/29 5:32 PM |
|---|---|---|

Font:Italic

| Page 12: [49] Formatted | Bingzhang Chen | 17/9/29 5:32 PM |
|---|---|---|

Font:Italic

| Page 12: [49] Formatted | Bingzhang Chen | 17/9/29 5:32 PM |
|---|---|---|

Font:Italic

| Page 12: [49] Formatted | Bingzhang Chen | 17/9/29 5:32 PM |
|---|---|---|

Font:Italic

[revised manuscript text omitted]

Normal

| Page 13: [54] Formatted | Chen Bingzhang | 17/8/22 11:12 PM |
|---|---|---|

Font:Bold, Italic

| Page 13: [55] Formatted | Chen Bingzhang | 17/8/8 1:44 PM |
|---|---|---|

Normal, Right

| Page 13: [56] Formatted | Chen Bingzhang | 17/8/8 1:45 PM |
|---|---|---|

Normal

| Page 13: [57] Formatted | Chen Bingzhang | 17/8/8 2:44 PM |
|---|---|---|

Font:Italic

| Page 13: [57] Formatted | Chen Bingzhang | 17/8/8 2:44 PM |
|---|---|---|

Font:Italic

| Page 13: [57] Formatted | Chen Bingzhang | 17/8/8 2:44 PM |
|---|---|---|

Font:Italic

| Page 13: [57] Formatted | Chen Bingzhang | 17/8/8 2:44 PM |
|---|---|---|

Font:Italic

| Page 13: [57] Formatted | Chen Bingzhang | 17/8/8 2:44 PM |
|---|---|---|

Font:Italic

| Page 13: [57] Formatted | Chen Bingzhang | 17/8/8 2:44 PM |
|---|---|---|

Font:Italic

| Page 13: [57] Formatted | Chen Bingzhang | 17/8/8 2:44 PM |
|---|---|---|

Font:Italic

| Page 13: [57] Formatted | Chen Bingzhang | 17/8/8 2:44 PM |
|---|---|---|

Font:Italic

| Page 13: [58] Formatted | Bingzhang Chen | 17/9/28 12:15 PM |
|---|---|---|

English (US)

| Page 13: [59] Formatted | Chen Bingzhang | 17/8/23 12:05 AM |

Font:Italic

| Page 13: [60] Formatted | Bingzhang Chen | 17/9/28 12:15 PM |

English (US)

| Page 13: [60] Formatted | Bingzhang Chen | 17/9/28 12:15 PM |

English (US)

| Page 13: [61] Formatted | Chen Bingzhang | 17/8/23 12:06 AM |

Font:Italic

| Page 13: [62] Formatted | Bingzhang Chen | 17/9/28 12:15 PM |

Font:Italic, English (US)

| Page 13: [62] Formatted | Bingzhang Chen | 17/9/28 12:15 PM |

Font:Italic, English (US)

| Page 13: [62] Formatted | Bingzhang Chen | 17/9/28 12:15 PM |

Font:Italic, English (US)

| Page 13: [63] Formatted | Chen Bingzhang | 17/8/8 2:05 PM |

Normal, Right

| Page 13: [64] Formatted | Chen Bingzhang | 17/8/8 2:07 PM |

Normal

| Page 13: [65] Formatted | Chen Bingzhang | 17/8/8 2:34 PM |

Font:Italic

| Page 13: [65] Formatted | Chen Bingzhang | 17/8/8 2:34 PM |

Font:Italic

| Page 13: [65] Formatted | Chen Bingzhang | 17/8/8 2:34 PM |

Font:Italic

| Page 13: [66] Formatted | Chen Bingzhang | 17/8/8 2:45 PM |

Normal, Right

| Page 13: [67] Formatted | Chen Bingzhang | 17/8/8 2:07 PM |

Normal

| Page 13: [68] Formatted | Chen Bingzhang | 17/8/8 2:46 PM |

Font:Italic

| Page 13: [68] Formatted | Chen Bingzhang | 17/8/8 2:46 PM |
|---|---|---|

Font:Italic

| Page 13: [68] Formatted | Chen Bingzhang | 17/8/8 2:46 PM |
|---|---|---|

Font:Italic

| Page 13: [69] Formatted | Chen Bingzhang | 17/8/8 2:49 PM |
|---|---|---|

Normal, Right

| Page 13: [70] Formatted | Chen Bingzhang | 17/8/8 2:50 PM |
|---|---|---|

Normal

| Page 13: [71] Formatted | Bingzhang Chen | 17/9/28 12:16 PM |
|---|---|---|

English (US)

| Page 13: [72] Formatted | Chen Bingzhang | 17/10/23 9:25 AM |
|---|---|---|

Normal, Centered

| Page 13: [73] Formatted | Chen Bingzhang | 17/8/8 2:50 PM |
|---|---|---|

Normal

| Page 13: [74] Formatted | Chen Bingzhang | 17/8/23 12:25 AM |
|---|---|---|

Normal, Centered

| Page 13: [75] Formatted | Chen Bingzhang | 17/8/8 2:50 PM |
|---|---|---|

Normal

| Page 13: [76] Formatted | Chen Bingzhang | 17/8/8 2:55 PM |
|---|---|---|

Font:Italic

| Page 13: [76] Formatted | Chen Bingzhang | 17/8/8 2:55 PM |
|---|---|---|

Font:Italic

| Page 13: [77] Formatted | Chen Bingzhang | 17/8/23 12:12 AM |
|---|---|---|

Normal, Right

| Page 13: [78] Formatted | Chen Bingzhang | 17/8/23 12:12 AM |
|---|---|---|

Font:Bold

| Page 13: [79] Formatted | Chen Bingzhang | 17/8/8 5:33 PM |
|---|---|---|

Normal

| Page 13: [80] Formatted | Chen Bingzhang | 17/8/8 5:38 PM |
|---|---|---|

Normal, Right

| Page 15: [81] Deleted | S. Lan Smith | 17/10/22 10:35 PM |

is used to generate a new set of parameters based on current position of the parameters.

| Page 16: [82] Deleted | S. Lan Smith | 17/10/22 10:40 PM |

, developed by

| Page 16: [82] Deleted | S. Lan Smith | 17/10/22 10:40 PM |

, developed by

| Page 16: [82] Deleted | S. Lan Smith | 17/10/22 10:40 PM |

, developed by

| Page 16: [82] Deleted | S. Lan Smith | 17/10/22 10:40 PM |

, developed by

| Page 16: [82] Deleted | S. Lan Smith | 17/10/22 10:40 PM |

, developed by

| Page 16: [83] Deleted | S. Lan Smith | 17/10/22 10:44 PM |

usually the main cause for the

| Page 16: [83] Deleted | S. Lan Smith | 17/10/22 10:44 PM |

usually the main cause for the

| Page 16: [83] Deleted | S. Lan Smith | 17/10/22 10:44 PM |

usually the main cause for the

| Page 16: [83] Deleted | S. Lan Smith | 17/10/22 10:44 PM |

usually the main cause for the

| Page 16: [83] Deleted | S. Lan Smith | 17/10/22 10:44 PM |

usually the main cause for the

| Page 16: [83] Deleted | S. Lan Smith | 17/10/22 10:44 PM |

usually the main cause for the

| Page 16: [83] Deleted | S. Lan Smith | 17/10/22 10:44 PM |

usually the main cause for the

| Page 16: [84] Deleted | S. Lan Smith | 17/10/22 10:46 PM |

DRAM

| Page 16: [84] Deleted | S. Lan Smith | 17/10/22 10:46 PM |

DRAM

| Page 16: [84] Deleted | S. Lan Smith | 17/10/22 10:46 PM |
|---|---|---|

DRAM

| Page 16: [84] Deleted | S. Lan Smith | 17/10/22 10:46 PM |
|---|---|---|

DRAM

| Page 16: [85] Deleted | S. Lan Smith | 17/10/22 10:48 PM |
|---|---|---|

workflow of the

| Page 16: [85] Deleted | S. Lan Smith | 17/10/22 10:48 PM |
|---|---|---|

workflow of the

| Page 16: [85] Deleted | S. Lan Smith | 17/10/22 10:48 PM |
|---|---|---|

workflow of the

| Page 16: [86] Deleted | S. Lan Smith | 17/10/22 10:49 PM |
|---|---|---|

. The initial $P_{cvm}$ is constructed to have

1)

| Page 16: [86] Deleted | S. Lan Smith | 17/10/22 10:49 PM |
|---|---|---|

. The initial $P_{cvm}$ is constructed to have

2)

| Page 16: [86] Deleted | S. Lan Smith | 17/10/22 10:49 PM |
|---|---|---|

. The initial $P_{cvm}$ is constructed to have

3)

| Page 16: [86] Deleted | S. Lan Smith | 17/10/22 10:49 PM |
|---|---|---|

. The initial $P_{cvm}$ is constructed to have

4)

| Page 16: [86] Deleted | S. Lan Smith | 17/10/22 10:49 PM |
|---|---|---|

. The initial $P_{cvm}$ is constructed to have

5)

| Page 17: [87] Formatted | Chen Bingzhang | 17/8/27 10:00 PM |
|---|---|---|

Font:Italic

| Page 17: [87] Formatted | Chen Bingzhang | 17/8/27 10:00 PM |
|---|---|---|

Font:Italic

| Page 17: [87] Formatted | Chen Bingzhang | 17/8/27 10:00 PM |
|---|---|---|

Font:Italic

| Page 17: [87] Formatted | Chen Bingzhang | 17/8/27 10:00 PM |
|---|---|---|

Font:Italic

| Page 17: [88] Formatted | Chen Bingzhang | 17/8/27 10:00 PM |
|---|---|---|

Subscript

| Page 17: [88] Formatted | Chen Bingzhang | 17/8/27 10:00 PM |
|---|---|---|

Subscript

| Page 17: [89] Formatted | Chen Bingzhang | 17/8/27 10:03 PM |
|---|---|---|

Font:Italic

| Page 17: [90] Formatted | Chen Bingzhang | 17/8/27 10:07 PM |
|---|---|---|

Font:Not Italic

| Page 17: [90] Formatted | Chen Bingzhang | 17/8/27 10:07 PM |
|---|---|---|

Font:Not Italic

| Page 17: [91] Formatted | Chen Bingzhang | 17/8/27 11:10 PM |
|---|---|---|

Font:Italic

| Page 17: [91] Formatted | Chen Bingzhang | 17/8/27 11:10 PM |
|---|---|---|

Font:Italic

| Page 17: [91] Formatted | Chen Bingzhang | 17/8/27 11:10 PM |
|---|---|---|

Font:Italic

| Page 17: [91] Formatted | Chen Bingzhang | 17/8/27 11:10 PM |
|---|---|---|

Font:Italic

| Page 17: [91] Formatted | Chen Bingzhang | 17/8/27 11:10 PM |
|---|---|---|

Font:Italic

| Page 17: [91] Formatted | Chen Bingzhang | 17/8/27 11:10 PM |
|---|---|---|

Font:Italic

| Page 17: [91] Formatted | Chen Bingzhang | 17/8/27 11:10 PM |
|---|---|---|

Font:Italic

| Page 17: [92] Formatted | Chen Bingzhang | 17/8/27 9:42 PM |
|---|---|---|

List Paragraph, Indent: Left:  0 cm, Hanging:  4.26 ch, Numbered + Level: 1 + Numbering Style: 1, 2, 3, ... + Start at: 1 + Alignment: Left + Aligned at:  0.75 cm + Indent at:  1.98 cm

| Page 17: [93] Formatted | Bingzhang Chen | 17/9/28 12:15 PM |
|---|---|---|

English (US)

| Page 17: [94] Formatted | Chen Bingzhang | 17/8/27 11:15 PM |
|---|---|---|

English (US)

| Page 17: [95] Formatted | Chen Bingzhang | 17/8/27 11:15 PM |
|---|---|---|

Indent: First line: 1.77 ch

| Page 17: [96] Deleted | S. Lan Smith | 17/10/22 10:52 PM |
|---|---|---|

um

| Page 17: [96] Deleted | S. Lan Smith | 17/10/22 10:52 PM |
|---|---|---|

um

| Page 17: [97] Formatted | Bingzhang Chen | 17/9/28 12:15 PM |
|---|---|---|

English (US)

| Page 17: [98] Formatted | Chen Bingzhang | 17/8/27 11:17 PM |
|---|---|---|

Font:Italic

| Page 17: [99] Formatted | Bingzhang Chen | 17/9/28 12:15 PM |
|---|---|---|

English (US)

| Page 17: [100] Formatted | Bingzhang Chen | 17/9/28 12:15 PM |
|---|---|---|

English (US)

| Page 17: [101] Formatted | Bingzhang Chen | 17/9/28 12:15 PM |
|---|---|---|

English (US)

| Page 17: [101] Formatted | Bingzhang Chen | 17/9/28 12:15 PM |
|---|---|---|

English (US)

[revised manuscript text omitted]

English (US)

| Page 30: [164] Deleted | Bingzhang Chen | 17/9/28 5:34 PM |
|---|---|---|

certainly

| Page 30: [165] Formatted | Bingzhang Chen | 17/9/28 10:31 PM |
|---|---|---|

Font:Not Bold, Italic

| Page 30: [165] Formatted | Bingzhang Chen | 17/9/28 10:31 PM |

Font:Not Bold, Italic

| Page 30: [165] Formatted | Bingzhang Chen | 17/9/28 10:31 PM |

Font:Not Bold, Italic

| Page 30: [166] Deleted | S. Lan Smith | 17/10/22 11:51 PM |

From t

| Page 30: [166] Deleted | S. Lan Smith | 17/10/22 11:51 PM |

From t

| Page 30: [166] Deleted | S. Lan Smith | 17/10/22 11:51 PM |

From t

| Page 30: [166] Deleted | S. Lan Smith | 17/10/22 11:51 PM |

From t

| Page 30: [166] Deleted | S. Lan Smith | 17/10/22 11:51 PM |

From t

| Page 30: [166] Deleted | S. Lan Smith | 17/10/22 11:51 PM |

From t

| Page 30: [166] Deleted | S. Lan Smith | 17/10/22 11:51 PM |

From t

| Page 30: [166] Deleted | S. Lan Smith | 17/10/22 11:51 PM |

From t

| Page 30: [166] Deleted | S. Lan Smith | 17/10/22 11:51 PM |

From t

| Page 30: [166] Deleted | S. Lan Smith | 17/10/22 11:51 PM |

From t

| Page 30: [166] Deleted | S. Lan Smith | 17/10/22 11:51 PM |

From t

| Page 30: [167] Deleted | Bingzhang Chen | 17/9/28 5:35 PM |

$v$) should be similar due to strong mixing, the approximation is not a big concern. The problem mainly lies at the bottom of the mixed layer where there is a sharp gradient of phytoplankton properties particularly biomass (Fig. 12c). We can roughly estimate that,

based on the eddy diffusivity ($\sim 2 \times 10^5$ m$^2$ s$^{-1}$), grid distance (20 m), and the phytoplankton biomass differences (0.2 $\mu$mol N L$^{-1}$), the transported phytoplankton biomass from the upper grid (higher $P$) to the lower grid (lower $P$) at one time step (0.5 min) is roughly 0.0002 $\mu$mol N L$^{-1}$, 1% of the $P$ in the lower grid. Therefore, oOur tentative conclusion is that

| Page 30: [168] Formatted | Chen Bingzhang | 17/8/7 4:50 PM |
|---|---|---|

Font:Not Bold

| Page 30: [168] Formatted | Chen Bingzhang | 17/8/7 4:50 PM |
|---|---|---|

Font:Not Bold

| Page 30: [168] Formatted | Chen Bingzhang | 17/8/7 4:50 PM |
|---|---|---|

Font:Not Bold

| Page 30: [168] Formatted | Chen Bingzhang | 17/8/7 4:50 PM |
|---|---|---|

Font:Not Bold

| Page 30: [168] Formatted | Chen Bingzhang | 17/8/7 4:50 PM |
|---|---|---|

Font:Not Bold

| Page 30: [168] Formatted | Chen Bingzhang | 17/8/7 4:50 PM |
|---|---|---|

Font:Not Bold

| Page 30: [168] Formatted | Chen Bingzhang | 17/8/7 4:50 PM |
|---|---|---|

Font:Not Bold

| Page 30: [168] Formatted | Chen Bingzhang | 17/8/7 4:50 PM |
|---|---|---|

Font:Not Bold

| Page 30: [168] Formatted | Chen Bingzhang | 17/8/7 4:50 PM |
|---|---|---|

Font:Not Bold

| Page 30: [168] Formatted | Chen Bingzhang | 17/8/7 4:50 PM |
|---|---|---|

Font:Not Bold

| Page 30: [168] Formatted | Chen Bingzhang | 17/8/7 4:50 PM |
|---|---|---|

Font:Not Bold

| Page 30: [168] Formatted | Chen Bingzhang | 17/8/7 4:50 PM |
|---|---|---|

Font:Not Bold

[revised manuscript text omitted]

d

| Page 37: [184] Deleted | Chen Bingzhang | 17/9/13 3:36 PM |
|---|---|---|

d

| Page 37: [185] Deleted | Chen Bingzhang | 17/9/14 2:56 PM |
|---|---|---|

1.34

| Page 37: [185] Deleted | Chen Bingzhang | 17/9/14 2:56 PM |
|---|---|---|

1.34

| Page 37: [186] Deleted | Chen Bingzhang | 17/9/13 2:36 PM |
|---|---|---|

$\alpha_\mu$        First-order size scaling component for $\mu_\mathrm{m}$        0.25[e] (0.1, 0.4)      0.27 (0.005)              dimensionle

$\beta_\mu$        Second-order size scaling component for $\mu_\mathrm{m}$      −0.025[e] (−0.05, 0)    −0.013 (0.0002)          dimensionle

| Page 37: [187] Deleted | Chen Bingzhang | 17/9/13 3:36 PM |
|---|---|---|

f

| Page 37: [187] Deleted | Chen Bingzhang | 17/9/13 3:36 PM |
|---|---|---|

f

| Page 37: [188] Deleted | Chen Bingzhang | 17/9/22 11:42 AM |
|---|---|---|

058

**Page 37: [188] Deleted**     **Chen Bingzhang**     **17/9/22 11:42 AM**

058

**Page 37: [188] Deleted**     **Chen Bingzhang**     **17/9/22 11:42 AM**

058

**Page 37: [189] Deleted**     **Chen Bingzhang**     **17/9/13 2:36 PM**

$\alpha_{fer}$     Size scaling exponent for $K_{0,fer}$     $0.27^c$ (0.1, 0.3)     0.30 (0.001)     dimensionle

**Page 37: [190] Deleted**     **Chen Bingzhang**     **17/9/13 3:36 PM**

g

**Page 37: [190] Deleted**     **Chen Bingzhang**     **17/9/13 3:36 PM**

g

**Page 37: [190] Deleted**     **Chen Bingzhang**     **17/9/13 3:36 PM**

g

**Page 37: [191] Deleted**     **Chen Bingzhang**     **17/9/13 3:25 PM**

092

**Page 37: [191] Deleted**     **Chen Bingzhang**     **17/9/13 3:25 PM**

092

**Page 37: [191] Deleted**     **Chen Bingzhang**     **17/9/13 3:25 PM**

092

**Page 37: [192] Deleted**     **Chen Bingzhang**     **17/8/7 2:58 PM**

*dustso*     Dust iron solubility     $0.02^h$ (0.01, 0.05)     0.022 (0.0007)     dimensionle

**Page 37: [193] Formatted**     **Bingzhang Chen**     **17/9/28 12:16 PM**

Spanish

**Page 37: [194] Deleted**     **Chen Bingzhang**     **17/9/13 3:34 PM**

Marañón et al., (2013); [b]

**Page 37: [194] Deleted**     **Chen Bingzhang**     **17/9/13 3:34 PM**

Marañón et al., (2013); [b]

| Page 39: [195] Formatted | Chen Bingzhang | 17/9/20 4:03 PM |

Font:Not Bold

| Page 39: [196] Formatted | Chen Bingzhang | 17/9/20 4:04 PM |

Line spacing:  double

| Page 39: [197] Formatted | Chen Bingzhang | 17/9/20 4:03 PM |

Font:Not Bold

| Page 39: [198] Formatted | Bingzhang Chen | 17/9/28 11:18 PM |

Font:10.5 pt

| Page 39: [199] Formatted | Bingzhang Chen | 17/9/29 12:05 PM |

Space Before:  0 pt

| Page 39: [200] Formatted Table | Bingzhang Chen | 17/9/29 12:05 PM |

Formatted Table

| Page 39: [201] Formatted | Bingzhang Chen | 17/9/28 11:18 PM |

Font:10.5 pt

| Page 39: [202] Formatted | Bingzhang Chen | 17/9/29 12:05 PM |

Space Before:  0 pt

| Page 39: [203] Formatted | Bingzhang Chen | 17/9/28 11:18 PM |

Font:10.5 pt

| Page 39: [204] Formatted | Bingzhang Chen | 17/9/29 12:05 PM |

Space Before:  0 pt

| Page 39: [205] Formatted | Bingzhang Chen | 17/9/29 12:05 PM |

Left, Space Before:  0 pt

| Page 39: [206] Formatted | Bingzhang Chen | 17/9/28 11:18 PM |

Font:10.5 pt

| Page 39: [207] Formatted | Bingzhang Chen | 17/9/29 12:05 PM |

Space Before:  0 pt

| Page 39: [208] Formatted | Bingzhang Chen | 17/9/28 11:18 PM |

Font:10.5 pt

| Page 39: [209] Formatted | Bingzhang Chen | 17/9/29 12:05 PM |

Space Before:  0 pt

| Page 39: [210] Formatted | Bingzhang Chen | 17/9/28 11:18 PM |

Font:10.5 pt

| Page 39: [211] Formatted | Bingzhang Chen | 17/9/29 12:05 PM |

Space Before:  0 pt

| Page 39: [212] Formatted | Bingzhang Chen | 17/9/28 11:18 PM |

Font:10.5 pt

| Page 39: [213] Formatted | Bingzhang Chen | 17/9/29 12:05 PM |

Space Before:  0 pt

| Page 39: [214] Formatted | Bingzhang Chen | 17/9/28 11:18 PM |

Font:10.5 pt

| Page 39: [215] Formatted | Bingzhang Chen | 17/9/29 12:05 PM |

Space Before:  0 pt

| Page 39: [216] Formatted | Bingzhang Chen | 17/9/28 11:18 PM |

Font:10.5 pt

| Page 39: [217] Formatted | Bingzhang Chen | 17/9/29 12:05 PM |

Space Before:  0 pt

| Page 39: [218] Formatted | Bingzhang Chen | 17/9/28 11:18 PM |

Font:10.5 pt

| Page 39: [219] Formatted | Bingzhang Chen | 17/9/29 12:05 PM |

Space Before:  0 pt

| Page 39: [220] Formatted | Bingzhang Chen | 17/9/28 11:18 PM |

Font:10.5 pt

| Page 39: [221] Formatted | Bingzhang Chen | 17/9/29 12:05 PM |

Space Before:  0 pt

| Page 39: [222] Formatted | Bingzhang Chen | 17/9/28 11:18 PM |

Font:10.5 pt

| Page 39: [223] Formatted | Bingzhang Chen | 17/9/29 12:06 PM |

Space Before:  6 pt

| Page 39: [224] Formatted | Bingzhang Chen | 17/9/29 12:08 PM |

Font:10.5 pt, (Asian) Japanese, (Other) English (US)

| Page 39: [225] Formatted | Bingzhang Chen | 17/9/28 11:18 PM |

Font:10.5 pt

| Page 39: [226] Formatted | Bingzhang Chen | 17/9/28 11:18 PM |

Font:10.5 pt

| Page 39: [227] Formatted | Bingzhang Chen | 17/9/29 12:06 PM |

Space Before:  6 pt

| Page 53: [228] Deleted | Bingzhang Chen | 17/9/28 11:32 PM |

[Figure]

[Figure]

Fig. 12

---

## Referee Report (RR1)

**Second Review of « CITRATE 1.0: Phytoplankton continuous trait-distribution model with one dimensional physical transport applied to the Northwest Pacific » by Bingzhang Chen and S. Lan Smith.**

I have read with interest the revised version of the manuscript by Bingzhang Chen and S. Lan Smith as well as their answers to the editor and the three referees' comments. The manuscript has been substantially improved. Indeed, the authors have extensively reworked some parts of the study which adds a significant value to the manuscript in terms of both methodological implications of their work and scientific outcome. In particular, new simulations have been conducted with a modified version of the model equations, including e.g. a second zooplankton variable corresponding to the mesozooplankton with grazing preferences scaled with prey size.

As suggested, and importantly, the method that has been conducted in this study, namely using the observations at two distinct stations to set up a single set of parameters values that can later be used as an initial estimate for new simulations (e.g. 3D simulations), has been clarified in the new version of the manuscript. Moreover, some efforts have also been made to validate the model and the previously obtained unique set of parameters using an independent station (ALOHA) with only limited success in reproducing the observed biogeochemical features in this region (understimation of Chl, NPP and PON). Some suggestions have been proposed to improve the optimization of the parameters for 3D GCM's simulations (Transport Matrix Technique).

The results section 3.4 has also been further developed with a detailed description of the relative weight of the different factors and their relationship in driving the simulated size variance (i.e diversity) and its variability over time at the two stations. Finally, the discussion section (4.1) has been deeply improved and better structured which contributes to a better connection between this work and more general concepts with regards to ecological mechanisms explaining plankton diversity (exclusive competition, evolutionary processes (trait diffusion) and physical transport, see section 4.1.1).

Overall, convincing and detailed arguments have been given by the authors on every points requested by the referees. Therefore, I recommend this revised manuscript for publication in GMD. Hereafter, I give a few remaining minor suggestions to improve the clarity of the manuscript.

P4 L.13-19 : *'The trait variance, treated as a tracer in the model, serves as a measure of trait diversity; although it cannot be simply equated to species richness, it can be converted to other diversity metrics such as the continuous entropy (Quintana et al., 2008). The diversity of functional traits is arguably a better diversity index than species richness relating to ecosystem functioning (Loreau et al., 2001).* **Thus***, the continuous trait-based model has the advantage that the factors controlling diversity can be directly quantified ...'*

Due to the inclusion of new sentences, text organization results in poor transitioning. I would suggest restructuring as follows:

*'The trait variance, treated as a tracer in the model, serves as a measure of trait diversity. **Thus, the continuous trait-based model has the advantage that the factors controlling diversity can be directly quantified … . Although the size variance** cannot be simply equated to species richness, it can be converted to other diversity metrics such as the continuous entropy (Quintana et al., 2008). **Moreover,** the diversity of functional traits is arguably a better diversity index than species richness relating to ecosystem functioning (Loreau et al., 2001).'*

P.5 L26-P.6 L4: Section 2: This added overview paragraph at the beginning of the model description section is very useful. However, the use of the term 'CITRATE 1.0' (P.4, L.26) is sometimes confusing as the reader might not know whether the authors are talking about the name of the model (which should be the correct use for CITRATE 1.0) or the method conducted in this study. Moreover, it would also be useful to mention here that the method that has been used aims at calibrating the model parameters to be applied for different oceanic regions. I would suggest something like:

*The aim of **the present study** is **to design and** implement **a continuous trait-based model (CITRATE 1.0) at two representative stations in the North Pacific. The overall goal of this model is not only to simulate** the phytoplankton size diversity but also to faithfully reproduce the seasonal and vertical dynamics of other important quantities such as nutrients Chl a, and productivity **in for later investigations of** the roles of phytoplankton diversity in biogeochemical cycles **in different oceanic regions (using 3D regional/global simulations). Therefore, the two contrasted stations were used to provide a single set of parameters values by fitting the model results to observations before the obtained model was validated in another independent station (ALOHA).** Hence, **CITRATE 1.0** consists of the following key features:*

P. 21 L. 6-10: For consistency reasons, this part on the comparison of estimated growth rate with literature values should be moved P. 20 L. 21 together with the paragraph on the test of sensitivity of the growth rate.

---

## Author Response (AR2)

Dear Dr. Yool,

We sincerely thank you and the two reviewers for the valuable comments, which have helped us improve the manuscript. We have considered carefully the comments and revised the manuscript accordingly. Please see below for detailed point-by-point replies (*in italics*) to the comments (in plain text). We hope that our revisions will be satisfactory to you and the reviewers.

Best regards,
Bingzhang Chen
S. Lan Smith
* * *
Reviewer 1

The lognormal distribution is a fundamental assumption in the authors' model. In their revised manuscript, they concede that it is also an unusual one. It is perfectly fine to say that the lognormal can be fit well to empirical data (e.g. Quintana et al. 2008, though another citation or two here would be nice) and has been used in other continuous size models and therefore is used here. However, the mathematical arguments used for why the lognormal was used instead of the power law are uncompelling and/or incorrect.

The authors say that the power-law distribution does not have an upper or lower limit on size, whereas the lognormal does; this is wrong. Power laws are supported (i.e. have non-zero probability density) on the range [x_min, infinity), whereas lognormals are supported on the range [0, infinity), so these distributions are not different in this regard. In fact, it is much more common to specify an upper cutoff on power laws via an exponential truncation than it is for a lognormal. It is quite easy to calculate the mean and variance of a power law, so these are bad examples of mathematical manipulations for which the power law is hard. The power law also has zero probability of negative size; actually if x_min for the power law can be set to the size of the

smallest phytoplankton in the system (e.g. Pro), which means that the power law has a zero probability for a cell smaller than x_min, while the lognormal always has a nonzero probability for arbitrarily small phytoplankton. The skewness of the log-size distribution discussed by the authors is further evidence for the power law potentially giving better results than the lognormal.

It's fine to justify the lognormal by citing that it can be fit to data well and has been used in continuous size models, but the other arguments for its use should be reconsidered, and the authors should be clear that their model hinges upon this unusual, even if justifiable, assumption. I recommend the authors read the paper "Power laws, Pareto distributions, and Zipf's law" by M. E. J. Newman, which gives a clear description of many of the relevant mathematical properties of power laws.

All other changes I found satisfactory!

*[Response]*
*First of all, we sincerely thank the reviewer for suggesting the paper of Newman (2005), which is extremely inspiring. Also thanks for suggesting the truncated power-law distribution, which could be a potential alternative to the lognormal distribution. We have looked into the literature and found that the controversy concerning the power law versus lognormal distribution has long existed in many fields of science.*

*We apologize that in the previous revised text, due to lack of statistics knowledge, we wrote incorrectly that the power-law distribution is unrealistic in representing phytoplankton at the size limits. We were also incorrect in stating that its mathematical manipulation is difficult. Now we have thoroughly revised Sect. 4.2.1. For the sake of space, we will not copy the whole text here, but just summarize the main points of Sect. 4.2.1:*

*1) The power law is indeed more often used to fit phytoplankton size data.*

*2)* *We need to impose an upper cutoff for power law to calculate mean and variance of phytoplankton size.*

*3)* *It is difficult to tell which distribution is better for fitting empirical data. The two distributions are intrinsically connected.*

*4)* *Neither of them can deal with the problem of multimodal distributions, which may be solved by having multiple functional groups, assuming a particular distribution for each group.*

\*\*\*\*\*\*\*\*\*\*\*\*\*\*\*\*\*\*\*\*\*\*\*\*\*\*\*\*\*\*\*\*\*\*\*\*\*\*\*\*\*\*\*\*\*\*\*\*\*\*\*\*\*\*\*\*\*\*\*\*\*\*\*\*\*\*\*\*\*\*\*\*\*\*\*\*\*\*\*\*\*\*\*\*\*\*\*\*\*\*\*\*\*\*

Reviewer 2

I have read with interest the revised version of the manuscript by Bingzhang Chen and S. Lan Smith as well as their answers to the editor and the three referees' comments. The manuscript has been substantially improved. Indeed, the authors have extensively reworked some parts of the study which adds a significant value to the manuscript in terms of both methodological implications of their work and scientific outcome. In particular, new simulations have been conducted with a modified version of the model equations, including e.g. a second zooplankton variable corresponding to the mesozooplankton with grazing preferences scaled with prey size.

As suggested, and importantly, the method that has been conducted in this study, namely using the observations at two distinct stations to set up a single set of parameters values that can later be used as an initial estimate for new simulations (e.g. 3D simulations), has been clarified in the new version of the manuscript. Moreover, some efforts have also been made to validate the model and the previously obtained unique set of parameters using an independent station (ALOHA) with only limited success in reproducing the observed biogeochemical features in this region (understimation of Chl, NPP and PON). Some suggestions have been proposed to improve the optimization of the parameters for 3D

GCM's simulations (Transport Matrix Technique).

The results section 3.4 has also been further developed with a detailed

description of the relative weight of the different factors and their relationship in driving the simulated size variance (i.e diversity) and its variability over time at the two stations. Finally, the discussion section (4.1) has been deeply improved and better structured which contributes to a better connection between this work and more general concepts with regards to ecological mechanisms explaining plankton diversity (exclusive competition, evolutionary processes (trait diffusion) and physical transport, see section 4.1.1).

Overall, convincing and detailed arguments have been given by the authors on every points requested by the referees. Therefore, I recommend this revised manuscript for publication in GMD. Hereafter, I give a few remaining minor suggestions to improve the clarity of the manuscript.

*[Response] Thank you very much for the encouragement!*

P4 L.13-19 : 'The trait variance, treated as a tracer in the model, serves as a measure of trait diversity; although it cannot be simply equated to species richness, it can be converted to other diversity metrics such as the continuous entropy (Quintana et al., 2008). The diversity of functional traits is arguably a better diversity index than species richness relating to ecosystem functioning (Loreau et al., 2001).

Thus, the continuous trait-based model has the advantage that the factors controlling diversity can be directly quantified …'

Due to the inclusion of new sentences, text organization results in poor transitioning. I would suggest restructuring as follows:

'The trait variance, treated as a tracer in the model, serves as a measure of trait diversity. Thus, the continuous trait-based model has the advantage that the factors controlling diversity can be directly quantified … . Although the size variance cannot be simply equated to species richness, it can be converted to other diversity metrics such as the continuous entropy (Quintana et al., 2008). Moreover, the diversity of functional traits is arguably a better diversity index than species richness relating to ecosystem functioning

(Loreau et al., 2001).'

*[Response] The suggested revisions are very nice. We have completely followed the suggestions.*

P.5 L26-P.6 L4: Section 2: This added overview paragraph at the beginning of the model description section is very useful. However, the use of the term 'CITRATE 1.0' (P.4, L.26) is sometimes confusing as the reader might not know whether the authors are talking about the name of the model (which should be the correct use for CITRATE 1.0) or the method conducted in this study. Moreover, it would also be useful to mention here that the method that has been used aims at calibrating the model parameters to be applied for different oceanic regions. I would suggest something like:

The aim of the present study is to design and implement a continuous trait-based model (CITRATE 1.0) at two representative stations in the North Pacific. The overall goal of this model is not only to simulate the phytoplankton size diversity but also to faithfully reproduce the seasonal and vertical dynamics of other important quantities such as nutrients Chl a, and productivity in for later investigations of the roles of phytoplankton diversity in biogeochemical cycles in different oceanic regions (using 3D regional/global simulations). Therefore, the two contrasted stations were used to provide a single set of parameters values by fitting the model results to observations before the obtained model was validated in another independent station (ALOHA). Hence, CITRATE 1.0 consists of the following key features:

*[Response] Thanks for the suggested revisions, which are really helpful. We have completely followed the suggestions.*

P. 21 L. 6-10: For consistency reasons, this part on the comparison of estimated growth rate with literature values should be moved P. 20 L. 21 together with the paragraph on the test of sensitivity of the growth rate.

*[Response] We are sorry that we might have caused confusion between*

*"u"and "µ". This part is actually on the sensitivity analysis of the trait diffusion coefficient "u", not on comparing simulated phytoplankton growth rates (µ) with those reported in the literature. To make it clearer, we have corrected the sentence to: "The optimized trait diffusion coefficient (u) was much higher than in Acevedo-Trejos et al. (2016)."*

**CITRATE 1.0: Phytoplankton continuous trait-distribution model with one-dimensional physical transport applied to the North Pacific**

Bingzhang Chen, S. Lan Smith

Research Center for Global Change Research, JAMSTEC (Japan Agency for Marine-Earth Science and Technology), 3173-25 Showa-machi, Kanazawa-ku, Yokohama 236-0001, Japan

*Correspondence to*: Bingzhang Chen (bzchen@jamstec.go.jp)

**Abstract.** Diversity plays critical roles in ecosystem functioning, but it remains challenging to model phytoplankton diversity in order to better understand those roles and reproduce consistently observed diversity patterns in the ocean. In contrast to the typical approach of resolving distinct species or functional groups, we present a ContInuous TRAiT-basEd phytoplankton model (CITRATE) that focuses on macroscopic system properties such as total biomass, mean trait values, and trait variance. This phytoplankton component is embedded within a Nitrogen-Phytoplankton-Zooplankton-Detritus-Iron model that itself is coupled with a simplified one-dimensional ocean model. Size is used as the master trait for phytoplankton. CITRATE also incorporates "trait diffusion" for sustaining diversity, as well as simple representations of physiological acclimation, i.e. flexible chlorophyll-to-carbon and nitrogen-to-carbon ratios. We have implemented CITRATE at two contrasting stations in the North Pacific where several years of observational data are available. The model is driven by physical forcing including vertical eddy diffusivity imported from three-dimensional general ocean circulation models (GCMs). One common set of model parameters for the two stations is optimized using the Delayed Rejection Adaptive Metropolis-Hasting Monte Carlo (DRAM) algorithm. The model faithfully reproduces most of the observed patterns and gives robust predictions on phytoplankton mean size and size diversity. CITRATE is suitable for applications in GCMs and constitutes a prototype upon which more sophisticated continuous trait-based models can be developed.


**1 Introduction**

[revised manuscript text omitted]

**2 Model description**

The aim of the present study is to design and implement a continuous trait-based model (**CITRATE** 1.0) at two representative stations in the North Pacific. The overall goal of this model is not only to simulate the phytoplankton size diversity but also to faithfully reproduce the seasonal and vertical dynamics of other important quantities such as nutrients, Chl $a$, and productivity for later investigations of the roles of phytoplankton diversity in biogeochemical cycles in different oceanic regions (using 3D regional/global simulations). Therefore, these two contrasting stations were used to provide a single set of parameters values by fitting the model to observations before the obtained model was validated against data from another independent station (ALOHA). 
[revised manuscript text omitted]
 models resolving a number of discrete species, the typical index for the intensity of resource competition under steady-state is R*, the lowest nutrient concentration allowing positive net growth (Tilman, 1982; Litchman et al., 2007; Barton et al., 2010). Under non-equilibrium conditions, it is the maximal growth rate instead of R* that determines the outcome of competition (Hutson, 1979;

10 Barton et al., 2010). In any case, it is the realised growth rate that determines the outcome of competition. Compared to R*, the second derivative $\frac{d^2\mu(l)}{dl^2}$ has two advantages as a proxy for quantifying the intensity of competition: 1) it applies under both equilibrium and non-equilibrium conditions and 2) it circumvents the problem of tracking many species. Using this approach, it is straightforward to test some ecological theories such as Huston's "general hypothesis of species

15 diversity" (Huston, 1979). For example, the absolute magnitude of $\frac{d^2\mu(l)}{dl^2}$ correlates positively with $\mu$ (Fig. 13), indicating that resource competition is more intense when growth rates are high. This is a mathematical manifestation of the verbal argument of the "dynamic equilibrium theory" proposed in Huston (1979), who emphasized that in natural environments where equilibrium is rarely achieved fast-growing species tend to outcompete slow-growing species (*see* also Barton et al., 2010), and hence

20 growth rates play a greater role in determining diversity than R* values.

Similarly, Eq. 7b specifies concisely the factors affecting mean phytoplankton size. In fact, Eqs. 7a-c can be understood as derived from a Taylor expansion representing an infinite number of discrete trait classes (Merico et al. 2009). Hence, even if a discrete version of a diversity model is used, it may



[revised manuscript text omitted]

**4.2 Model limitations**

**4.2.1 Assumption of trait distribution**

To facilitate calculation of trait moments, a certain distribution has to be assumed for the trait (Merico et al., 2009, 2014). The lognormal distribution can be fitted well to empirical data (Quintana et al., 2008, 2016), and because of its mathematical convenience it has been widely used in continuous size distribution models (Terseleer et al., 2014; Acevedo-Trejos et al., 2015, 2016; Smith et al., 2016). For these reasons we have assumed a lognormal distribution in the present study.

However, other probability distributions can also describe phytoplankton size. In the literature, phytoplankton abundance ($N$, cells $L^{-1}$) within the size interval from $V$ to $V + dV$ is more often modelled as a power-law function of cell volume $V$ (unit: $\mu m^3$; Gin et al., 1999; Cavender-Bares et al., 2001; Cermeño et al., 2006):

$$N(V) = N_0 V^\alpha \qquad (16a)$$

where $N_0$ represents the abundance of phytoplankton having cell volume 1 $\mu m^3$ and $\alpha$ is the exponent of the power law. Because models typically represent phytoplankton biomass, instead of abundance, we can convert the abundance to biomass ($B$, $\mu m^3 L^{-1}$):

$$B(V) = N(V)V = N_0 V^{\alpha+1} \qquad (16b)$$

Although the power law of Eq. (16b) may seem to be a suitable alternative distribution for continuous size-based models, empirical data suggest that $\alpha$ tends to vary between –0.7 and –1 (Cermeño et al., 2006), which means that the exponent ($\alpha + 1$) of the power law relating $B$ and $V$ should be between 0 and 0.3. In this case both the mean and variance of the power law distribution as shown in Eq. (16b) are infinite (Newman, 2005). This problem can be solved by adding an upper cutoff via an exponential truncation (Clauset et al., 2009):

$$B(V) = N_0 V^{\alpha+1} e^{-\lambda V} \qquad (16c)$$

where $\lambda$ is a positive constant.

Whether the power law or the lognormal distribution fits better to empirical data has been widely debated in the literature, and many results show that both can fit the data equally well (Allen et al., 2001; Mitzenmacher, 2004; Clauset et al., 2009). This is not surprising given that the two distributions are intrinsically connected (Mitzenmacher, 2004; Newman, 2005). We suspect that the power law with an

upper cutoff may be able to capture better the right skewness of phytoplankton size distributions, as is common in oligotrophic waters where large diatoms coexist with the dominant cyanobacteria such as *Prochlorococcus* (Campbell et al., 1994; Liu et al., 1997; Villareal et al., 1999). It remains to be investigated whether changing the distribution to the truncated power law can help solve the problem of

5  underestimating the fraction of $> 10 \mu$m size in the current study.

Neither the lognormal nor the power law with an upper cutoff can capture multimodal size distributions, as exemplified in fig. 1b of Marañón (2015) and reported by other studies (Banas, 2011; Bonachela et al., 2016; Coutinho et al., 2016). This is an inevitable consequence of aggregating the description of the entire community into only the three descriptors (i.e. total biomass, mean and

[revised manuscript text omitted]

[revised manuscript text omitted]

We believe that t

|---|---|---|

overweighs other technical advantages such a

|---|---|---|

can

can

equations themselves (Eq. 1) have already provided the genuine insights for

for

at mean size

at mean size

字体斜体

since

Thus, we propose that

|---|---|---|

Thus, we propose that

|---|---|---|

Thus, we propose that

|---|---|---|

Thus, we propose that

|---|---|---|

Thus, we propose that

|---|---|---|

Thus, we propose that

|---|---|---|

Thus, we propose that

|---|---|---|

a

|---|---|---|

a

a

'adaptive dynamics', it is easier to ability to easily quantify competition intensity (and other ecological quantities), which makes

positively

positively

the

the

This is because when environmental conditions favour highfast growth rates, it takes less time for the dominant species to predominate, thusand reducing diversitydiversity decreases. The positive correlation between the absolute value of

This is because when environmental conditions favour highfast growth rates, it takes less time for the dominant species to predominate, thusand reducing diversitydiversity decreases. The positive correlation between the absolute value of

also gives

also gives

|---|---|---|

also gives

|---|---|---|

字体斜体

|---|---|---|

also

also

also

evolutionary theory

and

and

a population with the

a population with the

字体斜体

字体斜体

字体斜体

字体斜体

字体斜体

字体斜体

字体斜体

字体斜体

字体斜体

字体斜体

字体斜体

字体斜体

字体斜体

Compared to previous continuous trait-based models (Terseleer et al., 2014; Acevedo-Trejos et al., 2015, 2016), **CITRATE** 1.0

Compared to previous continuous trait-based models (Terseleer et al., 2014; Acevedo-Trejos et al., 2015, 2016), **CITRATE** 1.0

Compared to previous continuous trait-based models (Terseleer et al., 2014; Acevedo-Trejos et al., 2015, 2016), **CITRATE** 1.0

the

the

the

the

of

of

of

.

.

.

opposite

opposite

opposite

opposite

also

also

also

also

also

also

also

also

also

also

also

also

also

also

also

also

,

,

，

，

，

，

，

，

addition

addition

addition

addition

. And also b

. And also b

. And also b

. And also b

| --- | --- | --- |

lognormal is not the only

| --- | --- | --- |

I

| --- | --- | --- |

字体斜体

| 页 30: [60] 带格式的 | **Bingzhang Chen** | **2017/11/24 AM11:13:00** |

字体斜体

usually

| 页 30: [62] 带格式的 | **Bingzhang Chen** | **2017/11/24 AM11:22:00** |

字体斜体

| 页 30: [63] 带格式的 | **Bingzhang Chen** | **2017/11/24 AM11:13:00** |

字体斜体

字体斜体

with

the common unit of

右对齐, 缩进 首行缩进 2.36 字符, 允许文字在单词中间换行

|---|---|---|

While

|---|---|---|

While

|---|---|---|

While

|---|---|---|

While

|---|---|---|

While

|---|---|---|

While

|---|---|---|

The slope of log abundance versus log size (e.g. cell volume)

with the consequence tha

右对齐, 缩进 首行缩进 1.18 字符, 允许文字在单词中间换行

$$B(V) = N \quad 0 \quad V \quad \alpha+1 \quad e \quad -\gamma V \quad B(V) = N \quad 0 \quad V \quad \alpha+1 \quad e \quad -\lambda V$$

| 页 30: [72] 批注 | S. Lan Smith | 2017/11/27 AM11:49:00 |

Is this term simply added to the eq. above? If so, I'd suggest writing out the full equation.

字体斜体

In the literature, there have been lots of debates on whether

might

might

might

might

might

unicellular

unicellular

unicellular

| 页 31: [77] 删除的内容 | **Bingzhang Chen** | **2017/11/24 AM11:59:00** |

However, aside from fact that the power-law distribution is unrealistic in predicting phytoplankton biomass at the size limits (there must be upper and lower limits of size at which phytoplankton biomass becomes close to zero, which cannot be predicted bywhich the power-law cannot reproduce), the power-law distribution is much more inconvenient in terms of for mathematical manipulations (e.g. calculating mean and variance) thancompared to the normal distribution. TheA lognormal distribution is usually appropriatea much better alternative for phytoplankton size in terms of mathematic properties (e.g., zero probability of negative size) and can be fit well to empirical data (Finkel, 2007Quintana et al., 2008). Therefore, it is not surprising that the lognormal distribution has been widely used in continuous size models; (Terseleer et al., 2014; Acevedo-Trejos et al., 2015, 2016; Smith et al., 2016). However, this does not guarantee that a fixed type of probability distribution can hold for all situations (Coutinho et al., 2016). In oligotrophic waters where picophytoplankton, particularly the unicellular cyanobacteria *Prochlorococcus* and *Synechococcus*, dominate (Campbell et al., 1994; Liu et al., 1997), the distribution of phytoplankton log size is more likely right skewed. In other words, abundances of

large species are higher than expected from a pure lognormal distribution, which is consistent with the observation that some large diatoms, with significant contributions to new production, can be found in the oligotrophic gyres (Villareal et al., 1999). This is probably one of the major reason that our model tends to underestimate the fraction of $> 10 \ \mu$m size.

|---|---|---|

ces

|---|---|---|

ces

|---|---|---|

ces

|---|---|---|

ces

|---|---|---|

ces

|---|---|---|

ces

ces

The immediate summation of

The immediate summation of

The immediate summation of

usually

usually

usually

| --- | --- | --- |

| 页 31: [82] 删除的内容 | **S. Lan Smith** | **2017/11/27 AM11:55:00** |
| --- | --- | --- |

al

| --- | --- | --- |

字体非粗体, 斜体

| 页 31: [84] 删除的内容 | **S. Lan Smith** | **2017/10/22 PM11:51:00** |
| --- | --- | --- |

From t

From t

) should be similar due to strong mixing, the approximation is not a big concern. The problem mainly lies at the bottom of the mixed layer where there is a sharp gradient of phytoplankton properties particularly biomass (Fig. 12c). We can roughly estimate that, based on the eddy diffusivity ($\sim$2 x $10^5$ m$^2$ s$^{-1}$), grid distance (20 m), and the phytoplankton biomass differences (0.2 $\mu$mol N L$^{-1}$), the transported phytoplankton biomass from the upper grid (higher *P*) to the lower grid (lower *P*) at one time step (0.5 min) is roughly 0.0002 $\mu$mol N L$^{-1}$, 1% of the *P* in the lower grid. Therefore, oOur tentative conclusion is that

[revised manuscript text omitted]

d

|---|---|---|

1.34

|---|---|---|

| $\alpha_\mu$ | First-order size scaling component for $\mu_\mathrm{m}$ | $0.25^e$ (0.1, 0.4) | 0.27 (0.005) | dimensio |
| $\beta_\mu$ | Second-order size scaling component for $\mu_\mathrm{m}$ | $-0.025^e$ (−0.05, 0) | −0.013 (0.0002) | dimensio |

|---|---|---|

f

|---|---|---|

f

|---|---|---|

058

|---|---|---|

058

|---|---|---|

058

|---|---|---|

| $\alpha_{fer}$ | Size scaling exponent for $K_{0,fer}$ | $0.27^c$ (0.1, 0.3) | 0.30 (0.001) | dimensio |
|---|---|---|---|---|

|---|---|---|

g

|---|---|---|

g

|---|---|---|

g

|---|---|---|

092

|---|---|---|

092

|---|---|---|

092

|---|---|---|

| *dustso* | Dust iron solubility | | $0.02^h$ (0.01, 0.05) | 0.022 (0.0007) | dimensio |
|---|---|---|---|---|---|

|---|---|---|

Marañón et al., (2013); [b]

|---|---|---|

Marañón et al., (2013); [b]

|---|---|---|

Marañón et al., (2013); [b]

|---|---|---|

Marañón et al., (2013); [b]

Marañón et al., (2013); [b]

[Figure]

[Figure]

Fig. 6 . Model fittings to vertical profiles of TIN, CHL, NPP, and PON at K2

[Figure]

Fig. 8 . Model fittings to vertical profiles of four size fractions at K2

[Figure]

Fig. 10 . Modelled seasonal patterns at K2

[Figure]

Fig. 12